# EXTRACTING ROBUST MODELS WITH UNCERTAIN EXAMPLES

**Guanlin Li**[1,2], **Guowen Xu**[1,*], **Shangwei Guo**[3], **Han Qiu**[4,5], **Jiwei Li**[6,7], **Tianwei Zhang**[1]
[1]Nanyang Technological University, [2]S-Lab, NTU, [3]Chongqing University,[4]Tsinghua University
[5]Zhongguancun Laboratory, [6]Shannon.AI, [7]Zhejiang University. * Corresponding author
`guanlin001@e.ntu.edu.sg`, {`guowen.xu, tianwei.zhang`}`@ntu.edu.sg`,
`swguo@cqu.edu.cn, qiuhan@tsinghua.edu.cn, jiwei_li@shannonai.com`

## ABSTRACT

Model extraction attacks are proven to be a severe privacy threat to Machine Learning as a Service (MLaaS). A variety of techniques have been designed to steal a remote machine learning model with high accuracy and fidelity. However, how to extract a robust model with similar resilience against adversarial attacks is never investigated. This paper presents the *first* study toward this goal. We first analyze that those existing extraction solutions either fail to maintain the model accuracy or model robustness, or lead to the robust overfitting issue. Then we propose Boundary Entropy Searching Thief (`BEST`), a novel model extraction attack to achieve both accuracy and robustness extraction under restricted attack budgets. `BEST` generates a new kind of *uncertain examples* for querying and reconstructing the victim model. These samples have uniform confidence scores across different classes, which can perfectly balance the trade-off between model accuracy and robustness. Extensive experiments demonstrate that `BEST` outperforms existing attack methods over different datasets and model architectures under limited data. It can also effectively invalidate state-of-the-art extraction defenses. Our codes can be found in `https://github.com/GuanlinLee/BEST`.

## 1 INTRODUCTION

Recent advances in deep learning (DL) and cloud computing technologies boost the popularity of Machine Learning as a Service (MLaaS), e.g., AWS SageMaker (sag, 2022), Azure Machine Learning (azu, 2022). This service can significantly simplify the DL application development and deployment at a lower cost. Unfortunately, it also brings new privacy threats: an adversarial user can query a target model and then reconstruct it based on the responses (Tramèr et al., 2016; Orekondy et al., 2019; Jagielski et al., 2020b; Yuan et al., 2020; Yu et al., 2020). Such model extraction attacks can severely compromise the intellectual property of the model owner (Jia et al., 2021), and facilitate other black-box attacks, e.g., data poisoning (Demontis et al., 2019), adversarial examples (Ilyas et al., 2018), membership inference (Shokri et al., 2017).

Existing model extraction attacks can be classified into two categories (Jagielski et al., 2020b). (1) *Accuracy extraction* aims to reconstruct a model with similar or superior accuracy compared with the target model. (2) *Fidelity extraction* aims to recover a model with similar prediction behaviors as the target one. In this paper, we propose and consider a new category of attacks: *robustness extraction*. As DNNs are well known to be vulnerable to adversarial attacks (Szegedy et al., 2014), it is common to train highly robust models for practical deployment, especially in critical scenarios such as autonomous driving (Shen et al., 2021), medical diagnosis (Rendle et al., 2016) and anomaly detection (Goodge et al., 2020). Then an interesting question is: *given a remote robust model, how can the adversary extract this model with similar robustness as well as accuracy under limited attack budgets?* We believe this question is important for two reasons. (1) With the increased understanding of adversarial attacks, it becomes a trend to deploy robust machine learning applications in the cloud (Goodman & Xin, 2020; Rendle et al., 2016; Shafique et al., 2020), giving the adversary opportunities to steal the model. (2) Training a robust model usually requires much more computation resources and data (Schmidt et al., 2018; Zhao et al., 2020), giving the adversary incentives to steal the model.

We review existing attack techniques and find that they are incapable of achieving this goal, unfortunately. Particularly, there can be two kinds of attack solutions. (1) The adversary adopts clean samples to query and extract the victim model (Tramèr et al., 2016; Orekondy et al., 2019; Pal et al., 2020). However, past works have proved that *it is impossible to obtain a robust model only from clean*

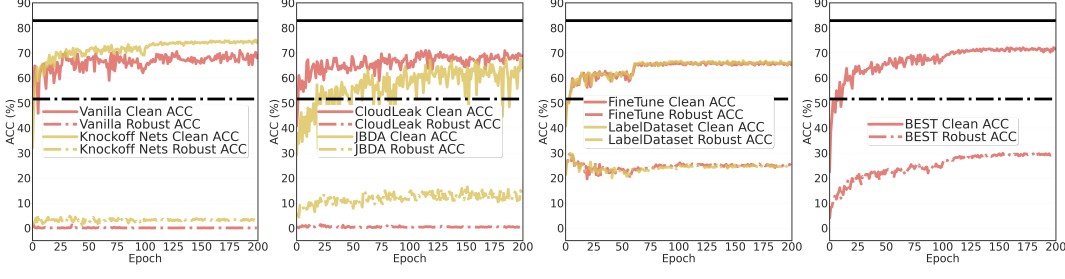

| (a) Extraction-clean sample | (b) Extraction-AE | (c) Extraction-AT | (d) BEST |

Figure 1: Model extraction results on CIFAR10. The victim model is ResNet18 trained by PGD-AT on CIFAR10. The adversary model is ResNet18. Black solid and dashed lines in each figure denote the clean and robust accuracy of the victim model.

*data* (Zhao et al., 2020; Rebuffi et al., 2021). Thus, these methods cannot preserve the robustness of a robust victim model, although they can effectively steal the model's clean accuracy. (2) The adversary crafts adversarial examples (AEs) to query and rebuild the victim model (Papernot et al., 2017; Yu et al., 2020). Unfortunately, building models with AEs leads to two unsolved problems: (1) *improving the robustness with AEs inevitably sacrifices the model's clean accuracy* (Tsipras et al., 2019); (2) *with more training epochs, the model's robustness will decrease as it overfits the generated AEs* (Rice et al., 2020). We will conduct experiments to validate the limitations of prior works in Section 3.

To overcome these challenges in achieving robustness extraction, we design a new attack methodology: **B**oundary **E**ntropy **S**earching **T**hief (BEST). The key insight of BEST is the introduction of uncertain examples (UEs). These samples are located close to the junctions of classification boundaries, making the model give uncertain predictions. We synthesize such samples based on their prediction entropy. Using UEs to query the victim model, the adversary can asymptotically shape the classification boundary of the extracted model following that of the victim model. With more extraction epochs, the boundaries of the two models will be more similar, and the overfitting phenomenon will be mitigated. We perform comprehensive experiments to show that BEST outperforms different types of baseline methods over various datasets and models. For instance, BEST can achieve 13% robust accuracy and 8% clean accuracy improvement compared with the JBDA attack (Papernot et al., 2017) on CIFAR10.

## 2 THREAT MODEL

We consider the standard MLaaS scenario, where the victim model $M_{\mathcal{V}}$ is deployed as a remote service for users to query. We further assume this model is established with adversarial training (Madry et al., 2018; Zhang et al., 2019; Li et al., 2022), and exhibits certain robustness against AEs. We consider adversarial training as it is still regarded as the most promising strategy for robustness enhancement, while some other solutions (Xu et al., 2017; Zhang et al., 2021a; Gu & Rigazio, 2014; Papernot et al., 2017) were subsequently proved to be ineffective against advanced adaptive attacks (Athalye et al., 2018; Tramer et al., 2020). We will consider more robustness approaches in future work (Section 7). An adversarial user $\mathcal{A}$ aims to reconstruct this model just based on the returned responses. The extracted model $M_{\mathcal{A}}$ should have a similar prediction performance as the target one, for both clean samples (clean accuracy) and AEs (robust accuracy). $\mathcal{A}$ has no prior knowledge of the victim model, including the model architecture, training algorithms, and hyperparameters. He is not aware of the adversarial training strategy for robustness enhancement, either. $\mathcal{A}$ can adopt a different model architecture for building $M_{\mathcal{A}}$, which can still achieve the same behaviors as the target model $M_{\mathcal{V}}$.

Prior works have made different assumptions about the adversary's knowledge of query samples. Some attacks assume the adversary has access to the original training set (Tramèr et al., 2016; Jagielski et al., 2020b; Pal et al., 2020), while some others assume the adversary can obtain the distribution of training samples (Papernot et al., 2017; Orekondy et al., 2019; Chandrasekaran et al., 2020; Yu et al., 2020; Pal et al., 2020). Different from those works, we consider a more practical adversary's capability: the adversary only needs to collect data samples from the same task domain of the victim model, which do not necessarily follow the same distribution of the victim's training set. This is feasible as the adversary knows the task of the victim model, and he can crawl relevant images from the Internet. More advanced attacks (e.g., data-free attacks (Truong et al., 2021; Kariyappa et al., 2021)) will be considered as future work.

The adversary can collect a small-scale dataset $D_{\mathcal{A}}$ with such samples to query the victim model $M_{\mathcal{V}}$. We consider two practical scenarios for the MLaaS: the service can return the predicted logits vector $\mathcal{Y}$ (Tramèr et al., 2016; Orekondy et al., 2019; Jagielski et al., 2020b; Pal et al., 2020) or a hard label

$Y$ (Tramèr et al., 2016; Papernot et al., 2017; Jagielski et al., 2020b; Pal et al., 2020) for every query sample. For each case, our attack is able to extract the model precisely.

**Attack cost.** Two types of attack budgets are commonly considered in model extraction attacks. (1) Query budget $B_Q$: this is defined as the number of queries the adversary sends to the victim model. As commercial MLaaS systems adopt the pay-as-you-use business scheme, the adversary wishes to perform fewer queries while achieving satisfactory attack performance. (2) Synthesis budget $B_S$: this is defined as the computation cost (e.g., number of optimization iterations) to generate each query sample. A smaller $B_S$ will be more cost-efficient to the adversary and reduce the attack time. The design of a model extraction attack needs to consider the reduction of both budgets.

To reduce the attack cost, we assume the adversary can download a public pre-trained model and then build the extracted model from it. This assumption is reasonable, as there are lots of public model zoos offering pre-trained models of various AI tasks (e.g., Hugging Face (hug, 2022) and ModelZoo (mod, 2022)). It is also adopted in (Yu et al., 2020), and justified in (Jagielski et al., 2020a). The training set of the pre-trained model can be totally different from that of the victim model.

## 3 Existing Attack Strategies and Their Limitations

A variety of attack techniques have been proposed to extract models with high accuracy and fidelity, which can be classified into the following two categories.

**Extraction with Clean Samples**. The adversary samples query data from a public dataset offline and trains the extracted model based on the data and victim model's predictions. The earlier work (Tramèr et al., 2016) adopts this simple strategy, and we denote this attack as "Vanilla" in the rest of this paper. Later on, advanced attacks are proposed, which leverage active learning (Chandrasekaran et al., 2020) to generate samples for querying the victim model and refine the local copy iteratively. Typical examples include Knockoff Nets (Orekondy et al., 2019) and ActiveThief (Pal et al., 2020) attacks. The adversary gets a huge database of different natural images. For each iteration, he actively searches the best samples from this database based on his current model for extraction.

**Extraction with Adversarial Examples**. The adversary crafts AEs to identify the classification boundaries. A representative example is CloudLeak (Yu et al., 2020). The adversary generates AEs based on a local surrogate model as the query samples. These AEs with the victim model's predictions form the training set for the adversary to train the extracted model. Some attacks also combine active learning to iteratively generate AEs. For instance, in the JBDA attack (Papernot et al., 2017), the adversary follows the FGSM (Goodfellow et al., 2015) idea to generate perturbed samples, queries the data, and then refines his local model repeatedly.

**Limitations**. These solutions may work well for accuracy or fidelity extraction. However, they are not effective in robustness extraction. We analyze their limitations from the following perspectives. First, according to previous studies (Zhao et al., 2020; Rebuffi et al., 2021), *it is impossible to train a robust model only with clean samples.* Therefore, the techniques using clean samples cannot steal the victim model's robustness. To confirm this conclusion, we train a robust model using the PGD-AT approach (Madry et al., 2018). This model adopts the ResNet-18 architecture (He et al., 2016) and is trained over CIFAR10. The black solid and dashed lines in Figure 1a denote the clean accuracy and robust accuracy of this model. We consider the scenario where this model only returns the predicted hard label for each query[1]. Then we adopt the Vanilla and Knockoff Nets attack techniques to extract this model using the samples from part of the CIFAR10 test set[2]. Figure 1a shows the model accuracy over different extraction epochs, which is evaluated by another part of the CIFAR10 test set, disjoint with the extraction set. We observe that for these two approaches, the clean accuracy of the stolen model is very high. However, the robust accuracy of the replicated model against the PGD20 attack is close to 0, which indicates the extracted model does not inherit the robustness from the victim model at all.

Second, we consider the techniques based on AEs. Training (extracting) models with AEs can incur two unsolved issues. (1) *The participation of AEs in model training can sacrifice the model's clean accuracy* (Tsipras et al., 2019). Figure 1b shows the attack results of CloudLeak and JBDA. We observe that the clean accuracy of the extracted model from JBDA drops significantly compared

---

[1]For all model extraction attacks in Figure 1, we adopt the same pre-trained model, which is trained on Tiny-Imagenet, to initialize the adversary's local model as an extraction start. More details of the pre-trained model can be found in Appendix B.1.

[2]ActiveThief adopts the similar active technique as Knockoff Nets, so we omit its results here.

to the extracted accuracy from Vanilla and Knockoff Nets (Figure 1a). (2) *Training a model with AEs can easily make the model overfit over the training data (Rice et al., 2020), which significantly decreases the robustness of the adversary's model with more query samples.* From Figure 1b, we observe the robust accuracy in JBDA decreases before the extraction is completed. It is worth noting that the clean and robust accuracy of CloudLeak is very close to the clean sample based approaches (Vanilla and Knockoff Nets). The reason is that pre-generated AEs have lower transferability towards the victim model compared to queries in active learning, and can still be regarded as clean samples.

Third, we further consider a straightforward strategy specifically for the robustness extraction scenario: the adversary conducts the accuracy extraction of the victim model, followed by adversarial training to obtain the robustness. We implement two attacks following this strategy: (1) *LabelDataset*: the adversary first obtains a labeled dataset from the victim model with Vanilla or CloudLeak, and then adopts adversarial training to train his model. (2) *FineTune*: the adversary first extracts the model with CloudLeak, and then fine-tunes it with AEs. We adopt the same training hyperparameters and protocol in (Rice et al., 2020). Furthermore, to avoid potential overfitting in the adversarial training process, we adopt Self-Adaptive Training (SAT) (Huang et al., 2020) combining with PGD-AT (Madry et al., 2018) in both attacks. The hyperparameters of SAT follow its paper. Figure 1c shows the extraction

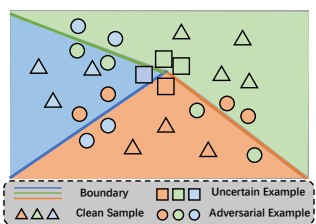

Figure 2: An illustration of clean samples, AEs and UEs. Each color represents a class.

results of these two attacks. We observe that their clean accuracy is still compromised, and robust accuracy decays at the beginning (i.e., robust overfitting). The main reason is that the adversary does not have enough data to apply adversarial training due to the attack budget constraint (5,000 samples in our experiments), which could easily cause training overfitting and low clean accuracy. We provide more experiment results in Appendix C.7 to show the advantages of our BEST.

## 4 METHODOLOGY

We introduce a new attack methodology, which can extract both the clean accuracy and robustness of the victim model. Figure 1d shows the extraction results of our method under the same setting as other approaches. We observe it can effectively overcome the above challenges and outperform other attack strategies. Below we give the detailed mechanism and algorithm of our solution.

### 4.1 UNCERTAIN EXAMPLE

Our methodology is inspired by the limitations of AE-based attacks. It is well-known that AEs are very close to the model's classification boundaries, and can better depict the boundary shape (He et al., 2018). However, they also exhibit the $\gamma$-*useful robust feature* (Ilyas et al., 2019), i.e., having very high confidence scores (larger than $\gamma$) for the correct label. Such robust features can lead to clean accuracy degradation (Tsipras et al., 2019; Ilyas et al., 2019) as well as robust overfitting (Rice et al., 2020). Therefore, to precisely extract both the clean and robust accuracy of the victim model, the query samples should satisfy two properties: (**P1**) they cannot have the robust feature obtained from the AE generation process to avoid overfitting; (**P2**) they should reflect the shape of the model classification boundaries. These properties motivate us to design a new way to depict the victim model's boundary. We propose a novel **uncertain example** (UE), which can meet the above requirements and is qualified for model extraction. It is formally defined as follows.

**Definition 1** ($\delta$-**uncertain example**) *Given a model $M : \mathcal{R}^N \mapsto \mathcal{R}^n$, an input $x \in \mathcal{R}^N$ is said to be a $\delta$-uncertain example ($\delta$-UE), if it satisfies the following relationship:*

$$\text{softmax}(M(x))_{\text{max}} - \text{softmax}(M(x))_{\text{min}} \leq \delta \qquad (1)$$

Figure 2 illustrates the positions of clean samples, AEs and UEs. Compared to other types of samples, a UE aims to make the model confused about its label. Clearly, every sample in $\mathcal{R}^N$ is a 1-UE. To query and extract the victim model, we expect to make $\delta$ as small as possible. On one hand, a UE with a small $\delta$ does not have the robust feature, satisfying the property **P1**. On the other hand, a sample far away from the model's classification boundary normally has higher prediction confidence (Cortes & Vapnik, 1995; Mayoraz & Alpaydin, 1999). The uncertainty in the UEs makes them closer to the boundary, satisfying the property **P2**. Therefore, model extraction with UEs can better preserve the clean accuracy without causing robust overfitting, compared to AE-based approaches.

We propose **B**oundary **E**ntropy **S**earching **T**hief (BEST), a novel extraction attack based on UEs. Particularly, similar as (Pal et al., 2020; Papernot et al., 2017; Orekondy et al., 2019), we also adopt the active learning fashion, as it can give more accurate extraction results than the non-active attacks. In each iteration, the adversary queries the victim model with $\delta$-UEs. Based on the responses, he refines the local model to make it closer to the victim one in terms of both clean and robust accuracy.

For some active learning based attacks (e.g., Knockoff Nets, ActiveThief), the adversary searches for the best sample from a huge database for each query. This strategy is not feasible under our threat model, where the adversary has a limited number of data samples. Besides, it is hard to directly sample qualified $\delta$-UEs from the adversary's training set $D_{\mathcal{A}}$: according to previous studies (Li & Liang, 2018; Allen-Zhu et al., 2019), the trained model will gradually converge on the training set, making the training samples far from the boundary and reducing the chances of finding UEs under a small $\delta$. Instead, the adversary can synthesize UEs from natural data in each iteration. This can be formulated as a double-minimization problem, with the following objective:

$$\min_{M_{\mathcal{A}}} L(x, y, M_{\mathcal{A}}) \min_{x} (\mathrm{softmax}(M_{\mathcal{A}}(x))_{\max} - \mathrm{softmax}(M_{\mathcal{A}}(x))_{\min}). \tag{2}$$

In the inner minimization, we first identify the UE $x$ that makes its confidence variance as small as possible. Then in the outer minimization, we optimize the adversary's model $M_{\mathcal{A}}$ with such UE $x$ and its label $y$ obtained from the victim model's prediction to minimize the loss function $L$. Our UEs are generated based on the adversary's restored model $M_{\mathcal{A}}$. Because we want the adversary to modify classification boundaries as much as possible to get close to the victim model's boundaries, the information obtained from the victim model should be maximized, which can be achieved by querying the victim model with UEs.

Algorithm 1 describes the BEST attack in detail. We first define a new uncertain label $\mathcal{Y}_p$ with the same confidence score of each class (**Line 2**). In each iteration within the query budget $B_Q$, we collect some natural samples from the adversary's dataset $D_{\mathcal{A}}$ and synthesize the corresponding UEs. We adopt the Kullback-Leibler divergence $\mathrm{KLD}(\cdot, \cdot)$ to compute the distance between $\mathrm{softmax}(M_{\mathcal{A}}(X_i))$ and $\mathcal{Y}_p$ (**Line 7**), and apply the PGD technique (Madry et al., 2018) under the synthesis budget $B_S$ to make $\mathrm{softmax}(M_{\mathcal{A}}(X_i))$ closer to $\mathcal{Y}_p$, i.e., minimizing $\delta$ (**Line 8**). Then we query the victim model $M_{\mathcal{V}}$ with the generated UEs and obtain the corresponding responses (**Line 10**). Different from previous works, we only need to obtain the hard label from $M_{\mathcal{V}}$, which is enough for minimizing the Cross-Entropy loss $L$ for model training

---

**Algorithm 1** Boundary Entropy Searching Thief

1: **Input:** Adversary model $M_{\mathcal{A}}$, victim model $M_{\mathcal{V}}$, adversary dataset $D_{\mathcal{A}}$, # of classes $K$, query budget $B_Q$, synthesis budget $B_S$, hyperparameters $\epsilon, \eta$
2: $\mathcal{Y}_p = [\frac{1}{K}, \ldots, \frac{1}{K}]$, $b_q = 0$
3: **while** $b_q \leq B_Q$ **do**
4:    **for** $X \in D_{\mathcal{A}}$ **do**
5:       $X_0 = X + \text{Random Noise}$
6:       **for** $i$ in range $(B_S)$ **do**
7:          $\mathcal{L} = -\mathrm{KLD}(\mathrm{softmax}(M_{\mathcal{A}}(X_i)), \mathcal{Y}_p)$
8:          $X_{i+1} = \Pi_{X,\epsilon}(X_i + \eta * \mathrm{sgn}(\nabla_{X_i}\mathcal{L}))$
9:       **end for**
10:      $\mathcal{Y} = M_{\mathcal{V}}(X_{B_S})$
11:      $Y = \arg\max \mathcal{Y}$
12:      Train $M_{\mathcal{A}}$ with $(X_{B_S}, Y)$
13:      Update $b_q$
14:    **end for**
15: **end while**
16: **Return** $M_{\mathcal{A}}$

---

(**Line 12**), which makes it harder to defeat our attack, as it can invalidate the mainstream extraction defenses which perturb the logits vectors (Lee et al., 2019). Note that Line 12 represents the training process on a batch of data. Specifically, we use data $X$ with the batch size of 128 in our experiments, i.e., the adversary first queries the victim model with 128 data samples, uses the 128 sample-response pairs to train his local model, and then adds 128 to the query budget $b_q$.

## 5 EXPERIMENTS

### 5.1 CONFIGURATIONS

**Datasets and Models**. Our attack method is general for different datasets, models, and adversarial training strategies. Without loss of generality, we choose two datasets: CIFAR10 (Krizhevsky et al., 2009) and CIFAR100 (Krizhevsky et al., 2009), which are the standard datasets for adversarial training studies (Madry et al., 2018; Zhang et al., 2019; Rice et al., 2020; Jang et al., 2019; Raghunathan et al., 2018; Xiao et al., 2020; Balaji et al., 2019). Prior model extraction attacks adopt data samples from or following the same distribution of the victim's training set (Tramèr et al., 2016; Jagielski et al., 2020b; Pal et al., 2020; Papernot et al., 2017; Orekondy et al., 2019; Chandrasekaran et al.,

| Method | | Best model with the highest robustness | | | | | | Final model after extraction | | | | | |
|---|---|---|---|---|---|---|---|---|---|---|---|---|---|
| | | CA | rCA | RA | | | | CA | rCA | RA | | | |
| | | | | PGD20 | PGD100 | CW100 | AA | | | PGD20 | PGD100 | CW100 | AA |
| Label | Vanilla | 58.74 | 64.02 | 5.26 | 4.84 | 5.60 | 3.70 | 74.12 | 81.58 | 3.22 | 2.74 | 3.20 | 1.92 |
| | JBDA | 64.20 | 71.80 | 16.48 | 16.22 | 16.80 | 15.42 | 63.96 | 71.36 | 13.24 | 12.96 | 13.44 | 12.08 |
| | ARD | 57.82 | 65.86 | 29.98 | 29.88 | 29.22 | 27.94 | 51.86 | 58.18 | 19.36 | 19.10 | 19.64 | 18.38 |
| | IAD | 55.10 | 62.92 | 30.04 | 29.98 | 29.20 | 28.08 | 52.86 | 60.00 | 20.44 | 20.18 | 20.76 | 19.34 |
| | RSLAD | 54.72 | 63.46 | 30.10 | 29.9 | 29.20 | 28.18 | 53.44 | 60.48 | 21.96 | 21.86 | 22.34 | 21.08 |
| Logits | Vanilla | 76.14 | 82.78 | 1.82 | 1.50 | 1.08 | 0.46 | 75.84 | 82.22 | 1.80 | 1.30 | 1.06 | 0.36 |
| | JBDA | 69.74 | 76.94 | 10.38 | 9.54 | 10.68 | 7.94 | 67.78 | 74.18 | 7.84 | 7.22 | 8.18 | 5.76 |
| | ARD | 67.36 | 76.32 | 30.52 | 30.08 | 28.86 | 26.98 | 64.10 | 73.06 | 26.44 | 26.06 | 24.80 | 22.98 |
| | IAD | 63.90 | 71.84 | 28.48 | 28.22 | 26.54 | 24.68 | 58.18 | 65.76 | 21.90 | 21.58 | 20.14 | 18.42 |
| | RSLAD | 66.16 | 74.92 | 28.92 | 28.56 | 27.04 | 24.88 | 60.40 | 67.94 | 22.00 | 21.64 | 19.64 | 17.98 |
| BEST | | 71.24 | 82.12 | 29.88 | 29.68 | 30.56 | 29.20 | 71.50 | 82.34 | 29.76 | 29.54 | 30.52 | 29.12 |

Table 1: Model extraction attack results on CIFAR10.

2020; Yu et al., 2020; Pal et al., 2020), which may not be possible in some practical scenarios. Our attack only requires the data from the same task domain. In our implementation, we split the test sets of CIFAR10 and CIFAR100 into two disjoint parts: (1) an extraction set $D_{\mathcal{A}}$ is used by the adversary to steal the victim model; (2) a validation set $D_{\mathcal{T}}$ is used to evaluate the attack results and the victim model's performance during its training process. Both $D_{\mathcal{A}}$ and $D_{\mathcal{T}}$ contain 5,000 samples. We also evaluate other types of extraction set in Section 5.3. The adversary adopts pre-trained models of various architectures trained over Tiny-ImageNet. The benefit from the pre-trained models is evaluated in Appendix C.3. We consider two types of extraction outcomes. (1) *Best model with the highest robustness*: the adversary picks the model with the highest robustness (against PGD20) during extraction. (2) *Final model after extraction*: the adversary picks the model from the last epoch.

The victim model $M_{\mathcal{V}}$ is selected from ResNet-18 (ResNet) (He et al., 2016) or WideResNet-28-10 (WRN) (Zagoruyko & Komodakis, 2016). The adversary model $M_{\mathcal{A}}$ may be different from the victim model, and we use two more model structures in our experiments, i.e., MobileNetV2 (MobileNet) (Sandler et al., 2018), and VGG19BN (VGG) (Simonyan & Zisserman, 2015). We adopt two mainstream adversarial training approaches, i.e., PGD-AT (Madry et al., 2018) and TRADES (Zhang et al., 2019), to enhance the robustness of victim models. This results in ResNet-AT (WRN-AT) and ResNet-TRADES (WRN-TRADES), respectively. The clean and robust accuracy of the victim models can be found in Appendix B.3.

**Baselines**. We adopt five different baseline methods for comparison. The first two are representatives of model extraction attacks discussed in Section 3. (1) `Vanilla` (Tramèr et al., 2016) is the most basic extraction technique using clean samples to query the victim model. (2) `JBDA` (Papernot et al., 2017) leverages active learning to generate AEs, which gives the best extraction performance over other methods in Section 3. We also choose three robust knowledge distillation methods as our baselines: (3) `ARD` (Goldblum et al., 2020), (4) `IAD` (Zhu et al., 2022) and (5) `RSLAD` (Zi et al., 2021). Robust knowledge distillation aims to train a student model from a large teacher model, where the student model can obtain better robustness than the teacher model. This is very similar to our robustness extraction goal. However, it requires the user to have the entire training set, as well as white-box access to the teacher model, which disobeys our threat model. So, we modify these methods with the same knowledge of the victim model and dataset for fair comparisons. For the details of robust knowledge distillation methods, we introduce them in Appendix A.2 and explain the differences between knowledge distillation methods and model extraction attacks. Due to the page limit, we leave the baseline details and settings of these attacks in Appendix B.1.

**Metrics**. We consider three metrics to comprehensively assess the attack performance. For clean accuracy evaluation, we measure **clean accuracy** (*CA*), which is the accuracy of the extracted models over clean samples in $D_{\mathcal{T}}$. We also consider **relative clean accuracy** (*rCA*), which checks whether the $M_{\mathcal{A}}$ gives the same label as $M_{\mathcal{V}}$ for each $x_i$ in $D_{\mathcal{T}}$. For robustness evaluation, we measure **robust accuracy** (*RA*) against various adversarial attacks. We choose four $L_{\infty}$-norm non-targeted attacks: PGD20, PGD100 (Madry et al., 2018), CW100 (Carlini & Wagner, 2017) and AutoAttack (AA) (Croce & Hein, 2020). The attack settings are $\epsilon = 8/255$ and $\eta = 2/255$. The number of attack steps is 20 for PGD20, and 100 for PGD100 and CW100. The results under $L_2$-norm attacks can be found in Appendix C.1. The formal definitions of our metrics can be found in Appendix B.4.

## 5.2 MAIN RESULTS

**Comparisons with Baselines**. We compare the attack effectiveness of `BEST` with other baselines. Due to page limit, we only show the results under one configuration: $M_{\mathcal{V}}$ is ResNet-AT, $M_{\mathcal{A}}$ is ResNet, and the dataset is CIFAR10. The other configurations give the same conclusion, and the results can be found in Appendix C.8. Table 1 shows the comparison results. First, our `BEST` generally performs much better than the baseline methods. It has similar clean accuracy *CA* as `Vanilla`, which is significantly higher than other methods. For robustness, it also outperforms these baselines, especially `Vanilla`. `Vanilla` can only obtain clean accuracy, but not

| Dataset | Adversary Model | Extracted model with the highest robustness | | | | | | Final model after extraction | | | | | |
|---|---|---|---|---|---|---|---|---|---|---|---|---|---|
| | | CA | rCA | RA | | | | CA | rCA | RA | | | |
| | | | | PGD20 | PGD100 | CW100 | AA | | | PGD20 | PGD100 | CW100 | AA |
| CIFAR10 | ResNet | 71.24 | 82.12 | 29.88 | 29.68 | 30.56 | 29.20 | 71.50 | 82.34 | 29.76 | 29.54 | 30.52 | 29.12 |
| | WRN | 74.30 | 84.64 | 31.90 | 31.70 | 32.52 | 31.22 | 73.40 | 83.86 | 31.24 | 31.08 | 31.84 | 30.72 |
| | VGG | 66.84 | 77.18 | 27.54 | 27.46 | 27.88 | 26.76 | 66.34 | 76.02 | 25.42 | 25.34 | 25.68 | 24.68 |
| | MobileNet | 68.50 | 78.54 | 26.52 | 26.22 | 26.96 | 25.72 | 68.70 | 78.48 | 24.74 | 24.46 | 25.44 | 23.88 |
| CIFAR100 | ResNet | 44.38 | 64.50 | 14.26 | 14.22 | 15.24 | 13.74 | 44.78 | 65.96 | 13.72 | 13.52 | 14.56 | 13.14 |
| | WRN | 47.20 | 68.56 | 15.78 | 15.66 | 17.00 | 15.02 | 46.90 | 67.82 | 15.40 | 15.24 | 16.62 | 14.68 |
| | VGG | 30.64 | 44.60 | 10.86 | 10.64 | 11.70 | 10.24 | 31.58 | 45.62 | 10.10 | 9.98 | 10.90 | 9.54 |
| | MobileNet | 41.48 | 60.74 | 12.84 | 12.68 | 13.44 | 12.20 | 41.40 | 60.84 | 12.64 | 12.50 | 13.34 | 12.04 |

Table 2: Results of `BEST` under different adversary model architectures. Victim model is ResNet-AT.

| Dataset | Victim Model | Extracted model with the highest robustness | | | | | | Final model after extraction | | | | | |
|---|---|---|---|---|---|---|---|---|---|---|---|---|---|
| | | CA | rCA | RA | | | | CA | rCA | RA | | | |
| | | | | PGD20 | PGD100 | CW100 | AA | | | PGD20 | PGD100 | CW100 | AA |
| CIFAR10 | ResNet-AT | 71.24 | 82.12 | 29.88 | 29.68 | 30.56 | 29.20 | 71.50 | 82.34 | 29.76 | 29.54 | 30.52 | 29.12 |
| | ResNet-TRADES | 68.60 | 78.78 | 28.82 | 28.62 | 29.30 | 28.36 | 70.06 | 79.58 | 28.10 | 27.90 | 28.66 | 27.52 |
| | WRN-AT | 72.06 | 78.84 | 26.58 | 26.24 | 27.22 | 25.72 | 72.56 | 79.40 | 25.86 | 25.64 | 26.44 | 25.20 |
| | WRN-TRADES | 70.20 | 78.34 | 28.44 | 28.28 | 29.04 | 27.82 | 69.70 | 78.26 | 27.90 | 27.74 | 28.56 | 27.26 |
| CIFAR100 | ResNet-AT | 44.38 | 64.50 | 14.26 | 14.22 | 15.24 | 13.74 | 44.78 | 65.96 | 13.72 | 13.52 | 14.56 | 13.14 |
| | ResNet-TRADES | 41.86 | 60.44 | 12.64 | 12.48 | 13.28 | 12.14 | 42.30 | 60.74 | 12.26 | 12.18 | 12.98 | 11.78 |
| | WRN-AT | 45.52 | 60.96 | 12.36 | 12.20 | 13.34 | 11.72 | 45.86 | 61.12 | 11.78 | 11.52 | 12.66 | 11.10 |
| | WRN-TRADES | 42.80 | 58.26 | 12.32 | 12.06 | 12.96 | 11.56 | 43.04 | 58.18 | 12.00 | 11.88 | 12.70 | 11.42 |

Table 3: Results of `BEST` under different victim model architectures and adversarial training approaches. The adversary model is ResNet.

robustness. Other baseline methods can increase the robustness of the extracted model, but their (relative) clean accuracy *(r)CA* is much lower. Particularly for the robust knowledge distillation methods, when we restrict the adversary to have black-box access towards the victim model and limited test samples, they cannot obtain the expected performance for both clean and robust accuracy. Second, for most baseline methods, the final model has lower robust accuracy *RA*, compared to the best model during extraction. It is caused by the robust overfitting issue in the training process (Rice et al., 2020). In contrast, our `BEST` can reduce the accuracy gap between the best model and the final model. Because the UEs generated in `BEST` give the extracted model a lower risk to overfit data. Third, when the victim model returns logits vectors, the (relative) clean accuracy *(r)CA* of the baseline methods increases, while the model robustness decreases. Because the robust features make the victim model give more uncertain predictions, and learning such logits vectors directly is more difficult. Our `BEST` does not depend on the returned prediction type.

**Impact of Model Architecture and Adversarial Training Strategy**. We first consider the case where the adversary adopts different model architectures from the victim. Table 2 shows the results when we vary the architecture of the adversary model. We observe that our methodology enables the adversary to obtain the maximal performance within the selected architecture. The clean and robust accuracy of VGG and MobileNet is a bit lower than ResNet and WRN, which is caused by the capability of the architectures themselves. Table 3 shows the attack performance against different victim model architectures with different adversarial training approaches. We observe that the attack performance is very stable across different configurations, and the deviations of *rCA* and robustness *RA* do not exceed 7% and 4%, respectively.

**Impact of Attack Budgets**. We first explore how the query budget $B_Q$ can affect the performance of our `BEST`. We perform model extraction with different sizes of $D_A$ from 1,000 to 5,000. Figure 3a shows the clean and robust accuracy trends during the extraction under different query budgets. We clearly observe that a larger $B_Q$ can increase both the clean and robust accuracy. Importantly, even using a very small $D_A$, the overfitting issue does not occur at the end of the attack, which indicates our method is stable and powerful. We give a detailed analysis in Appendix C.2.

We further consider the impact of the synthesis budget. We vary the value of $B_S$ and measure the relative clean accuracy *rCA* and robust accuracy *RA* against AA for the extracted model with the highest robustness during extraction. The results are shown in Figure 3b. First, we observe that our `BEST` can achieve excellent attack performance even under a very low synthesis budget. Second, a larger $B_S$ will not increase the *rCA* and *RA* significantly. We think this is because it is easy to generate query samples, and increasing $B_S$ does not improve the quality or quantity of the query samples with a smaller $\delta$. This indicates our attack is much more efficient than previous works, which rely on larger synthesis budgets. We give a more detailed analysis in Appendix C.2, where we use a single V100 GPU card to generate all required data in one epoch, and present the GPU time (in seconds) to prove our method's efficiency.

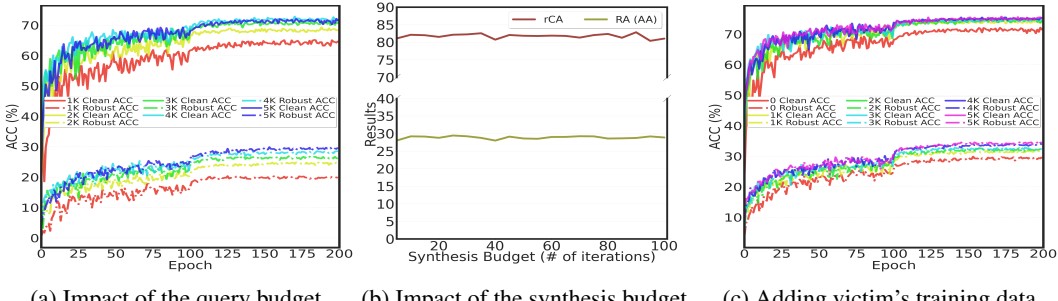

| (a) Impact of the query budget. | (b) Impact of the synthesis budget. | (c) Adding victim's training data. |

Figure 3: Exploration of the attack budget and training data. The dataset is CIFAR10. The victim model is ResNet-AT. The adversary model is ResNet.

| Data Augmentation | Extracted model with the highest robustness | | | | | | Final model after extraction | | | | | |
|---|---|---|---|---|---|---|---|---|---|---|---|---|
| | CA | rCA | RA | | | | CA | rCA | RA | | | |
| | | | PGD20 | PGD100 | CW100 | AA | | | PGD20 | PGD100 | CW100 | AA |
| ✓ | 71.24 | 82.12 | 29.88 | 29.68 | 30.56 | 29.2 | 71.5 | 82.34 | 29.76 | 29.54 | 30.52 | 29.12 |
| ✗ | 65.54 | 73.6 | 22.1 | 21.76 | 22.64 | 21.02 | 70.3 | 77.9 | 19.4 | 19.06 | 20.34 | 18.46 |

Table 4: Attack results with and without data augmentation. The victim model is ResNet-AT trained on CIFAR10. The adversary model is ResNet. The adversary's dataset is from CIFAR10.

## 5.3 MODEL EXTRACTION WITH DIFFERENT TYPES OF DATA

In the above experiments, the adversary adopts the samples from the same distribution of the victim model's test data to synthesize uncertain examples. In this section, we consider and evaluate some alternatives for query sample generation.

**Incorporating Training Samples**. In some cases, the adversary may have the victim's original training data, e.g., the victim's model is trained over a public dataset. Then the adversary can add the training samples into $D_{\mathcal{A}}$ for model extraction. This threat model has been considered in prior works (Tramèr et al., 2016; Jagielski et al., 2020b; Pal et al., 2020). In our experiments, we first set $D_{\mathcal{A}}$ with 5,000 samples of the test data's distribution, and then add different numbers of victim's training samples into $D_{\mathcal{A}}$. Figure 3c shows the extracted clean and robust accuracy with different configurations. We observe that the incorporation of training samples is very helpful for improving the attack performance since they are directly related to the victim model. Even with 1,000 training samples, the clean and robust accuracy is improved by 2.64% and 1.88%, respectively.

**Applying Data Augmentations**. Data augmentation has been a popular strategy to enhance the model's robustness. We can also leverage this technique to generate uncertain examples, which could possibly improve the attack performance. Table 4 compares the results with and without augmentation. Details about the adopted augmentation operations can be found in Appendix B.1. Clearly, when the adversary uses data augmentation to first augment the clean sample and then generate the query sample, the clean accuracy and robustness are significantly higher than the case without data augmentation. Besides, using data augmentation can also help the adversary bypass the victim's defense, which will be discussed in Section 5.4.

## 5.4 BYPASSING MODEL EXTRACTION DEFENSES

Past works proposed several defense solutions to alleviate model extraction threats, which can be classified into three categories. The first kind is to add perturbations into the logits vectors without changing the prediction labels (Lee et al., 2019). Since our BEST only needs the hard labels of the query samples to extract models, such defenses do not work. The second type is to detect malicious query samples. Upon the identification of a suspicious sample, the victim model will return an incorrect prediction. We consider two typical detection methods. (1) PRADA (Juuti et al., 2019) is a global detection approach. It detects malicious samples based on a priori hypothesis that the differences between normal samples in the same class obey a Gaussian distribution, while the differences between synthesized samples often follow long-tailed distributions. We reproduce this method and evaluate its effectiveness in detecting BEST. We observe that initially, PRADA needs to establish knowledge about the anomalous distributions of malicious samples. After 6,180 queries, it is able to identify each uncertain example. To bypass such detection, the adversary can apply data augmentation for generating uncertain examples (Section 5.1). The randomness in these augmentation

operations can disrupt the defender's knowledge about anomalous queries. Our experiments show that PRADA fails to detect any adversary's query sample generated with data augmentation. (2) SEAT (Zhang et al., 2021b) is an account-based detection method. It detects and bans suspicious accounts which send similar query samples multiple times. To bypass SEAT, the adversary only needs to register more accounts and use them to query the victim model, which can reduce the attack cost as well (Appendix C.2). The third kind of strategy is to increase the computational cost of model extraction. Dziedzic et al. (2022) introduced proof-of-work (POW) to increase the query time of malicious samples. This is a strong defense against existing extraction attacks with specially-crafted query samples. We observe the query cost of our attack grows exponentially with the number of queries. It is interesting to improve our attack to bypass this method. For instance, the adversary can try to behave like normal users when querying the model. He can set up large numbers of accounts and ensure the queries in each account will not exceed the privacy budget. Since there are no evaluations about the possibility of bypassing POW with multiple accounts in (Dziedzic et al., 2022), we will consider it as an interesting future work. We discuss more details in Appendix C.9.

## 5.5 More Evaluations

We perform more perspectives of evaluations, and the results are in the Appendix. In particular,

**Extracting Models with Out-Of-Distribution Data**. We further consider a weaker adversary, who can only obtain out-of-distribution data to query the victim model. The results and analysis can be found in Appendix C.4. BEST can still restore the victim model in some degree. In Appendix C.5, we prove the using additional out-of-distribution data can improve the extraction results.

**Extracting Non-robust Models**. Our method is general and can extract non-robust models as well. Given a model trained normally, BEST is able to precisely extract its clean accuracy. Extraction results can be found in Appendix C.6.

**Transferability Stabilization**. Our attack enjoys high transferability stabilization (Papernot et al., 2017), i.e., AEs generated from the victim model can achieve similar accuracy as the extracted models, and vice versa. We demonstrate this feature with different model architectures in Appendix C.10.

## 6 Discussions

The traditional model extraction problem was introduced many years ago and has been well-studied. In contrast, this is the first time to propose *robustness extraction*. As an initial attempt, our attack method also has some limitations. We expect this paper can attract more researchers to explore this problem and come up with better solutions. Below, we discuss some open questions for future work.

- Although our method outperforms existing SOTA solutions, there still exists a robustness gap between the extracted model and the victim model. One possible solution to reduce such a gap is to increase the number of query samples (Appendix C.5). In the future, it is important to improve the extracted robustness in a more efficient way.

- In this paper, we mainly consider adversarial training for building a robust model, which is the most popular strategy. There can be other robust solutions, e.g., certified defense (Cohen et al., 2019; Li et al., 2019), which will be considered in the future. Besides, we mainly focus on the image classification task. It is also interesting to extend this problem to other AI tasks and domains.

- Recent works proposed data-free attacks (Truong et al., 2021; Kariyappa et al., 2021), where the adversary trains a GAN to generate query samples from noise. We find these techniques cannot achieve promising results for extracting the model's robustness. How to design data-free techniques for robustness extraction is a challenging problem, and we leave it for future work.

- In Section 5.4 we show that our attack can invalidate existing defense solutions. It is important to design more effective approaches to protect a remote model from robustness extraction. Possible directions include detection of UEs and extraction-aware adversarial training algorithms.

## 7 Conclusion

In this paper, we present the first study on the robustness extraction of deep learning models. We design BEST, a new model extraction technique, which synthesizes uncertain examples to obtain the clean accuracy and robustness of the victim model simultaneously. Experimental results indicate that BEST outperforms prior attack methods, which are designed only for accuracy or fidelity extraction.

# 8 ACKNOWLEDGEMENT

This work is supported under the RIE2020 Industry Alignment Fund–Industry Collaboration Projects (IAF-ICP) Funding Initiative, as well as cash and in-kind contributions from the industry partner(s). It is also supported in part by Singapore Ministry of Education (MOE) AcRF Tier 2 MOE-T2EP20121-0006 and AcRF Tier 1 RS02/19. Furthermore, we appreciate the help from Dr. Adam Dziedzic and anonymous reviewers during the rebuttal periods.

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

# A    RELATED WORKS

## A.1    ACTIVE LEARNING IN MODEL EXTRACTION ATTACKS

Model extraction attacks can adopt active learning algorithms (Chandrasekaran et al., 2020). There are three basic types of sampling-based active learning algorithms. The first type is randomly sampling, which means for each query, the adversary randomly sends some samples to the victim, and uses the returned label to train his model. The second one is uncertainty strategy (Lewis & Gale, 1994). The adversary will choose the most uncertain samples to query the victim and use them to train his model. The third one is k-center strategy (Sener & Savarese, 2018; Pal et al., 2020). The adversary will generate cluster centers based on each sample's prediction, and then choose the most distant sample for each cluster to make the query sample. The model is trained based on these samples, and the adversary will update the cluster centers after every training step. The sampling-based active learning methods require millions or billions of data, and their threat model assumes that the adversary can have lots of data and use them to query the victim model. However, this threat model gives the adversary too much power, and the sampling process needs lots of local computation, which decreases the willingness to steal the victim model.

Besides these basic active learning methods, Papernot et al. (Papernot et al., 2017) proposed Jacobian-Based Dataset Augmentation (JBDA) can be seen as an active learning algorithm performing automated data augmentation, which aims to make the boundary of the adversary's model close to the boundary of the victim model. Orekondy et al. (Orekondy et al., 2019) introduced reinforcement learning into model extraction attacks to restructure the adversary's dataset, improving the attack efficiency. Yuan et al. (Yuan et al., 2020) proposed a GAN-based model extraction attack, and then data-free attacks (Truong et al., 2021; Kariyappa et al., 2021) and data-free adversarial training (Zhou et al., 2020) followed this work and improved it. The adversary trains a GAN during the model extraction process through the gradient estimation from the victim model's outputs. Then the adversary train his model with GAN-generated samples. The GAN-based active learning methods do not need the adversary to have any data. However, the adversary must query the victim model to train a GAN. For a low-resolution dataset, for example, CIFAR-10, training such a GAN is not difficult. But, when we do a model extraction attack on a high-resolution dataset, for example, ImageNet, the computational overhead of training a GAN appears.

In the experiments in Section 3, the results indicate that JBDA outperforms CloudLeak (Yu et al., 2020) in robustness extraction. This stems from the distinct goals between JBDA and CloudLeak, i.e., CloudLeak is designed to only restore clean accuracy of models, while JBDA aims to restore model's boundaries. Specifically, CloudLeak is a non-active learning method. The adversary only generates adversarial examples (AEs) once and uses them to query the victim model. Then the adversary adopts these AEs with the returned labels to train his model. For JBDA, it is an active learning method, which means that the adversary will generate data before each training step. So, the restored model by JBDA has more similar boundaries as the victim model's boundaries, making the restored model obtain partial robustness of the victim model. That is the reason that the robust performance of JBDA is better than CloudLeak.

Compared with our `BEST`, conventional model extraction attacks cannot generate classification boundary-sensitive queries. In our experiment, we show previous works cannot steal the victim model's robustness and analyze the gap between robust model extraction and naive model extraction.

## A.2    ROBUST KNOWLEDGE DISTILLATION

Knowledge distillation (Hinton et al., 2015) aims to use a small student model to learn a larger teacher model's knowledge. Robustness can be seen as specific knowledge from the teacher model. To distill robust knowledge, Goldblum et al. (Goldblum et al., 2020) propose a robust knowledge distillation method, named ARD. They aim to train a small student model from a large teacher model, and the student model can obtain better robustness than the teacher model. Zhu et al. (Zhu et al., 2022) propose the IAD to distill robust knowledge from a teacher model to a student model with the same model architecture. The IAD can automatically detect whether the teacher is good at predicting adversarial examples and adopt proper training loss to learn the teacher's knowledge. Zi et al. (Zi et al., 2021) propose RSLAD to further improve the student model's robustness. The core insight

behind these methods is using the student model to generate adversarial examples and using the prediction of clean images from the teacher model as the label of these adversarial examples.

Specifically, ARD adopts PGD attack (Madry et al., 2018) to generate adversarial examples for the student model, and uses Kullback–Leibler divergence to minimize the differences between the predictions of the student model and the predictions of the teacher model for the clean data and adversarial examples. IAD further proposes an adaptive distillation process to overcome the challenge that the teacher model will return unreliable answers, with longer training time. IAD adopts PGD attack to generate adversarial examples for the student model, and uses teacher guidance loss combined with a student introspection loss to train the student model. RSLAD replaces the PGD generated adversarial examples with TRADES (Zhang et al., 2019) generated adversarial examples in ARD to train the student model.

There are two main differences between knowledge distillation and model extraction. First, in knowledge distillation, the user has full knowledge of the teacher model's training set and details of the model. So the user can adopt the same training set to train a student model. Second, in knowledge distillation, the user can obtain the logits vectors from the teacher model, So the user can adopt a loss function such as Kullback–Leibler divergence or mean squared error loss to align the logits vectors of the student model and the teacher model. In contrast, in the model extraction scenario, the user normally does not have the same training set, and does not get the whole logits vectors for the query samples. Hence, the adversary has to use a different set of query samples and mainly adopts the cross-entropy loss to train the local model. So, blindly using robust knowledge distillation as a model extraction attack to steal the victim model's robustness can cause reliability problems. In our experiments, we perform comprehensive experiments to verify that under the model extraction threat model, robust knowledge distillation cannot guarantee advanced results.

## B  EXPERIMENT SETTINGS.

### B.1  DETAILS OF CONFIGURATIONS

In all experiments, the learning rate of model extraction is set as 0.1 at the beginning and decays at the 100th and 150th epoch with a factor of 0.1. The optimizer in all experiments is SGD, with a start learning rate of 0.1, momentum of 0.9 and weight decay of 0.0001. The total number of extraction epochs is 200. In each epoch, the adversary queries all data in his training set $D_{\mathcal{A}}$. The batch size is 128. The hyperparameter in JBDA for Jacobian matrix multiplication is $\beta = 0.1$. For ARD, IAD, RSLAD and our BEST, the hyperparameters for query sample generation under $L_\infty$-norm are $\epsilon = 8/255$, $\eta = 2/255$ and $B_S = 10$.

For all baseline methods and our BEST, the adversary adopts a pre-trained model to facilitate the model extraction process. Specifically, the pretrained model is used to initialize the adversary's local model, which means that the adversary uses a pretrained model as a start to restore the victim's model. All baseline methods and our attack follow the same attack pipeline, i.e., using the same pre-trained model as a start and restoring the victim model by training the pre-trained model with the query data. All the pre-trained models are downloaded from the open repository in GitHub. The pre-trained models are trained on Tiny-ImageNet. There are four network structures for the pre-trained models: ResNet, VGG, MobileNet and WideResNet.

We consider two settings for the victim's MLaaS: returning the logits vector or the hard label for each query. For the former setting, all baseline methods adopt Kullback-Leibler divergence as the loss function. For the later setting, we replace the Kullback-Leibler divergence with the Cross-Entropy loss, which is the same as previous works.

For both our method and baselines, we apply data augmentation when generating query samples. It includes central cropping, adding Gaussian noise, random image flipping, and random rotation. We show this augmentation can help increase the attack performance and bypass the defense in the experiments.

In terms of the query budgets, for all the methods in our experiments, we use all data in the extraction set in one training epoch. So, the number of training epochs is proportional to the query budget. For instance, 1 training epoch means the query budget of 5,000, and 10 training epoch means the query budget of 50,000. We use 200 training epochs for all methods as an upper bound and compare the best

results during the model extraction process and the last results each method can obtain. Furthermore, we discuss how to reduce our query budget in Appendix C.2. The query budget will not cause more financial cost in practice, as many MLaaS providers offer certain free queries for users.

## B.2 DETAILS OF QUERY SAMPLES

In our experiments, our attack augments the clean data during the model extraction attack. For our BEST, the query data are the clean data adding uncertain perturbation, converting into UEs. As shown in Algorithm 1 in Section 4.2, the perturbation is generated by solving a minimization problem. We restrict the perturbation size and the number of iterations in the perturbation generation process, which is consistent with other baselines. Similar to our method, the query data in JBDA, ARD, IAD and RSLAD are derived from the clean samples. For JBDA, the query data are the clean data after Jacobian augmentation. For ARD, IAD and RSLAD, the query data are the clean data adding adversarial perturbation.

## B.3 DETAILS OF VICTIM MODELS

In Table 1, we show the detailed information of all victim models used in our experiments, including their clean accuracy and robustness under various attacks.

| Dataset | Model | Clean | PGD20 | PGD100 | CW100 | AA |
|---|---|---|---|---|---|---|
| CIFAR10 | ResNet-AT | 82.92 | 51.62 | 51.42 | 50.14 | 47.90 |
| | ResNet-TRADES | 83.24 | 53.30 | 53.02 | 51.08 | 49.62 |
| | WRN-AT | 86.86 | 53.68 | 53.48 | 53.22 | 50.92 |
| | WRN-TRADES | 84.78 | 55.68 | 55.34 | 53.46 | 52.16 |
| CIFAR100 | ResNet-AT | 56.58 | 29.24 | 28.94 | 27.30 | 24.82 |
| | ResNet-TRADES | 57.44 | 29.70 | 29.58 | 26.10 | 25.10 |
| | WRN-AT | 62.54 | 31.88 | 31.70 | 30.42 | 27.96 |
| | WRN-TRADES | 62.02 | 31.76 | 31.56 | 28.58 | 27.12 |

Table 1: The detailed information of victim models.

## B.4 DETAILS OF METRICS

The formula of *rCA* is:

$$rCA(M_{\mathcal{A}}, M_{\mathcal{V}}, D_{\mathcal{T}}) = \frac{1}{N} \sum_{i=1}^{N} \mathbb{1}(\max(M_{\mathcal{A}}(x_i)) = \max(M_{\mathcal{V}}(x_i))), x_i \in D_{\mathcal{T}} \tag{3}$$

where $M_{\mathcal{A}}$ is the adversary's model, $M_{\mathcal{V}}$ is the victim's model, and $D_{\mathcal{T}}$ is the validation set.

The formula of *RA* is:

$$RA(M_{\mathcal{A}}, D_{\mathcal{T}}) = \frac{1}{N} \sum_{i=1}^{N} \Pr[M_{\mathcal{A}}(x_i + \epsilon_i) = M_{\mathcal{A}}(x_i) | \epsilon_i \in \mathcal{B}_{(\mathbf{0}, \epsilon)}^{p}], x_i \in D_{\mathcal{T}} \tag{4}$$

where $p$ is the norm and $\epsilon$ is the maximum perturbation margin, which together constrain the perturbation $\epsilon_i$ in a hypersphere $\mathcal{B}_{(\mathbf{0}, \epsilon)}^{p}$, whose center is the origin.

## C MORE RESULTS

### C.1 EXTRACTION RESULTS UNDER $L_2$-NORM ATTACKS

In Table 2, we evaluate the results under various $L_2$-norm attacks. The $\epsilon = 0.5$ and $\eta = 0.1$. The victim model is ResNet-18, which is trained by PGD-AT. The adversary's model is ResNet18. The dataset is CIFAR10. The results indicate that our method can achieve better robustness and avoid robust overfitting during the model extraction, which are the same conclusions in our main paper.

| Method | | Extracted model with the highest robustness | | | | | Final model after extraction | | | | |
|---|---|---|---|---|---|---|---|---|---|---|---|
| | | $CA$ | $rCA$ | $RA$ | | | $CA$ | $rCA$ | $RA$ | | |
| | | | | PGD20 | PGD100 | CW | | | PGD20 | PGD100 | CW |
| Label | Vanilla | 58.74 | 64.02 | 23.74 | 23.36 | 47.54 | 74.12 | 81.58 | 28.74 | 28.00 | 62.06 |
| | JBDA | 64.20 | 71.80 | 39.20 | 39.08 | 57.52 | 63.96 | 71.36 | 36.06 | 35.84 | 56.72 |
| | ARD | 57.82 | 65.86 | 42.56 | 42.50 | 53.92 | 51.86 | 58.18 | 32.12 | 32.08 | 47.22 |
| | IAD | 55.10 | 62.92 | 41.98 | 41.92 | 51.56 | 52.86 | 60.00 | 33.92 | 33.80 | 47.70 |
| | RSLAD | 54.72 | 63.46 | 40.80 | 40.72 | 51.02 | 53.44 | 60.48 | 35.06 | 35.00 | 48.76 |
| Logits | Vanilla | 76.14 | 82.78 | 25.10 | 23.60 | 61.22 | 75.84 | 82.22 | 24.38 | 22.98 | 61.48 |
| | JBDA | 69.74 | 76.94 | 38.54 | 38.22 | 60.36 | 67.78 | 74.18 | 35.44 | 35.02 | 57.34 |
| | ARD | 67.36 | 76.32 | 49.38 | 49.44 | 62.72 | 64.10 | 73.06 | 45.94 | 45.88 | 58.98 |
| | IAD | 63.90 | 71.84 | 47.72 | 47.66 | 58.72 | 58.18 | 65.76 | 40.48 | 40.42 | 52.84 |
| | RSLAD | 66.16 | 74.92 | 49.42 | 49.36 | 60.98 | 60.40 | 67.94 | 41.46 | 41.44 | 54.46 |
| BEST | | 71.24 | 82.12 | 49.68 | 49.52 | 65.98 | 71.50 | 82.34 | 49.48 | 49.38 | 66.06 |

Table 2: Model extraction attack results on CIFAR10.

| Query Budget | Extracted model with the highest robustness | | | | | |
|---|---|---|---|---|---|---|
| | $CA$ | $rCA$ | $RA$ | | | |
| | | | PGD20 | PGD100 | CW100 | AA |
| 5k * 100 | 71.24 | 82.12 | 29.88 | 29.68 | 30.56 | 29.20 |
| 5k * 80 | 68.50 | 78.78 | 27.02 | 26.90 | 27.90 | 26.40 |

Table 3: Model extraction attack results under different Query Budgets.

## C.2 ANALYSIS OF ATTACK BUDGETS

**Query Budget Analysis**. Our method requires additional queries to obtain the boundary information of the victim model, which is an indispensable procedure in robustness improvement. It is inevitable because training a robust model requires much more computational cost (Zhao et al., 2020). We can adopt early learning decay to decrease the query budget. In our experiments, all the highest robustness models are with query budgets of about 1K – 5K * 100. With the early learning decay method, we restore the victim model with 5K * 80 query budgets. The results in Table 3 indicate that reducing the query budgets will not significantly decrease the restored model's clean accuracy and robustness.

On the other hand, we find that using more accounts can reduce query costs. For example, AWS provides a Free Tier for new accounts to analyze 5,000 images per month for free[3]. Google provides all accounts with a discount of predicting 1,000 images per month free[4]. Microsoft provides all accounts with a discount of predicting 5,000 images per month for free[5]. It is feasible to use more accounts to steal the victim model, which can significantly reduce the financial cost as creating new accounts is easy and trivial. Hence, the query budget is not the principal limitation in model extraction attacks.

**Synthesis Budget Analysis**. Our method is computational efficiency because the scale of UEs that need to be generated in one training epoch for the adversary is small. To better quantify the $B_S$ in the model extraction attack, we measure the time cost for the UE generation process. In our experiments, the adversary only needs to generate 5,000 UEs in one epoch. Specifically, when $B_S = 10$, for ResNet-18, it costs about 16s on V100 to generate 5,000 UEs. For WRN-28-10, it costs about 80s on V100 to generate 5,000 UEs.

**Attack Cost**. We compare the total time cost of stealing a restored model and training a robust model from scratch. For a training set containing 5,000 data, we set the batch size as 128 and use SGD as our optimizer. When training a ResNet-18 on a single V100 card, the time cost for one epoch is 17s. When training a WideResNet-28-10 on a single V100 card, the time cost for one epoch is 80s. If the adversary adopts ResNet-18 as his model structure to extract a victim model (ResNet-18 or WideResNet-28-10), the time cost for 5,000 query budgets (including querying victim model and training local model) is 19s (without considering network latency). So, model extraction will cost

---

[3] https://aws.amazon.com/rekognition/pricing/?nc1=h_ls
[4] https://cloud.google.com/vision/product-search/pricing
[5] https://azure.microsoft.com/en-us/pricing/details/cognitive-services/computer-vision/

less time when stealing a bigger model. The reason is that in our method, we adopt a similar training pipeline as in the adversarial training. First, the UE generation process is similar to the AE generation process. We only modify the loss function in the original AE generation process. So, the time cost for AE generation and UE generation is the same. Second, the model extraction requires the adversary to query the victim model, which will not cost too much time if we do not consider the network latency. Third, the model training process is the same as the adversarial training. Overall, model extraction attacks are more efficient when restoring a huge deep learning model.

## C.3 IMPACT OF PRE-TRAINED MODELS

We evaluate the improvement from a pre-trained model in Table 4. The results indicate that under limited data, the pre-trained model can significantly improve clean accuracy and robustness. In fact, using pre-trained models does not affect the superiority of our method, due to the following reasons. First, all the baseline methods adopt the same pre-trained model to initialize the adversary's model. This gives us very fair comparisons. Second, the adopted pre-trained models are normal without any robustness features. This explains why other baseline methods using these models cannot extract the robustness of the victim model (Section 5.2 and Appendix C.8). Third, Table 4 proves that even without pre-trained models, our method can still restore robustness from the victim models. The main reason is that using UEs to query the victim model can obtain the most informative outputs from the victim model and using such samples to train the adversary's model can better shape the classification boundaries to fit the victim model's boundaries, which makes the adversary's model achieve higher robustness.

On the other hand, using pre-trained models in model extraction attacks is based on a very practical fact that they are widely existed in various tasks, beyond image domain and computer vision tasks, and can be easily downloaded. For example, the well-known website ModelZoo mod (2022) provides various pre-trained models for different tasks, including natural language processing, text-to-speech, audio generation, and image-to-text. It covers different intelligent tasks, like NLP, Audio, and Multimodality.

## C.4 ADOPTING DIFFERENT DISTRIBUTIONS OF SAMPLES

We consider another scenario where the adversary does not know the distribution of the victim model's test data. He may use samples from a different distribution to synthesize the uncertain examples for extraction. Table 5 shows the evaluation result of such a case, where the victim model is trained over CIFAR10, while the adversary uses data from CIFAR10 (in-distribution) as well as SVHN, CIFAR100, STL10 (out-of-distribution) to perform attacks. To be specific, the data distribution of STL10 is the closest one to the CIFAR10, while the data distribution of SVHN is the furthest one to the CIFAR10. We observe that model extraction with a different distribution of samples has much lower clean and robust accuracy. Combining the results in the above paragraph, we conclude that a reduced gap between the distributions of the victim's training data and the adversary's extraction data can increase the clean accuracy and robust accuracy. We provide more discussions about how to enhance the attack with out-of-the-distribution data in Appendix C.5.

## C.5 ADOPTING MORE SAMPLES

We explore how to further improve the results under our threat model. In experiments in our main paper, there are only 5,000 data that the adversary can use to query the victim model, so the gap between the victim model and the restored model can be decreased by adding more data. So, we compare the results under 5,000 CIFAR10 data and 5,000 CIFAR100 or 5,000 STL10 data. The results in Table 6 indicate that increasing the number of query data is an efficient way to improve our results, even though the data distribution is different. It is to say that our method can suit a mixture of distributions, which is meaningful if there is not enough data from a single distribution for the adversary to collect.

## C.6 EXTRACTING NON-ROBUST VICTIM MODEL

In addition to robust models, our approach can also extract non-robust models, just for clean accuracy. Figure 4 shows the attack results, where the victim model is trained with the normal method (ResNet

| Pre-trained Model | Extracted model with the highest robustness | | | | | |
|---|---|---|---|---|---|---|
| | $CA$ | $rCA$ | RA | | | |
| | | | PGD20 | PGD100 | CW100 | AA |
| w/ | 71.24 | 82.12 | 29.88 | 29.68 | 30.56 | 29.20 |
| w/o | 60.84 | 70.30 | 21.56 | 21.28 | 22.20 | 20.72 |

Table 4: Model extraction attack results when the pre-trained model is adopted or not.

| $D_A$ | Extracted model with the highest robustness | | | | | | Final model after extraction | | | | | |
|---|---|---|---|---|---|---|---|---|---|---|---|---|
| | $CA$ | $rCA$ | RA | | | | $CA$ | $rCA$ | RA | | | |
| | | | PGD20 | PGD100 | CW100 | AA | | | PGD20 | PGD100 | CW100 | AA |
| CIFAR10 | 71.24 | 82.12 | 29.88 | 29.68 | 30.56 | 29.20 | 71.50 | 82.34 | 29.76 | 29.54 | 30.52 | 29.12 |
| SVHN | 40.36 | 47.58 | 15.50 | 15.38 | 15.78 | 15.10 | 42.02 | 48.78 | 15.00 | 14.80 | 15.26 | 14.66 |
| CIFAR100 | 59.82 | 69.44 | 20.42 | 20.30 | 20.94 | 19.96 | 59.16 | 69.98 | 20.24 | 20.10 | 21.10 | 19.76 |
| STL10 | 63.48 | 73.00 | 22.62 | 22.50 | 23.60 | 22.26 | 63.90 | 73.20 | 21.86 | 21.64 | 22.64 | 21.30 |

Table 5: Results of different query distributions. The victim model is ResNet-AT trained on CIFAR10. The adversary model is ResNet.

architecture and CIFAR10 dataset). The adversary uses ResNet for model extraction. Black solid and dashed lines denote the clean and robust accuracy of the victim model. We observe that the extracted model can inherit clean accuracy as well as non-robustness (against PGD20) from the victim model. Therefore, we can draw two conclusions: (1) our BEST is general for both robust and non-robust models. (2) For robustness extraction, the high robustness of the extracted model is indeed learned from the victim, rather than from the synthesized uncertain examples.

## C.7 COMPARISONS WITH EXTRACTION-AT

We compare the results between BEST and Extraction-AT (i.e., restoring the victim model first and then performing adversarial training). For extraction-AT, we adopt various model extraction techniques to restore the victim model, and then use PGD-based adversarial training (AT) to train the restored model. In Table 7 reports the results of the model with the highest robust accuracy and results of the final model. The victim model is ResNet18 and the adversary's model is ResNet18. The results prove that our method is better than Extraction-AT methods. Models restored with BEST can achieve higher clean accuracy and robustness.

## C.8 RESULTS OF VARIOUS VICTIM MODELS

In Tables 8 to 22, we display the results of different attack scenarios on CIFAR10. In Tables 23 to 38, we display the results of different attack scenarios on CIFAR100. The victim models include ResNet-AT, ResNet-TRADES, WRN-AT, and WRN-TRADES. The adversary models include ResNet, WRN, VGG, and MobileNet. Clearly, our BEST outperforms other baselines under various settings. And we have the same conclusions as in the main paper. Especially, when the adversary adopts VGG as his model to steal a victim model with logits, other baselines cannot make the model converge on CIFAR100. This is because using logits as labels can introduce more noise during the training process, and training VGG is more difficult when compared with training other models. Our BEST can keep stable when the adversary uses VGG as his model, as our method only requires the hard labels.

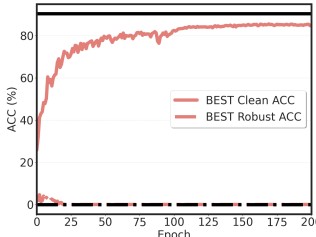

Figure 4: Clean and robust accuracy of extracting a non-robust model.

Based on the results of JBDA, ARD, IAD, RSLAD and BEST, we can find that the robust feature and overfitting can have a close connection. First, the technique used to augment the clean samples in JBDA is very close to the FGSM attack, which will introduce robust features into the generated samples. Straightforwardly, when comparing our method with ARD, IAD and RSLAD (these three attacks generated adversarial examples first, which contain robust features), we also find that the robust features in query samples of ARD, IAD and RSLAD cause overfitting. So, the Property **P1** can be proven from the experiments results. Our UEs do not contain robust features, and obtain the robustness of the victim model by shaping the restored classification boundaries as the victim model's boundaries.

| $D_{\mathcal{A}}$ | Extracted model with the highest robustness | | | | | | Final model after extraction | | | | | |
|---|---|---|---|---|---|---|---|---|---|---|---|---|
| | $CA$ | $rCA$ | $RA$ | | | | $CA$ | $rCA$ | $RA$ | | | |
| | | | PGD20 | PGD100 | CW100 | AA | | | PGD20 | PGD100 | CW100 | AA |
| CIFAR10 | 71.24 | 82.12 | 29.88 | 29.68 | 30.56 | 29.20 | 71.50 | 82.34 | 29.76 | 29.54 | 30.52 | 29.12 |
| CIFAR10+CIFAR100 | 73.30 | 85.48 | 33.38 | 33.20 | 33.88 | 32.72 | 73.70 | 85.60 | 32.34 | 32.28 | 33.00 | 31.76 |
| CIFAR10+STL10 | 74.46 | 86.14 | 34.26 | 34.04 | 34.74 | 33.50 | 74.66 | 86.66 | 33.42 | 33.24 | 34.20 | 32.66 |

Table 6: Model extraction attack results under different extraction datasets.

| Dataset | Method | Extracted model with the highest robustness | | Final model after extraction | |
|---|---|---|---|---|---|
| | | $CA$ | $RA$ | $CA$ | $RA$ |
| | | | PGD20 | | PGD20 |
| CIFAR10 | `Vanilla + AT` | 63.54 | 30.38 | 68.58 | 23.41 |
| | `JBDA + AT` | 61.96 | 32.92 | 68.00 | 24.62 |
| | `ARD + AT` | 60.46 | 29.74 | 67.16 | 25.58 |
| | `IAD + AT` | 61.44 | 29.12 | 65.74 | 25.50 |
| | `RSLAD + AT` | 60.34 | 30.20 | 65.70 | 23.62 |
| | `BEST` | 71.24 | 29.88 | 71.50 | 29.76 |
| CIFAR100 | `Vanilla + AT` | 34.86 | 13.80 | 39.38 | 11.70 |
| | `JBDA + AT` | 36.52 | 15.50 | 38.92 | 12.32 |
| | `ARD + AT` | 32.52 | 13.62 | 36.68 | 11.14 |
| | `IAD + AT` | 32.80 | 13.68 | 37.04 | 12.86 |
| | `RSLAD + AT` | 36.72 | 13.32 | 37.20 | 12.66 |
| | `BEST` | 44.38 | 14.26 | 44.78 | 13.72 |

Table 7: Comparisons between `BEST` and Extraction-AT. The victim model is ResNet-AT. The adversary's model is ResNet.

So, can we further improve the results of JBDA to beat `BEST`? One way to enhance the power of JBDA is to use FGSM on the augmented data. However, based on our analysis, we find it is impossible. First, the samples generated by JBDA with FGSM are still adversarial examples, which will contain robust features. And based on our experimental analysis, robust features have a close connection with robust overfitting. Second, JBDA with FGSM can only generate weak AEs. When considering ARD, IAD and RSLAD, which can generate AEs with PGD attacks, they adopt stronger AEs to extract the victim model's robustness. Comparing the results between ARD, IAD, RSLAD and our method, we find that even using stronger AEs, the ARD, IAD and RSLAD cannot beat our method. So, the JBDA with FGSM will not outperform our method.

To summarize, we conclude that `BEST` has three main advantages compared to other baselines. First, `BEST` can restore high clean accuracy and relative clean accuracy, which is impossible for robust knowledge distillation methods (e.g., ARD, IAD and RSLAD). The reason is that `BEST` adopts UEs to reshape the local model's boundaries to be similar to the victim's boundaries, obtaining higher clean accuracy. Second, `BEST` can obtain high robustness under limited clean data when restoring a robust victim model. Because UEs can help the local model obtain a similar classification boundary as the victim model's boundary, models restored with `BEST` can exhibit similar behaviors on clean data and adversarial examples, which is challenging for other baseline methods. Third, `BEST` can relieve the annoying robust overfitting problem. The robust overfitting is very common and severe in ARD, IAD and RSLAD. However, our method does not rely on the adversarial examples. The results indicate that our proposed UEs can successfully address the robust overfitting challenges. Overall, `BEST` is better than previous baselines and achieves higher clean accuracy and robust accuracy under limited clean data.

## C.9 ATTACK AGAINST POW

There are two ways to implement the POW defense (Dziedzic et al., 2022) in MLaaS. The first way is to count the per-query cost for each user. In this way, the adversary cannot adopt multiple accounts to decrease the total time cost. Furthermore, the time cost will grow at a linear speed, if the cost for each query is almost the same. For our model extraction attack method, because the adversary needs lots of queries to restore the robustness of the victim model, the total time cost is not negligible. The second way is to count the cumulative cost of queries for each user. Based on this implementation, the time cost for a query will increase exponentially. For a normal user, it can introduce additional waiting time (Dziedzic et al., 2022). For an adversary, due to the privacy leakage caused by the query samples, the time cost will be thousands of times larger than that of the normal query. For our `BEST`,

| Method | | Extracted model with the highest robustness | | | | | | Final model after extraction | | | | | |
|---|---|---|---|---|---|---|---|---|---|---|---|---|---|
| | | CA | rCA | RA | | | | CA | rCA | RA | | | |
| | | | | PGD20 | PGD100 | CW100 | AA | | | PGD20 | PGD100 | CW100 | AA |
| Label | Vanilla | 61.44 | 64.96 | 3.74 | 3.10 | 3.86 | 2.34 | 76.14 | 80.82 | 1.58 | 1.36 | 1.58 | 0.84 |
| | JBDA | 65.38 | 70.34 | 16.00 | 15.80 | 16.78 | 15.18 | 64.06 | 68.74 | 10.56 | 10.24 | 11.18 | 9.28 |
| | ARD | 55.60 | 61.28 | 29.06 | 28.78 | 26.96 | 25.44 | 50.20 | 53.96 | 16.58 | 16.34 | 16.70 | 15.50 |
| | IAD | 56.40 | 61.32 | 30.08 | 30.02 | 28.02 | 26.50 | 57.22 | 62.24 | 19.54 | 19.20 | 19.52 | 18.44 |
| | RSLAD | 58.36 | 63.50 | 29.52 | 29.32 | 27.82 | 26.14 | 57.44 | 62.48 | 18.86 | 18.48 | 18.90 | 17.70 |
| Logits | Vanilla | 78.44 | 82.14 | 0.86 | 0.52 | 0.48 | 0.12 | 78.30 | 82.18 | 0.76 | 0.36 | 0.40 | 0.06 |
| | JBDA | 67.10 | 71.64 | 9.96 | 9.14 | 10.14 | 7.60 | 68.88 | 73.00 | 7.00 | 6.32 | 7.40 | 5.16 |
| | ARD | 65.88 | 71.32 | 27.70 | 27.10 | 25.64 | 23.80 | 61.18 | 66.36 | 22.58 | 22.42 | 20.58 | 19.32 |
| | IAD | 62.60 | 67.84 | 27.42 | 27.14 | 25.30 | 23.26 | 57.76 | 62.96 | 18.82 | 18.50 | 16.78 | 15.38 |
| | RSLAD | 65.76 | 71.26 | 25.52 | 25.14 | 23.62 | 21.88 | 58.86 | 63.64 | 18.48 | 18.18 | 16.10 | 14.74 |
| BEST | | 72.06 | 78.84 | 26.58 | 26.24 | 27.22 | 25.72 | 72.56 | 79.40 | 25.86 | 25.64 | 26.44 | 25.20 |

Table 8: Results of model extraction attacks on CIFAR10. The victim model is WRN-AT. The adversary model is ResNet.

| Method | | Extracted model with the highest robustness | | | | | | Final model after extraction | | | | | |
|---|---|---|---|---|---|---|---|---|---|---|---|---|---|
| | | CA | rCA | RA | | | | CA | rCA | RA | | | |
| | | | | PGD20 | PGD100 | CW100 | AA | | | PGD20 | PGD100 | CW100 | AA |
| Label | Vanilla | 66.68 | 73.18 | 5.36 | 4.82 | 5.44 | 4.20 | 73.20 | 79.90 | 3.04 | 2.62 | 3.10 | 1.90 |
| | JBDA | 61.66 | 68.58 | 17.88 | 17.70 | 18.52 | 17.20 | 59.72 | 66.94 | 14.92 | 14.60 | 15.64 | 13.98 |
| | ARD | 53.56 | 61.74 | 32.20 | 32.16 | 31.38 | 30.20 | 57.14 | 65.86 | 27.68 | 27.52 | 27.42 | 26.56 |
| | IAD | 54.42 | 62.28 | 31.18 | 31.12 | 30.14 | 28.88 | 54.72 | 62.44 | 22.06 | 21.86 | 21.86 | 20.92 |
| | RSLAD | 56.38 | 64.74 | 31.66 | 31.62 | 30.82 | 29.76 | 58.90 | 67.28 | 25.54 | 25.30 | 25.84 | 24.72 |
| Logits | Vanilla | 72.26 | 79.64 | 2.72 | 1.96 | 0.82 | 0.38 | 72.66 | 79.72 | 2.42 | 1.74 | 0.74 | 0.34 |
| | JBDA | 61.76 | 69.02 | 10.26 | 9.44 | 8.62 | 6.70 | 63.50 | 70.40 | 7.82 | 7.14 | 7.26 | 5.46 |
| | ARD | 59.52 | 68.50 | 31.50 | 31.56 | 28.56 | 27.44 | 59.54 | 68.64 | 28.96 | 28.92 | 26.26 | 25.12 |
| | IAD | 61.18 | 70.14 | 30.04 | 29.88 | 27.46 | 25.86 | 57.46 | 66.14 | 25.14 | 24.84 | 22.24 | 20.56 |
| | RSLAD | 61.50 | 70.84 | 30.58 | 30.44 | 27.04 | 25.66 | 59.44 | 68.44 | 26.04 | 25.90 | 22.64 | 21.52 |
| BEST | | 68.60 | 78.78 | 28.82 | 28.62 | 29.30 | 28.36 | 70.06 | 79.58 | 28.10 | 27.90 | 28.66 | 27.52 |

Table 9: Results of model extraction attacks on CIFAR10. The victim model is ResNet-TRADES. The adversary model is ResNet.

because the uncertain samples will obtain the boundary information from the victim model, which is a type of privacy leakage, the total query time will be too long and unacceptable.

Overall, the POW attack is robust to defend against our model extraction attack, because of its diverse implementation methods. It will be our future work to explore how to overcome such a defense with a robust model extraction attack. For instance, the adversary can try to behave like a normal user when querying the model. Since normal users may also possibly send normal images which the victim model has low confidence or uncertainty about their classes (e.g., images in the wild following different distributions from the training set), to reduce such false positives and make the service practical, the model owner should allow certain privacy budget for each account. Then the adversary can set up a large number of accounts and ensure the queries in each account will not exceed such privacy budget. Although the POW paper discussed that the adoption of multiple accounts can be defeated by summing over all the users, we believe the adversary can still succeed if he tries to mimic normal users for each account. The more accounts he has, the higher feasible it is to mimic normal users within the privacy budget.

## C.10 TRANSFERABILITY STABILIZATION

Transferability stabilization is defined as the AEs generated from the victim model that can achieve similar accuracy over the extracted models. Simultaneously, it requires the extracted models with different structures can generate AEs having similar transferability among each extracted model.

Our BEST can help $M_\mathcal{A}$ have various architectures obtain similar classification boundaries as the victim model's ones to achieve transferability stabilization. To verify this point, we plot the adversarial examples' transferability in Figure 5. The results show that the accuracy of the adversary model under the victim model's adversarial examples is close to the accuracy of the victim model's accuracy. Furthermore, the adversary models having different structures can obtain similar accuracy under attacks from other structure adversarial model's adversarial examples. These two points indicate that our BEST can make $M_\mathcal{A}$ achieve transferability stabilization.

| | Method | Extracted model with the highest robustness | | | | | | Final model after extraction | | | | | |
|---|---|---|---|---|---|---|---|---|---|---|---|---|---|
| | | CA | rCA | RA | | | AA | CA | rCA | RA | | | AA |
| | | | | PGD20 | PGD100 | CW100 | | | | PGD20 | PGD100 | CW100 | |
| Label | Vanilla | 61.64 | 66.96 | 5.00 | 4.58 | 5.02 | 3.74 | 72.30 | 78.60 | 2.76 | 2.28 | 2.74 | 1.62 |
| | JBDA | 62.20 | 68.40 | 17.72 | 17.48 | 18.26 | 16.86 | 58.84 | 63.74 | 11.10 | 10.82 | 11.60 | 10.28 |
| | ARD | 47.30 | 54.74 | 28.04 | 28.00 | 26.56 | 25.80 | 47.60 | 54.76 | 23.86 | 23.80 | 22.94 | 22.26 |
| | IAD | 47.48 | 54.80 | 28.10 | 28.10 | 26.90 | 26.18 | 47.90 | 54.94 | 23.48 | 23.50 | 22.80 | 22.18 |
| | RSLAD | 55.70 | 63.46 | 30.84 | 30.72 | 30.42 | 29.12 | 56.26 | 63.64 | 24.66 | 24.68 | 24.98 | 24.28 |
| Logits | Vanilla | 73.14 | 79.98 | 2.04 | 1.48 | 0.76 | 0.34 | 73.22 | 79.76 | 1.76 | 1.34 | 0.54 | 0.26 |
| | JBDA | 62.42 | 68.36 | 10.22 | 9.32 | 9.46 | 7.58 | 63.82 | 69.92 | 6.78 | 6.00 | 5.92 | 4.46 |
| | ARD | 60.10 | 68.80 | 30.86 | 30.76 | 27.68 | 26.60 | 58.34 | 66.56 | 28.02 | 27.98 | 24.64 | 23.50 |
| | IAD | 59.96 | 67.86 | 29.36 | 29.14 | 26.80 | 25.36 | 54.98 | 62.40 | 23.08 | 22.82 | 20.10 | 18.88 |
| | RSLAD | 60.44 | 69.16 | 29.56 | 29.50 | 26.24 | 25.28 | 58.20 | 66.18 | 25.26 | 25.12 | 21.54 | 20.54 |
| BEST | | 70.20 | 78.34 | 28.44 | 28.28 | 29.04 | 27.82 | 69.70 | 78.26 | 27.90 | 27.74 | 28.56 | 27.26 |

Table 10: Results of model extraction attacks on CIFAR10. The victim model is WRN-TRADES. The adversary model is ResNet.

| | Method | Extracted model with the highest robustness | | | | | | Final model after extraction | | | | | |
|---|---|---|---|---|---|---|---|---|---|---|---|---|---|
| | | CA | rCA | RA | | | AA | CA | rCA | RA | | | AA |
| | | | | PGD20 | PGD100 | CW100 | | | | PGD20 | PGD100 | CW100 | |
| Label | Vanilla | 71.78 | 78.16 | 5.02 | 4.64 | 5.24 | 3.44 | 76.72 | 83.72 | 2.54 | 2.22 | 2.32 | 1.36 |
| | JBDA | 66.46 | 74.16 | 18.94 | 18.68 | 19.72 | 17.98 | 65.42 | 72.16 | 14.62 | 14.38 | 15.40 | 13.76 |
| | ARD | 56.24 | 63.80 | 33.90 | 33.82 | 32.96 | 31.78 | 59.76 | 67.90 | 26.16 | 25.94 | 26.60 | 25.18 |
| | IAD | 56.22 | 63.78 | 32.54 | 32.40 | 32.40 | 31.14 | 51.26 | 57.28 | 17.84 | 17.56 | 17.80 | 16.74 |
| | RSLAD | 55.40 | 62.60 | 31.32 | 31.18 | 30.20 | 29.46 | 57.72 | 64.32 | 17.46 | 17.00 | 17.88 | 16.22 |
| Logits | Vanilla | 79.34 | 84.36 | 2.56 | 1.78 | 1.24 | 0.48 | 78.78 | 84.72 | 2.34 | 1.74 | 1.42 | 0.48 |
| | JBDA | 54.58 | 58.88 | 6.38 | 5.76 | 5.96 | 4.48 | 62.10 | 67.96 | 6.12 | 5.28 | 5.98 | 3.84 |
| | ARD | 70.78 | 79.36 | 29.40 | 28.82 | 28.20 | 26.10 | 65.76 | 74.10 | 25.50 | 25.12 | 23.86 | 21.84 |
| | IAD | 66.92 | 74.80 | 26.92 | 26.88 | 24.72 | 23.04 | 62.46 | 70.20 | 20.02 | 19.58 | 18.48 | 16.02 |
| | RSLAD | 70.12 | 79.04 | 27.32 | 26.92 | 25.94 | 23.72 | 64.14 | 72.78 | 21.12 | 20.92 | 19.28 | 17.24 |
| BEST | | 74.30 | 84.64 | 31.90 | 31.70 | 32.52 | 31.22 | 73.40 | 83.86 | 31.24 | 31.08 | 31.84 | 30.72 |

Table 11: Results of model extraction attacks on CIFAR10. The victim model is ResNet-AT. The adversary model is WRN.

| | Method | Extracted model with the highest robustness | | | | | | Final model after extraction | | | | | |
|---|---|---|---|---|---|---|---|---|---|---|---|---|---|
| | | CA | rCA | RA | | | AA | CA | rCA | RA | | | AA |
| | | | | PGD20 | PGD100 | CW100 | | | | PGD20 | PGD100 | CW100 | |
| Label | Vanilla | 65.44 | 68.68 | 4.06 | 3.72 | 4.10 | 3.00 | 78.76 | 82.30 | 1.36 | 1.12 | 0.92 | 0.50 |
| | JBDA | 64.38 | 68.20 | 15.50 | 15.22 | 16.52 | 14.62 | 67.24 | 72.14 | 11.40 | 11.20 | 12.16 | 10.66 |
| | ARD | 62.12 | 67.72 | 32.54 | 32.40 | 31.80 | 30.36 | 60.00 | 64.26 | 14.42 | 14.14 | 15.18 | 13.46 |
| | IAD | 59.74 | 65.86 | 32.96 | 32.84 | 31.38 | 29.74 | 58.26 | 62.98 | 19.30 | 18.84 | 19.94 | 18.28 |
| | RSLAD | 61.50 | 65.96 | 29.96 | 29.78 | 28.66 | 27.30 | 56.00 | 60.02 | 13.24 | 13.04 | 13.60 | 12.56 |
| Logits | Vanilla | 81.02 | 83.74 | 1.40 | 0.94 | 0.78 | 0.16 | 80.60 | 83.46 | 1.20 | 0.76 | 0.64 | 0.16 |
| | JBDA | 69.18 | 72.52 | 7.14 | 6.22 | 6.62 | 4.30 | 66.26 | 69.14 | 5.80 | 5.26 | 5.50 | 3.84 |
| | ARD | 70.00 | 75.26 | 27.86 | 27.34 | 26.92 | 24.92 | 66.50 | 71.62 | 22.26 | 21.76 | 21.52 | 19.14 |
| | IAD | 67.10 | 72.46 | 25.60 | 25.18 | 24.30 | 21.98 | 62.64 | 67.32 | 17.52 | 16.76 | 16.30 | 14.30 |
| | RSLAD | 69.36 | 73.88 | 25.52 | 25.12 | 24.04 | 21.82 | 63.38 | 68.28 | 16.94 | 16.46 | 15.74 | 13.54 |
| BEST | | 75.76 | 82.38 | 28.68 | 28.50 | 29.66 | 28.08 | 73.82 | 80.40 | 27.56 | 27.28 | 28.46 | 27.02 |

Table 12: Results of model extraction attacks on CIFAR10. The victim model is WRN-AT. The adversary model is WRN.

| | Method | Extracted model with the highest robustness | | | | | | Final model after extraction | | | | | |
|---|---|---|---|---|---|---|---|---|---|---|---|---|---|
| | | CA | rCA | RA | | | AA | CA | rCA | RA | | | AA |
| | | | | PGD20 | PGD100 | CW100 | | | | PGD20 | PGD100 | CW100 | |
| Label | Vanilla | 66.08 | 71.42 | 6.26 | 5.86 | 5.94 | 4.32 | 75.46 | 81.80 | 2.60 | 2.26 | 1.98 | 1.14 |
| | JBDA | 64.80 | 72.40 | 19.76 | 19.62 | 20.80 | 19.22 | 63.64 | 69.60 | 16.12 | 15.84 | 17.00 | 15.10 |
| | ARD | 63.12 | 72.36 | 33.04 | 32.78 | 32.22 | 31.24 | 61.66 | 70.10 | 26.76 | 26.48 | 27.16 | 25.82 |
| | IAD | 58.42 | 66.52 | 34.10 | 34.06 | 33.24 | 32.24 | 57.78 | 64.76 | 19.72 | 19.56 | 20.10 | 18.66 |
| | RSLAD | 61.30 | 69.46 | 32.84 | 32.88 | 32.64 | 31.54 | 58.40 | 65.74 | 20.52 | 20.22 | 21.12 | 19.76 |
| Logits | Vanilla | 76.02 | 82.50 | 2.92 | 2.08 | 0.94 | 0.40 | 76.10 | 82.28 | 2.84 | 2.00 | 0.96 | 0.38 |
| | JBDA | 66.68 | 72.74 | 5.50 | 4.54 | 3.40 | 2.48 | 61.42 | 66.04 | 4.44 | 3.68 | 3.74 | 2.32 |
| | ARD | 61.78 | 70.40 | 30.00 | 29.88 | 27.34 | 25.88 | 60.22 | 69.44 | 27.38 | 27.30 | 24.90 | 23.26 |
| | IAD | 64.64 | 73.36 | 31.38 | 31.20 | 28.94 | 27.26 | 61.36 | 70.56 | 27.06 | 26.78 | 24.34 | 22.58 |
| | RSLAD | 65.66 | 74.38 | 29.00 | 28.74 | 26.38 | 24.70 | 62.40 | 71.38 | 25.38 | 25.14 | 22.10 | 20.40 |
| BEST | | 72.18 | 82.32 | 31.02 | 30.82 | 31.80 | 30.30 | 71.26 | 81.26 | 30.42 | 30.22 | 31.18 | 30.12 |

Table 13: Results of model extraction attacks on CIFAR10. The victim model is ResNet-TRADES. The adversary model is WRN.

| | Method | Extracted model with the highest robustness | | | | | | Final model after extraction | | | | | |
|---|---|---|---|---|---|---|---|---|---|---|---|---|---|
| | | CA | rCA | RA | | | | CA | rCA | RA | | | |
| | | | | PGD20 | PGD100 | CW100 | AA | | | PGD20 | PGD100 | CW100 | AA |
| Label | Vanilla | 71.52 | 77.08 | 4.30 | 3.88 | 4.14 | 2.28 | 76.04 | 81.22 | 1.28 | 1.02 | 1.30 | 0.56 |
| | JBDA | 64.36 | 70.02 | 17.54 | 17.30 | 18.46 | 16.88 | 64.24 | 69.34 | 14.00 | 13.64 | 14.82 | 13.18 |
| | ARD | 57.82 | 65.02 | 32.64 | 32.58 | 31.22 | 30.24 | 58.78 | 65.62 | 19.10 | 18.88 | 19.76 | 18.10 |
| | IAD | 58.78 | 66.44 | 33.64 | 33.52 | 32.88 | 31.66 | 56.54 | 63.36 | 19.72 | 19.56 | 20.08 | 18.94 |
| | RSLAD | 59.58 | 66.70 | 31.02 | 30.96 | 30.14 | 28.86 | 54.00 | 60.74 | 15.26 | 15.10 | 15.66 | 14.48 |
| Logits | Vanilla | 76.64 | 81.88 | 2.58 | 1.84 | 1.16 | 0.50 | 76.32 | 81.66 | 2.42 | 1.76 | 0.96 | 0.32 |
| | JBDA | 50.02 | 52.24 | 5.80 | 5.12 | 4.68 | 3.52 | 55.96 | 60.06 | 4.38 | 3.56 | 3.10 | 2.02 |
| | ARD | 62.68 | 70.54 | 29.78 | 29.54 | 27.46 | 25.80 | 59.40 | 67.14 | 25.08 | 24.90 | 22.46 | 21.02 |
| | IAD | 61.98 | 69.70 | 30.26 | 29.98 | 27.34 | 25.96 | 58.48 | 66.02 | 23.44 | 23.12 | 21.38 | 19.66 |
| | RSLAD | 63.56 | 71.06 | 28.08 | 27.94 | 25.32 | 24.08 | 61.32 | 68.56 | 21.80 | 21.34 | 18.80 | 17.18 |
| BEST | | 70.76 | 80.02 | 31.24 | 31.08 | 31.70 | 30.72 | 70.30 | 79.58 | 29.80 | 29.54 | 30.36 | 29.22 |

Table 14: Results of model extraction attacks on CIFAR10. The victim model is WRN-TRADES. The adversary model is WRN.

| | Method | Extracted model with the highest robustness | | | | | | Final model after extraction | | | | | |
|---|---|---|---|---|---|---|---|---|---|---|---|---|---|
| | | CA | rCA | RA | | | | CA | rCA | RA | | | |
| | | | | PGD20 | PGD100 | CW100 | AA | | | PGD20 | PGD100 | CW100 | AA |
| Label | Vanilla | 65.24 | 72.24 | 7.86 | 7.30 | 7.60 | 6.36 | 73.10 | 80.64 | 6.24 | 5.64 | 6.20 | 4.86 |
| | JBDA | 57.80 | 64.40 | 32.06 | 31.74 | 19.22 | 17.36 | 55.26 | 59.76 | 24.38 | 23.82 | 15.88 | 14.24 |
| | ARD | 49.84 | 57.18 | 30.64 | 30.60 | 29.24 | 27.90 | 46.50 | 52.76 | 27.02 | 27.08 | 25.82 | 24.64 |
| | IAD | 48.68 | 55.28 | 30.48 | 30.30 | 29.54 | 27.98 | 44.88 | 50.94 | 27.32 | 27.30 | 25.90 | 24.88 |
| | RSLAD | 50.82 | 58.54 | 29.96 | 29.82 | 29.14 | 27.60 | 45.46 | 52.38 | 26.84 | 26.92 | 25.44 | 24.46 |
| Logits | Vanilla | 13.92 | 13.36 | 10.00 | 10.00 | 0.30 | 0.02 | 73.50 | 79.78 | 2.30 | 1.66 | 2.00 | 0.58 |
| | JBDA | 54.00 | 58.80 | 13.70 | 13.02 | 14.00 | 10.28 | 48.10 | 51.58 | 10.12 | 9.60 | 10.20 | 8.06 |
| | ARD | 58.98 | 66.64 | 27.80 | 27.58 | 25.90 | 23.66 | 57.84 | 64.88 | 24.94 | 24.58 | 23.48 | 21.18 |
| | IAD | 54.02 | 61.08 | 26.40 | 26.16 | 24.22 | 22.06 | 48.40 | 54.58 | 22.04 | 21.96 | 20.44 | 18.58 |
| | RSLAD | 57.80 | 66.16 | 25.64 | 25.24 | 23.72 | 22.10 | 54.84 | 61.92 | 22.32 | 22.14 | 21.02 | 19.18 |
| BEST | | 66.84 | 77.18 | 27.54 | 27.46 | 27.88 | 26.76 | 66.34 | 76.02 | 25.42 | 25.34 | 25.68 | 24.68 |

Table 15: Results of model extraction attacks on CIFAR10. The victim model is ResNet-AT. The adversary model is VGG.

| | Method | Extracted model with the highest robustness | | | | | | Final model after extraction | | | | | |
|---|---|---|---|---|---|---|---|---|---|---|---|---|---|
| | | CA | rCA | RA | | | | CA | rCA | RA | | | |
| | | | | PGD20 | PGD100 | CW100 | AA | | | PGD20 | PGD100 | CW100 | AA |
| Label | Vanilla | 45.70 | 48.00 | 5.94 | 5.74 | 8.18 | 4.60 | 74.78 | 79.22 | 4.56 | 3.86 | 4.50 | 3.38 |
| | JBDA | 59.98 | 64.22 | 33.38 | 32.76 | 18.42 | 16.74 | 62.38 | 67.04 | 29.00 | 27.90 | 17.00 | 15.18 |
| | ARD | 49.46 | 54.24 | 30.34 | 30.40 | 28.30 | 27.06 | 47.72 | 52.06 | 27.76 | 27.72 | 26.62 | 25.60 |
| | IAD | 50.12 | 54.36 | 28.22 | 28.00 | 27.58 | 26.08 | 40.44 | 43.28 | 23.84 | 23.78 | 22.68 | 21.94 |
| | RSLAD | 57.14 | 62.72 | 30.64 | 30.60 | 29.84 | 28.50 | 56.44 | 61.68 | 27.42 | 27.34 | 27.08 | 25.72 |
| Logits | Vanilla | 10.00 | 8.62 | 10.00 | 10.00 | 10.00 | 10.00 | 76.40 | 80.56 | 1.08 | 0.66 | 0.96 | 0.30 |
| | JBDA | 56.24 | 59.18 | 13.12 | 12.22 | 12.82 | 9.70 | 58.78 | 62.14 | 11.74 | 11.04 | 11.84 | 9.32 |
| | ARD | 58.76 | 64.74 | 27.08 | 26.86 | 26.38 | 23.50 | 56.38 | 61.34 | 24.20 | 23.94 | 23.50 | 20.62 |
| | IAD | 53.00 | 56.58 | 25.38 | 25.22 | 23.94 | 21.52 | 50.00 | 53.00 | 21.64 | 21.48 | 20.66 | 17.86 |
| | RSLAD | 57.92 | 63.54 | 24.12 | 23.94 | 23.38 | 21.54 | 50.78 | 55.62 | 19.60 | 19.22 | 18.56 | 17.18 |
| BEST | | 67.84 | 74.50 | 27.92 | 27.66 | 28.16 | 26.90 | 68.72 | 75.50 | 27.08 | 26.88 | 27.46 | 26.32 |

Table 16: Results of model extraction attacks on CIFAR10. The victim model is WRN-AT. The adversary model is VGG.

| | Method | Extracted model with the highest robustness | | | | | | Final model after extraction | | | | | |
|---|---|---|---|---|---|---|---|---|---|---|---|---|---|
| | | CA | rCA | RA | | | | CA | rCA | RA | | | |
| | | | | PGD20 | PGD100 | CW100 | AA | | | PGD20 | PGD100 | CW100 | AA |
| Label | Vanilla | 40.22 | 43.40 | 8.30 | 8.16 | 9.24 | 7.44 | 71.30 | 77.94 | 6.78 | 6.14 | 6.58 | 5.42 |
| | JBDA | 51.74 | 58.14 | 33.46 | 33.18 | 19.06 | 18.16 | 49.22 | 52.70 | 21.98 | 21.50 | 15.44 | 13.76 |
| | ARD | 49.96 | 57.86 | 31.68 | 31.74 | 29.92 | 29.00 | 46.84 | 52.98 | 28.50 | 28.46 | 26.50 | 25.72 |
| | IAD | 46.02 | 52.88 | 29.88 | 29.84 | 27.38 | 26.22 | 41.92 | 47.40 | 26.40 | 26.32 | 25.36 | 24.16 |
| | RSLAD | 50.98 | 59.10 | 31.38 | 31.48 | 29.22 | 28.36 | 51.38 | 59.38 | 28.58 | 28.64 | 27.26 | 26.18 |
| Logits | Vanilla | 10.00 | 8.54 | 10.00 | 10.00 | 9.98 | 10.00 | 69.48 | 76.08 | 2.66 | 1.90 | 1.78 | 0.82 |
| | JBDA | 49.10 | 53.44 | 12.76 | 12.00 | 13.72 | 9.16 | 46.18 | 49.44 | 8.62 | 8.18 | 9.08 | 7.00 |
| | ARD | 52.44 | 60.28 | 27.98 | 27.94 | 25.30 | 23.24 | 51.86 | 59.12 | 25.84 | 25.66 | 23.52 | 21.44 |
| | IAD | 51.40 | 59.10 | 26.74 | 26.62 | 24.46 | 22.24 | 46.96 | 53.88 | 24.12 | 24.04 | 21.64 | 19.56 |
| | RSLAD | 52.02 | 60.70 | 27.22 | 27.16 | 24.82 | 23.56 | 52.02 | 60.14 | 24.82 | 24.58 | 23.24 | 21.52 |
| BEST | | 65.26 | 75.40 | 28.54 | 28.42 | 29.24 | 27.72 | 66.12 | 75.22 | 27.66 | 27.52 | 28.24 | 26.78 |

Table 17: Results of model extraction attacks on CIFAR10. The victim model is ResNet-TRADES. The adversary model is VGG.

| | Method | Extracted model with the highest robustness | | | | | | Final model after extraction | | | | | |
|---|---|---|---|---|---|---|---|---|---|---|---|---|---|
| | | CA | rCA | RA | | | | CA | rCA | RA | | | |
| | | | | PGD20 | PGD100 | CW100 | AA | | | PGD20 | PGD100 | CW100 | AA |
| Label | Vanilla | 43.68 | 47.16 | 7.78 | 7.42 | 8.20 | 6.76 | 71.48 | 77.60 | 5.94 | 5.22 | 5.88 | 4.30 |
| | JBDA | 53.78 | 58.90 | 31.40 | 31.14 | 21.14 | 19.68 | 56.26 | 60.98 | 27.82 | 27.16 | 19.04 | 17.48 |
| | ARD | 45.92 | 52.78 | 29.64 | 29.66 | 27.48 | 26.56 | 42.86 | 49.32 | 26.88 | 26.82 | 25.08 | 24.48 |
| | IAD | 43.80 | 49.90 | 29.12 | 29.12 | 27.22 | 26.38 | 37.74 | 43.10 | 25.30 | 25.30 | 23.96 | 23.40 |
| | RSLAD | 50.90 | 58.20 | 31.84 | 31.80 | 29.54 | 28.66 | 50.08 | 57.46 | 28.50 | 28.48 | 27.38 | 26.50 |
| Logits | Vanilla | 10.00 | 10.38 | 10.00 | 10.00 | 10.00 | 10.00 | 71.28 | 77.28 | 2.08 | 1.34 | 1.30 | 0.46 |
| | JBDA | 48.56 | 52.00 | 12.02 | 11.24 | 12.64 | 8.74 | 54.86 | 59.70 | 9.96 | 9.40 | 10.58 | 7.70 |
| | ARD | 51.48 | 58.52 | 27.18 | 27.12 | 25.36 | 23.40 | 49.38 | 56.00 | 25.68 | 25.52 | 23.20 | 21.36 |
| | IAD | 49.16 | 55.72 | 26.98 | 26.86 | 24.50 | 22.98 | 46.26 | 52.68 | 23.88 | 23.74 | 21.84 | 20.26 |
| | RSLAD | 53.92 | 61.76 | 27.20 | 26.98 | 24.92 | 23.72 | 50.02 | 57.20 | 24.84 | 24.74 | 22.80 | 21.46 |
| BEST | | 65.76 | 75.14 | 28.88 | 28.54 | 29.20 | 27.88 | 66.22 | 75.24 | 27.72 | 27.70 | 28.36 | 27.22 |

Table 18: Results of model extraction attacks on CIFAR10. The victim model is WRN-TRADES. The adversary model is VGG.

| | Method | Extracted model with the highest robustness | | | | | | Final model after extraction | | | | | |
|---|---|---|---|---|---|---|---|---|---|---|---|---|---|
| | | CA | rCA | RA | | | | CA | rCA | RA | | | |
| | | | | PGD20 | PGD100 | CW100 | AA | | | PGD20 | PGD100 | CW100 | AA |
| Label | Vanilla | 62.12 | 67.66 | 3.54 | 3.06 | 3.26 | 2.42 | 72.28 | 79.80 | 2.00 | 1.60 | 2.02 | 0.92 |
| | JBDA | 51.26 | 57.24 | 12.22 | 11.96 | 12.92 | 11.40 | 53.34 | 59.02 | 6.40 | 6.08 | 7.00 | 5.42 |
| | ARD | 57.30 | 66.32 | 32.58 | 32.50 | 31.26 | 30.00 | 56.46 | 64.96 | 26.82 | 26.58 | 26.32 | 25.12 |
| | IAD | 58.42 | 67.64 | 32.18 | 32.04 | 31.32 | 30.06 | 55.56 | 63.98 | 26.36 | 26.18 | 26.04 | 25.20 |
| | RSLAD | 58.32 | 67.10 | 31.56 | 31.26 | 30.78 | 29.94 | 57.14 | 65.66 | 26.38 | 26.14 | 26.16 | 25.18 |
| Logits | Vanilla | 74.58 | 80.32 | 2.08 | 1.46 | 1.38 | 0.48 | 75.88 | 81.62 | 1.68 | 1.20 | 0.98 | 0.44 |
| | JBDA | 62.24 | 68.68 | 11.62 | 10.84 | 11.02 | 8.58 | 62.10 | 68.58 | 7.82 | 7.34 | 7.86 | 6.08 |
| | ARD | 66.06 | 74.96 | 30.28 | 30.16 | 28.48 | 26.60 | 64.22 | 72.40 | 27.38 | 26.92 | 25.18 | 23.32 |
| | IAD | 62.80 | 71.80 | 31.08 | 30.70 | 27.82 | 26.18 | 59.54 | 67.72 | 25.32 | 25.18 | 22.36 | 20.72 |
| | RSLAD | 64.22 | 74.24 | 31.46 | 31.30 | 29.22 | 27.58 | 62.82 | 71.84 | 27.28 | 27.08 | 24.34 | 22.64 |
| BEST | | 68.50 | 78.54 | 26.52 | 26.22 | 26.96 | 25.72 | 68.70 | 78.48 | 24.74 | 24.46 | 25.44 | 23.88 |

Table 19: Results of model extraction attacks on CIFAR10. The victim model is ResNet-AT. The adversary model is MobileNet.

| | Method | Extracted model with the highest robustness | | | | | | Final model after extraction | | | | | |
|---|---|---|---|---|---|---|---|---|---|---|---|---|---|
| | | CA | rCA | RA | | | | CA | rCA | RA | | | |
| | | | | PGD20 | PGD100 | CW100 | AA | | | PGD20 | PGD100 | CW100 | AA |
| Label | Vanilla | 51.26 | 53.04 | 3.78 | 3.54 | 3.30 | 2.64 | 72.92 | 77.06 | 0.70 | 0.48 | 0.72 | 0.24 |
| | JBDA | 44.28 | 48.04 | 10.60 | 10.42 | 11.12 | 10.10 | 47.34 | 51.68 | 8.20 | 7.94 | 8.52 | 7.48 |
| | ARD | 55.50 | 61.06 | 30.60 | 30.62 | 29.04 | 27.78 | 46.58 | 51.16 | 23.56 | 23.42 | 21.98 | 21.28 |
| | IAD | 53.80 | 59.52 | 31.52 | 31.38 | 29.38 | 28.10 | 44.28 | 48.40 | 24.06 | 24.04 | 22.54 | 21.76 |
| | RSLAD | 59.62 | 65.54 | 30.42 | 30.32 | 30.24 | 29.02 | 54.30 | 54.74 | 21.62 | 21.36 | 21.04 | 20.08 |
| Logits | Vanilla | 76.04 | 80.30 | 1.20 | 0.76 | 0.58 | 0.28 | 76.62 | 80.62 | 1.02 | 0.62 | 0.60 | 0.14 |
| | JBDA | 64.78 | 69.32 | 9.80 | 8.94 | 9.78 | 7.28 | 59.90 | 64.34 | 7.98 | 7.40 | 8.04 | 6.38 |
| | ARD | 65.48 | 71.02 | 27.72 | 27.24 | 26.80 | 24.42 | 60.54 | 65.90 | 23.66 | 23.36 | 21.58 | 19.94 |
| | IAD | 60.84 | 66.02 | 28.40 | 28.34 | 25.56 | 24.02 | 54.30 | 58.70 | 21.74 | 21.56 | 18.52 | 17.02 |
| | RSLAD | 63.34 | 68.38 | 28.54 | 28.54 | 25.90 | 24.04 | 61.18 | 66.08 | 23.84 | 23.70 | 20.52 | 18.90 |
| BEST | | 70.18 | 76.38 | 24.48 | 24.16 | 25.04 | 23.58 | 69.20 | 75.06 | 23.44 | 23.12 | 24.16 | 22.52 |

Table 20: Results of model extraction attacks on CIFAR10. The victim model is WRN-AT. The adversary model is MobileNet.

| | Method | Extracted model with the highest robustness | | | | | | Final model after extraction | | | | | |
|---|---|---|---|---|---|---|---|---|---|---|---|---|---|
| | | CA | rCA | RA | | | | CA | rCA | RA | | | |
| | | | | PGD20 | PGD100 | CW100 | AA | | | PGD20 | PGD100 | CW100 | AA |
| Label | Vanilla | 65.02 | 70.86 | 3.14 | 2.76 | 2.70 | 1.86 | 70.88 | 78.26 | 1.92 | 1.60 | 1.86 | 0.78 |
| | JBDA | 59.14 | 65.92 | 14.24 | 13.86 | 14.88 | 13.38 | 49.78 | 55.42 | 9.40 | 9.24 | 9.88 | 8.64 |
| | ARD | 56.10 | 65.48 | 33.10 | 33.08 | 31.80 | 30.84 | 55.14 | 63.98 | 26.84 | 26.62 | 26.22 | 25.24 |
| | IAD | 56.56 | 65.02 | 33.12 | 33.04 | 32.06 | 30.78 | 55.42 | 63.90 | 27.10 | 26.88 | 26.14 | 25.10 |
| | RSLAD | 57.16 | 66.78 | 32.26 | 32.22 | 31.40 | 30.14 | 59.34 | 68.94 | 26.90 | 26.70 | 26.86 | 25.84 |
| Logits | Vanilla | 70.86 | 77.18 | 3.14 | 2.28 | 1.34 | 0.70 | 72.20 | 78.56 | 2.42 | 1.84 | 1.06 | 0.50 |
| | JBDA | 61.10 | 67.46 | 10.96 | 10.12 | 8.92 | 7.38 | 58.40 | 64.16 | 6.58 | 6.16 | 5.98 | 4.94 |
| | ARD | 59.70 | 68.86 | 32.28 | 32.18 | 29.20 | 28.08 | 59.28 | 68.40 | 30.86 | 30.66 | 27.80 | 26.30 |
| | IAD | 59.00 | 68.20 | 32.34 | 32.14 | 29.02 | 27.72 | 56.12 | 64.82 | 28.36 | 28.32 | 25.22 | 24.08 |
| | RSLAD | 63.72 | 73.86 | 31.98 | 31.72 | 29.12 | 27.28 | 57.96 | 67.12 | 28.14 | 27.96 | 24.48 | 23.56 |
| BEST | | 67.14 | 76.60 | 25.88 | 25.60 | 26.28 | 25.12 | 67.14 | 76.30 | 24.86 | 24.66 | 25.22 | 24.04 |

Table 21: Results of model extraction attacks on CIFAR10. The victim model is ResNet-TRADES. The adversary model is MobileNet.

| | Method | Extracted model with the highest robustness | | | | | | Final model after extraction | | | | | |
|---|---|---|---|---|---|---|---|---|---|---|---|---|---|
| | | CA | rCA | RA | | | | CA | rCA | RA | | | |
| | | | | PGD20 | PGD100 | CW100 | AA | | | PGD20 | PGD100 | CW100 | AA |
| Label | Vanilla | 52.08 | 57.34 | 5.92 | 5.28 | 3.78 | 3.14 | 70.00 | 76.36 | 1.64 | 1.22 | 1.28 | 0.76 |
| | JBDA | 55.16 | 60.84 | 13.76 | 13.56 | 14.18 | 12.98 | 49.44 | 53.98 | 9.94 | 9.80 | 10.26 | 9.22 |
| | ARD | 52.68 | 61.08 | 32.58 | 32.62 | 30.90 | 30.34 | 47.90 | 55.40 | 26.20 | 26.14 | 25.18 | 24.46 |
| | IAD | 54.24 | 62.60 | 32.48 | 32.40 | 31.12 | 30.18 | 48.88 | 56.24 | 26.70 | 26.70 | 25.70 | 24.86 |
| | RSLAD | 57.48 | 65.78 | 31.50 | 31.50 | 30.62 | 29.80 | 50.92 | 58.58 | 25.02 | 25.08 | 24.22 | 23.46 |
| Logits | Vanilla | 69.32 | 74.80 | 2.56 | 1.88 | 1.10 | 0.56 | 71.72 | 77.30 | 2.20 | 1.68 | 0.64 | 0.34 |
| | JBDA | 59.52 | 64.86 | 10.60 | 9.80 | 9.38 | 7.94 | 54.96 | 59.62 | 8.94 | 8.62 | 8.30 | 7.02 |
| | ARD | 59.42 | 67.80 | 31.42 | 31.32 | 28.28 | 26.86 | 56.64 | 64.42 | 27.60 | 27.36 | 24.00 | 23.08 |
| | IAD | 58.20 | 65.40 | 30.90 | 30.86 | 28.12 | 26.84 | 52.48 | 59.52 | 25.54 | 25.42 | 21.60 | 20.72 |
| | RSLAD | 61.84 | 70.20 | 30.84 | 30.58 | 27.50 | 26.22 | 58.08 | 65.98 | 27.80 | 27.60 | 23.56 | 22.56 |
| BEST | | 66.40 | 74.84 | 25.82 | 25.46 | 26.26 | 25.04 | 66.02 | 74.34 | 24.80 | 24.58 | 25.22 | 24.14 |

Table 22: Results of model extraction attacks on CIFAR10. The victim model is WRN-TRADES. The adversary model is MobileNet.

| | Method | Extracted model with the highest robustness | | | | | | Final model after extraction | | | | | |
|---|---|---|---|---|---|---|---|---|---|---|---|---|---|
| | | CA | rCA | RA | | | | CA | rCA | RA | | | |
| | | | | PGD20 | PGD100 | CW100 | AA | | | PGD20 | PGD100 | CW100 | AA |
| Label | Vanilla | 38.30 | 50.72 | 3.22 | 2.86 | 3.38 | 2.24 | 45.78 | 61.54 | 2.54 | 2.24 | 2.82 | 1.86 |
| | JBDA | 31.42 | 41.94 | 7.74 | 7.56 | 8.32 | 7.14 | 28.62 | 38.02 | 4.96 | 4.92 | 5.42 | 4.50 |
| | ARD | 33.64 | 48.86 | 14.80 | 14.66 | 14.24 | 13.66 | 31.72 | 46.40 | 11.42 | 11.26 | 11.36 | 10.44 |
| | IAD | 34.52 | 50.50 | 14.96 | 14.78 | 14.62 | 13.62 | 31.58 | 45.54 | 10.60 | 10.50 | 10.58 | 9.92 |
| | RSLAD | 31.26 | 43.36 | 13.92 | 13.76 | 12.96 | 12.22 | 31.48 | 45.34 | 10.16 | 10.16 | 10.08 | 9.48 |
| Logits | Vanilla | 38.86 | 47.88 | 2.10 | 1.70 | 1.42 | 0.58 | 38.56 | 47.22 | 2.02 | 1.72 | 1.46 | 0.62 |
| | JBDA | 17.16 | 21.28 | 1.52 | 1.28 | 1.20 | 0.64 | 16.62 | 19.94 | 0.94 | 0.68 | 0.80 | 0.42 |
| | ARD | 13.32 | 18.62 | 5.04 | 5.02 | 2.54 | 2.04 | 13.42 | 18.90 | 4.36 | 4.26 | 2.52 | 1.90 |
| | IAD | 16.26 | 22.54 | 10.32 | 10.28 | 8.52 | 8.00 | 15.72 | 21.74 | 9.08 | 8.98 | 7.44 | 6.92 |
| | RSLAD | 23.74 | 32.96 | 10.88 | 10.92 | 7.62 | 6.96 | 23.28 | 32.56 | 10.24 | 10.24 | 7.50 | 6.52 |
| BEST | | 44.38 | 64.50 | 14.26 | 14.22 | 15.24 | 13.74 | 44.78 | 65.96 | 13.72 | 13.52 | 14.56 | 13.14 |

Table 23: Results of model extraction attacks on CIFAR100. The victim model is ResNet-AT. The adversary model is ResNet.

| | Method | Extracted model with the highest robustness | | | | | | Final model after extraction | | | | | |
|---|---|---|---|---|---|---|---|---|---|---|---|---|---|
| | | CA | rCA | RA | | | | CA | rCA | RA | | | |
| | | | | PGD20 | PGD100 | CW100 | AA | | | PGD20 | PGD100 | CW100 | AA |
| Label | Vanilla | 41.60 | 51.30 | 2.22 | 1.94 | 2.46 | 1.62 | 47.40 | 58.92 | 1.44 | 1.16 | 1.56 | 0.92 |
| | JBDA | 31.54 | 38.58 | 6.86 | 6.70 | 7.20 | 6.24 | 28.36 | 34.84 | 4.54 | 4.42 | 4.82 | 4.16 |
| | ARD | 31.22 | 41.60 | 13.30 | 13.14 | 12.42 | 11.62 | 29.46 | 39.90 | 9.56 | 9.38 | 9.40 | 8.64 |
| | IAD | 34.40 | 46.06 | 13.80 | 13.60 | 13.66 | 12.76 | 31.38 | 41.86 | 9.76 | 9.58 | 9.64 | 8.92 |
| | RSLAD | 33.22 | 44.24 | 13.20 | 13.04 | 12.68 | 11.72 | 31.58 | 41.80 | 9.36 | 9.26 | 9.18 | 8.52 |
| Logits | Vanilla | 40.58 | 47.96 | 1.62 | 1.32 | 1.08 | 0.34 | 40.04 | 47.70 | 1.44 | 1.18 | 1.10 | 0.34 |
| | JBDA | 18.00 | 21.04 | 1.70 | 1.48 | 1.50 | 0.94 | 18.54 | 21.84 | 1.24 | 1.08 | 1.10 | 0.78 |
| | ARD | 13.28 | 16.60 | 4.72 | 4.64 | 2.28 | 1.98 | 13.02 | 16.38 | 4.24 | 4.18 | 2.38 | 2.06 |
| | IAD | 17.22 | 22.36 | 10.44 | 10.34 | 8.60 | 8.08 | 16.68 | 22.04 | 9.14 | 9.18 | 7.70 | 7.18 |
| | RSLAD | 25.56 | 32.08 | 10.76 | 10.78 | 7.70 | 6.82 | 23.62 | 30.40 | 9.20 | 9.24 | 6.78 | 5.92 |
| BEST | | 45.52 | 60.96 | 12.36 | 12.20 | 13.34 | 11.72 | 45.86 | 61.12 | 11.78 | 11.52 | 12.66 | 11.10 |

Table 24: Results of model extraction attacks on CIFAR100. The victim model is WRN-AT. The adversary model is ResNet.

| | Method | Extracted model with the highest robustness | | | | | | Final model after extraction | | | | | |
|---|---|---|---|---|---|---|---|---|---|---|---|---|---|
| | | CA | rCA | RA | | | | CA | rCA | RA | | | |
| | | | | PGD20 | PGD100 | CW100 | AA | | | PGD20 | PGD100 | CW100 | AA |
| Label | Vanilla | 32.42 | 42.80 | 2.94 | 2.68 | 3.30 | 2.28 | 43.42 | 56.40 | 1.80 | 1.74 | 2.20 | 1.34 |
| | JBDA | 32.16 | 42.30 | 7.72 | 7.68 | 8.16 | 7.30 | 29.28 | 38.38 | 5.76 | 5.76 | 6.20 | 5.62 |
| | ARD | 29.66 | 42.10 | 13.66 | 13.52 | 12.42 | 11.78 | 28.86 | 41.84 | 9.58 | 9.48 | 9.50 | 8.90 |
| | IAD | 27.58 | 39.96 | 13.60 | 13.48 | 12.66 | 11.80 | 28.50 | 40.44 | 9.08 | 8.94 | 8.86 | 8.30 |
| | RSLAD | 32.10 | 45.52 | 13.24 | 13.24 | 12.96 | 12.18 | 30.08 | 42.64 | 10.36 | 10.28 | 10.34 | 9.62 |
| Logits | Vanilla | 31.98 | 39.84 | 2.50 | 2.14 | 1.22 | 0.56 | 32.50 | 40.66 | 2.18 | 1.98 | 1.10 | 0.56 |
| | JBDA | 14.44 | 18.58 | 1.02 | 0.88 | 0.72 | 0.36 | 14.20 | 18.28 | 0.62 | 0.48 | 0.46 | 0.20 |
| | ARD | 10.40 | 16.02 | 4.08 | 4.02 | 2.00 | 1.76 | 10.60 | 16.36 | 3.76 | 3.78 | 1.90 | 1.64 |
| | IAD | 14.10 | 20.44 | 9.46 | 9.42 | 7.62 | 7.22 | 13.68 | 19.88 | 8.46 | 8.52 | 6.96 | 6.52 |
| | RSLAD | 19.32 | 27.98 | 9.30 | 9.28 | 6.90 | 6.32 | 19.32 | 28.26 | 8.72 | 8.58 | 6.50 | 5.90 |
| BEST | | 41.86 | 60.44 | 12.64 | 12.48 | 13.28 | 12.14 | 42.30 | 60.74 | 12.26 | 12.18 | 12.98 | 11.78 |

Table 25: Results of model extraction attacks on CIFAR100. The victim model is ResNet-TRADES. The adversary model is ResNet.

| | Method | Extracted model with the highest robustness | | | | | | Final model after extraction | | | | | |
|---|---|---|---|---|---|---|---|---|---|---|---|---|---|
| | | CA | rCA | RA | | | | CA | rCA | RA | | | |
| | | | | PGD20 | PGD100 | CW100 | AA | | | PGD20 | PGD100 | CW100 | AA |
| Label | Vanilla | 39.16 | 48.00 | 2.76 | 2.46 | 2.82 | 1.90 | 44.88 | 56.32 | 1.54 | 1.34 | 1.72 | 1.10 |
| | JBDA | 28.96 | 37.06 | 7.60 | 7.44 | 8.04 | 7.24 | 29.22 | 37.28 | 6.50 | 6.36 | 6.74 | 5.92 |
| | ARD | 29.80 | 40.82 | 13.66 | 13.68 | 12.60 | 11.70 | 29.40 | 40.56 | 9.26 | 9.14 | 9.32 | 8.62 |
| | IAD | 31.98 | 44.12 | 13.62 | 13.58 | 12.84 | 12.32 | 29.34 | 40.74 | 9.32 | 9.16 | 9.34 | 8.56 |
| | RSLAD | 32.30 | 43.92 | 13.56 | 13.54 | 12.78 | 11.78 | 30.32 | 40.88 | 9.40 | 9.36 | 9.18 | 8.66 |
| Logits | Vanilla | 33.66 | 40.44 | 2.52 | 2.12 | 1.18 | 0.60 | 33.48 | 40.60 | 2.42 | 2.00 | 1.04 | 0.54 |
| | JBDA | 15.02 | 18.28 | 1.08 | 0.80 | 0.66 | 0.34 | 14.52 | 17.80 | 0.94 | 0.70 | 0.84 | 0.36 |
| | ARD | 9.60 | 13.44 | 2.98 | 3.00 | 1.26 | 1.12 | 9.58 | 13.42 | 2.88 | 2.80 | 1.30 | 1.12 |
| | IAD | 15.52 | 20.18 | 9.58 | 9.66 | 7.90 | 7.48 | 15.00 | 19.44 | 8.82 | 8.72 | 7.16 | 6.54 |
| | RSLAD | 21.54 | 28.76 | 9.52 | 9.44 | 6.90 | 6.04 | 20.60 | 27.26 | 8.64 | 8.54 | 6.28 | 5.58 |
| BEST | | 42.80 | 58.26 | 12.32 | 12.06 | 12.96 | 11.56 | 43.04 | 58.18 | 12.00 | 11.88 | 12.70 | 11.42 |

Table 26: Results of model extraction attacks on CIFAR100. The victim model is WRN-TRADES. The adversary model is ResNet.

| | Method | Extracted model with the highest robustness | | | | | | Final model after extraction | | | | | |
|---|---|---|---|---|---|---|---|---|---|---|---|---|---|
| | | CA | rCA | RA | | | | CA | rCA | RA | | | |
| | | | | PGD20 | PGD100 | CW100 | AA | | | PGD20 | PGD100 | CW100 | AA |
| Label | Vanilla | 40.12 | 52.04 | 3.50 | 3.06 | 3.96 | 2.38 | 47.94 | 64.54 | 2.72 | 2.32 | 2.82 | 1.64 |
| | JBDA | 32.56 | 42.70 | 7.68 | 7.56 | 8.22 | 7.14 | 26.60 | 35.62 | 3.82 | 3.76 | 4.46 | 3.52 |
| | ARD | 33.92 | 49.10 | 14.18 | 14.12 | 14.00 | 13.16 | 31.44 | 45.22 | 9.82 | 9.62 | 9.90 | 9.14 |
| | IAD | 34.78 | 48.78 | 14.68 | 14.64 | 14.30 | 13.54 | 30.34 | 43.14 | 9.64 | 9.54 | 9.60 | 9.00 |
| | RSLAD | 34.58 | 49.10 | 14.58 | 14.52 | 13.68 | 12.76 | 32.00 | 44.98 | 9.46 | 9.30 | 9.54 | 8.64 |
| Logits | Vanilla | 35.36 | 40.90 | 2.38 | 2.10 | 0.92 | 0.30 | 35.32 | 41.22 | 2.08 | 1.86 | 0.82 | 0.26 |
| | JBDA | 12.98 | 15.20 | 1.24 | 1.12 | 0.42 | 0.24 | 11.08 | 13.28 | 0.82 | 0.78 | 0.30 | 0.16 |
| | ARD | 6.96 | 9.06 | 2.64 | 2.64 | 1.04 | 0.90 | 7.54 | 9.80 | 2.18 | 2.16 | 0.92 | 0.74 |
| | IAD | 13.08 | 17.00 | 8.02 | 8.00 | 6.36 | 5.82 | 12.16 | 16.88 | 6.94 | 6.88 | 5.48 | 5.10 |
| | RSLAD | 16.96 | 21.92 | 7.96 | 7.96 | 5.74 | 5.30 | 15.86 | 21.52 | 6.76 | 6.68 | 5.18 | 4.62 |
| BEST | | 47.20 | 68.56 | 15.78 | 15.66 | 17.00 | 15.02 | 46.90 | 67.82 | 15.40 | 15.24 | 16.62 | 14.68 |

Table 27: Results of model extraction attacks on CIFAR100. The victim model is ResNet-AT. The adversary model is WRN.

| | Method | Extracted model with the highest robustness | | | | | | Final model after extraction | | | | | |
|---|---|---|---|---|---|---|---|---|---|---|---|---|---|
| | | CA | rCA | RA | | | | CA | rCA | RA | | | |
| | | | | PGD20 | PGD100 | CW100 | AA | | | PGD20 | PGD100 | CW100 | AA |
| Label | Vanilla | 42.80 | 50.68 | 2.44 | 2.24 | 2.40 | 1.58 | 50.62 | 62.02 | 1.38 | 1.18 | 1.38 | 0.78 |
| | JBDA | 37.68 | 45.60 | 7.66 | 7.42 | 8.26 | 6.70 | 33.88 | 40.86 | 5.62 | 5.50 | 6.10 | 5.12 |
| | ARD | 34.58 | 45.26 | 14.28 | 14.12 | 13.54 | 12.54 | 30.32 | 40.14 | 8.12 | 7.94 | 8.34 | 7.56 |
| | IAD | 34.58 | 45.26 | 14.18 | 14.08 | 13.62 | 12.54 | 30.18 | 39.92 | 8.18 | 8.00 | 8.22 | 7.38 |
| | RSLAD | 37.68 | 48.60 | 14.10 | 13.86 | 13.96 | 13.08 | 32.96 | 44.00 | 9.10 | 8.94 | 9.20 | 8.36 |
| Logits | Vanilla | 35.28 | 39.52 | 1.74 | 1.44 | 0.92 | 0.32 | 35.70 | 39.98 | 1.56 | 1.30 | 0.94 | 0.28 |
| | JBDA | 12.18 | 13.84 | 1.12 | 0.92 | 0.40 | 0.14 | 11.52 | 12.72 | 0.78 | 0.64 | 0.36 | 0.12 |
| | ARD | 6.28 | 8.06 | 2.10 | 2.12 | 1.12 | 0.92 | 6.38 | 8.10 | 1.88 | 1.86 | 0.90 | 0.80 |
| | IAD | 14.24 | 17.54 | 8.32 | 8.32 | 6.52 | 6.14 | 13.32 | 17.08 | 7.06 | 7.04 | 5.38 | 4.92 |
| | RSLAD | 18.22 | 22.02 | 8.22 | 8.12 | 6.20 | 5.62 | 16.96 | 21.04 | 6.70 | 6.54 | 4.86 | 4.38 |
| BEST | | 49.22 | 65.68 | 14.84 | 14.64 | 15.78 | 14.16 | 49.20 | 64.80 | 13.66 | 13.44 | 14.68 | 12.94 |

Table 28: Results of model extraction attacks on CIFAR100. The victim model is WRN-AT. The adversary model is WRN.

| | Method | Extracted model with the highest robustness | | | | | | Final model after extraction | | | | | |
|---|---|---|---|---|---|---|---|---|---|---|---|---|---|
| | | CA | rCA | RA | | | | CA | rCA | RA | | | |
| | | | | PGD20 | PGD100 | CW100 | AA | | | PGD20 | PGD100 | CW100 | AA |
| Label | Vanilla | 37.48 | 46.98 | 2.92 | 2.68 | 3.06 | 1.80 | 46.64 | 58.62 | 1.18 | 1.02 | 1.34 | 0.74 |
| | JBDA | 32.56 | 42.90 | 8.40 | 8.24 | 8.94 | 8.08 | 32.60 | 43.34 | 8.00 | 7.84 | 8.44 | 7.56 |
| | ARD | 34.94 | 48.56 | 14.16 | 13.90 | 13.40 | 12.70 | 30.12 | 42.08 | 9.98 | 9.86 | 9.96 | 9.26 |
| | IAD | 34.46 | 48.18 | 14.84 | 14.80 | 14.26 | 13.54 | 30.68 | 42.54 | 11.28 | 11.16 | 11.18 | 10.44 |
| | RSLAD | 32.74 | 44.62 | 13.54 | 13.48 | 13.10 | 12.16 | 30.36 | 41.82 | 8.66 | 8.58 | 8.68 | 8.02 |
| Logits | Vanilla | 26.66 | 31.72 | 3.22 | 2.70 | 1.04 | 0.34 | 26.24 | 31.56 | 2.72 | 2.34 | 0.82 | 0.30 |
| | JBDA | 9.28 | 11.22 | 1.10 | 0.96 | 0.12 | 0.06 | 8.38 | 10.18 | 0.60 | 0.56 | 0.08 | 0.06 |
| | ARD | 6.52 | 8.62 | 1.78 | 1.74 | 0.58 | 0.44 | 6.84 | 9.44 | 1.48 | 1.40 | 0.42 | 0.34 |
| | IAD | 10.96 | 16.04 | 7.44 | 7.42 | 5.70 | 5.54 | 10.80 | 15.92 | 6.72 | 6.76 | 5.28 | 5.12 |
| | RSLAD | 14.72 | 20.18 | 6.56 | 6.50 | 4.34 | 4.00 | 14.10 | 19.06 | 5.96 | 5.86 | 4.08 | 3.70 |
| BEST | | 45.92 | 64.18 | 14.54 | 14.34 | 15.16 | 13.98 | 44.94 | 62.54 | 13.84 | 13.70 | 14.72 | 13.46 |

Table 29: Results of model extraction attacks on CIFAR100. The victim model is ResNet-TRADES. The adversary model is WRN.

| | Method | Extracted model with the highest robustness | | | | | | Final model after extraction | | | | | |
|---|---|---|---|---|---|---|---|---|---|---|---|---|---|
| | | CA | rCA | RA | | | | CA | rCA | RA | | | |
| | | | | PGD20 | PGD100 | CW100 | AA | | | PGD20 | PGD100 | CW100 | AA |
| Label | Vanilla | 40.12 | 49.52 | 2.82 | 2.48 | 2.88 | 1.76 | 49.42 | 61.02 | 1.62 | 1.46 | 1.70 | 1.12 |
| | JBDA | 34.80 | 44.12 | 8.56 | 8.42 | 9.08 | 8.10 | 29.48 | 37.16 | 6.62 | 6.44 | 7.06 | 6.28 |
| | ARD | 35.34 | 48.74 | 15.46 | 15.34 | 14.76 | 13.98 | 32.34 | 43.90 | 10.96 | 10.86 | 10.88 | 9.94 |
| | IAD | 34.70 | 46.94 | 14.48 | 14.34 | 14.12 | 13.12 | 28.80 | 38.80 | 8.54 | 8.54 | 8.52 | 7.80 |
| | RSLAD | 36.68 | 49.72 | 15.40 | 15.24 | 14.76 | 13.98 | 34.24 | 46.66 | 11.92 | 11.84 | 11.82 | 11.10 |
| Logits | Vanilla | 27.66 | 30.64 | 2.34 | 2.00 | 0.78 | 0.30 | 27.74 | 30.72 | 2.34 | 1.94 | 0.80 | 0.32 |
| | JBDA | 9.36 | 11.30 | 1.18 | 1.02 | 0.34 | 0.16 | 8.30 | 9.92 | 0.80 | 0.78 | 0.34 | 0.20 |
| | ARD | 5.30 | 6.78 | 1.52 | 1.54 | 0.48 | 0.42 | 5.38 | 6.82 | 1.38 | 1.38 | 0.42 | 0.36 |
| | IAD | 10.12 | 13.20 | 6.70 | 6.70 | 5.00 | 4.82 | 10.16 | 13.18 | 5.80 | 5.76 | 4.38 | 4.10 |
| | RSLAD | 14.04 | 17.50 | 6.12 | 6.08 | 4.04 | 3.58 | 14.04 | 17.34 | 5.30 | 5.22 | 3.60 | 3.22 |
| BEST | | 48.30 | 63.58 | 14.68 | 14.52 | 15.26 | 14.08 | 46.78 | 61.70 | 13.60 | 13.38 | 14.66 | 13.12 |

Table 30: Results of model extraction attacks on CIFAR100. The victim model is WRN-TRADES. The adversary model is WRN.

| | Method | Extracted model with the highest robustness | | | | | | Final model after extraction | | | | | |
|---|---|---|---|---|---|---|---|---|---|---|---|---|---|
| | | CA | rCA | RA | | | | CA | rCA | RA | | | |
| | | | | PGD20 | PGD100 | CW100 | AA | | | PGD20 | PGD100 | CW100 | AA |
| Label | Vanilla | 36.32 | 50.00 | 3.10 | 2.84 | 3.50 | 2.52 | 38.40 | 52.12 | 2.72 | 2.50 | 2.92 | 2.08 |
| | JBDA | 14.80 | 19.12 | 9.00 | 8.90 | 4.32 | 3.68 | 13.74 | 18.00 | 7.26 | 7.30 | 3.64 | 3.22 |
| | ARD | 20.02 | 30.02 | 11.80 | 11.82 | 11.16 | 10.58 | 19.40 | 29.24 | 10.92 | 10.90 | 10.02 | 9.66 |
| | IAD | 20.14 | 29.44 | 12.08 | 12.04 | 11.14 | 10.64 | 19.48 | 28.52 | 11.30 | 11.26 | 10.70 | 10.14 |
| | RSLAD | 28.46 | 41.04 | 13.08 | 12.90 | 12.82 | 11.82 | 27.54 | 40.04 | 12.22 | 12.14 | 12.06 | 11.14 |
| Logits | Vanilla | 4.92 | 5.56 | 1.78 | 1.56 | 0.30 | 0.12 | 5.08 | 5.90 | 1.72 | 1.46 | 0.32 | 0.12 |
| | JBDA | 1.00 | 1.12 | 0.92 | 0.94 | 0.02 | 0.02 | 1.72 | 1.78 | 0.16 | 0.10 | 0.04 | 0.00 |
| | ARD | 1.00 | 1.06 | 0.98 | 0.94 | 0.04 | 0.00 | 1.00 | 1.06 | 1.00 | 1.00 | 1.00 | 1.00 |
| | IAD | 1.00 | 1.06 | 1.00 | 0.98 | 0.46 | 0.22 | 1.00 | 1.06 | 1.00 | 1.00 | 1.00 | 1.00 |
| | RSLAD | 1.00 | 1.06 | 1.00 | 1.00 | 0.72 | 0.60 | 1.00 | 1.06 | 1.00 | 1.00 | 1.00 | 1.00 |
| BEST | | 30.64 | 44.60 | 10.86 | 10.64 | 11.70 | 10.24 | 31.58 | 45.62 | 10.10 | 9.98 | 10.90 | 9.54 |

Table 31: Results of model extraction attacks on CIFAR100. The victim model is ResNet-AT. The adversary model is VGG.

| | Method | Extracted model with the highest robustness | | | | | | Final model after extraction | | | | | |
|---|---|---|---|---|---|---|---|---|---|---|---|---|---|
| | | CA | rCA | RA | | | | CA | rCA | RA | | | |
| | | | | PGD20 | PGD100 | CW100 | AA | | | PGD20 | PGD100 | CW100 | AA |
| Label | Vanilla | 28.06 | 34.76 | 2.40 | 2.24 | 2.70 | 1.88 | 38.74 | 48.46 | 1.98 | 1.74 | 2.02 | 1.50 |
| | JBDA | 16.06 | 20.42 | 8.06 | 7.86 | 5.26 | 4.64 | 12.54 | 15.40 | 5.64 | 5.64 | 3.60 | 3.10 |
| | ARD | 21.30 | 28.80 | 11.50 | 11.48 | 10.26 | 9.76 | 20.38 | 27.96 | 10.70 | 10.72 | 9.86 | 9.30 |
| | IAD | 18.78 | 25.24 | 11.28 | 11.20 | 10.24 | 9.52 | 18.58 | 25.06 | 11.08 | 11.08 | 10.06 | 9.54 |
| | RSLAD | 25.14 | 33.44 | 12.14 | 12.02 | 11.50 | 10.82 | 25.38 | 33.74 | 11.12 | 11.12 | 11.04 | 9.96 |
| Logits | Vanilla | 1.34 | 1.50 | 1.82 | 1.78 | 0.22 | 0.20 | 5.48 | 5.88 | 1.46 | 1.38 | 0.52 | 0.08 |
| | JBDA | 1.78 | 1.74 | 1.08 | 1.06 | 0.70 | 0.62 | 2.56 | 2.80 | 0.64 | 0.52 | 0.10 | 0.08 |
| | ARD | 1.14 | 0.92 | 0.98 | 0.96 | 0.00 | 0.00 | 1.00 | 0.38 | 1.00 | 1.00 | 0.48 | 0.22 |
| | IAD | 1.08 | 1.10 | 0.98 | 0.94 | 0.02 | 0.02 | 1.00 | 1.00 | 1.00 | 1.00 | 0.98 | 0.98 |
| | RSLAD | 1.10 | 1.12 | 0.94 | 0.92 | 0.00 | 0.00 | 1.00 | 1.08 | 1.00 | 1.00 | 0.94 | 0.94 |
| BEST | | 32.60 | 43.54 | 10.02 | 9.78 | 10.62 | 9.30 | 32.62 | 43.76 | 9.90 | 9.80 | 10.76 | 9.32 |

Table 32: Results of model extraction attacks on CIFAR100. The victim model is WRN-AT. The adversary model is VGG.

| | Method | Extracted model with the highest robustness | | | | | | Final model after extraction | | | | | |
|---|---|---|---|---|---|---|---|---|---|---|---|---|---|
| | | CA | rCA | RA | | | | CA | rCA | RA | | | |
| | | | | PGD20 | PGD100 | CW100 | AA | | | PGD20 | PGD100 | CW100 | AA |
| Label | Vanilla | 31.54 | 41.96 | 3.26 | 3.06 | 3.50 | 2.58 | 35.42 | 45.90 | 2.68 | 2.46 | 2.76 | 2.08 |
| | JBDA | 16.04 | 20.80 | 9.32 | 9.22 | 5.16 | 4.64 | 13.66 | 17.20 | 7.44 | 7.36 | 4.36 | 3.92 |
| | ARD | 20.36 | 29.86 | 11.58 | 11.52 | 10.58 | 9.94 | 20.00 | 29.54 | 11.22 | 11.14 | 10.46 | 9.68 |
| | IAD | 18.96 | 28.34 | 11.48 | 11.40 | 10.66 | 10.12 | 18.96 | 27.70 | 10.72 | 10.74 | 10.26 | 9.56 |
| | RSLAD | 23.92 | 34.90 | 12.54 | 12.50 | 12.02 | 11.24 | 24.94 | 35.36 | 11.70 | 11.64 | 11.58 | 10.74 |
| Logits | Vanilla | 2.06 | 2.70 | 1.48 | 1.44 | 0.14 | 0.12 | 3.54 | 4.46 | 1.46 | 1.46 | 0.14 | 0.10 |
| | JBDA | 1.12 | 1.18 | 1.08 | 1.10 | 0.02 | 0.02 | 1.06 | 1.12 | 0.92 | 0.80 | 0.54 | 0.46 |
| | ARD | 0.96 | 1.48 | 1.00 | 1.00 | 0.00 | 0.00 | 1.00 | 1.54 | 1.00 | 1.00 | 0.00 | 0.00 |
| | IAD | 1.00 | 1.54 | 1.00 | 1.00 | 0.00 | 0.00 | 1.00 | 1.54 | 1.00 | 1.00 | 1.00 | 1.00 |
| | RSLAD | 0.94 | 1.48 | 0.98 | 0.94 | 0.00 | 0.00 | 1.00 | 1.54 | 1.00 | 1.00 | 0.32 | 0.44 |
| BEST | | 29.74 | 41.90 | 10.76 | 10.56 | 11.34 | 9.98 | 29.74 | 41.98 | 9.90 | 9.80 | 10.66 | 9.36 |

Table 33: Results of model extraction attacks on CIFAR100. The victim model is ResNet-TRADES. The adversary model is VGG.

| | Method | Extracted model with the highest robustness | | | | | | Final model after extraction | | | | | |
|---|---|---|---|---|---|---|---|---|---|---|---|---|---|
| | | CA | rCA | RA | | | | CA | rCA | RA | | | |
| | | | | PGD20 | PGD100 | CW100 | AA | | | PGD20 | PGD100 | CW100 | AA |
| Label | Vanilla | 33.80 | 43.22 | 3.18 | 2.90 | 3.62 | 2.50 | 35.92 | 46.08 | 2.82 | 2.62 | 3.00 | 2.24 |
| | JBDA | 17.88 | 23.00 | 9.30 | 9.32 | 6.40 | 5.32 | 14.84 | 18.94 | 7.84 | 7.74 | 4.98 | 4.24 |
| | ARD | 19.90 | 27.50 | 11.52 | 11.46 | 10.68 | 10.28 | 19.76 | 27.22 | 10.86 | 10.86 | 9.90 | 9.44 |
| | IAD | 18.58 | 25.80 | 11.42 | 11.34 | 10.20 | 9.82 | 18.32 | 24.80 | 10.60 | 10.60 | 9.82 | 9.44 |
| | RSLAD | 25.26 | 34.82 | 12.34 | 12.38 | 11.78 | 11.06 | 24.18 | 32.94 | 11.62 | 11.60 | 11.32 | 10.68 |
| Logits | Vanilla | 1.90 | 2.26 | 1.66 | 1.52 | 0.00 | 0.00 | 3.26 | 3.68 | 1.44 | 1.34 | 0.00 | 0.00 |
| | JBDA | 1.14 | 1.54 | 1.02 | 1.00 | 0.12 | 0.04 | 1.50 | 1.82 | 0.42 | 0.40 | 0.12 | 0.10 |
| | ARD | 1.00 | 1.02 | 1.00 | 0.98 | 0.02 | 0.04 | 1.00 | 1.02 | 1.00 | 1.00 | 1.00 | 1.00 |
| | IAD | 1.04 | 1.10 | 0.96 | 0.94 | 0.06 | 0.00 | 1.00 | 1.06 | 1.00 | 1.00 | 1.00 | 1.00 |
| | RSLAD | 1.00 | 1.06 | 1.00 | 1.00 | 1.00 | 1.00 | 1.00 | 1.06 | 1.00 | 1.00 | 0.24 | 0.06 |
| BEST | | 29.66 | 40.04 | 10.40 | 10.18 | 10.78 | 9.70 | 29.66 | 40.04 | 10.38 | 10.18 | 10.72 | 9.70 |

Table 34: Results of model extraction attacks on CIFAR100. The victim model is WRN-TRADES. The adversary model is VGG.

| | Method | Extracted model with the highest robustness | | | | | | Final model after extraction | | | | | |
|---|---|---|---|---|---|---|---|---|---|---|---|---|---|
| | | CA | rCA | RA | | | | CA | rCA | RA | | | |
| | | | | PGD20 | PGD100 | CW100 | AA | | | PGD20 | PGD100 | CW100 | AA |
| Label | Vanilla | 43.26 | 60.20 | 2.74 | 2.32 | 3.00 | 1.56 | 43.78 | 60.64 | 2.36 | 2.20 | 2.70 | 1.52 |
| | JBDA | 23.70 | 30.32 | 5.88 | 5.78 | 6.12 | 5.42 | 20.62 | 26.08 | 4.20 | 4.02 | 4.48 | 3.86 |
| | ARD | 33.30 | 48.40 | 16.34 | 16.22 | 15.56 | 14.64 | 31.60 | 46.46 | 13.36 | 13.36 | 12.82 | 12.10 |
| | IAD | 32.74 | 48.12 | 16.06 | 15.98 | 15.22 | 14.22 | 31.88 | 46.14 | 13.58 | 13.48 | 13.06 | 12.18 |
| | RSLAD | 34.14 | 50.10 | 15.14 | 14.98 | 14.32 | 13.42 | 32.00 | 47.20 | 13.18 | 13.06 | 12.44 | 11.82 |
| Logits | Vanilla | 37.24 | 46.90 | 1.74 | 1.42 | 1.04 | 0.36 | 37.32 | 47.10 | 1.70 | 1.40 | 1.00 | 0.40 |
| | JBDA | 12.02 | 14.48 | 1.26 | 1.08 | 1.10 | 0.52 | 11.60 | 13.76 | 0.98 | 0.84 | 0.94 | 0.54 |
| | ARD | 14.24 | 19.44 | 5.30 | 5.16 | 2.74 | 2.14 | 14.54 | 19.70 | 4.94 | 4.88 | 2.76 | 2.12 |
| | IAD | 17.84 | 25.32 | 11.00 | 10.96 | 9.12 | 8.52 | 16.66 | 23.98 | 10.06 | 10.04 | 8.24 | 7.66 |
| | RSLAD | 24.36 | 34.30 | 11.96 | 11.96 | 9.04 | 7.92 | 23.98 | 34.12 | 11.36 | 11.32 | 8.66 | 7.50 |
| BEST | | 41.48 | 60.74 | 12.84 | 12.68 | 13.44 | 12.20 | 41.40 | 60.84 | 12.64 | 12.50 | 13.34 | 12.04 |

Table 35: Results of model extraction attacks on CIFAR100. The victim model is ResNet-AT. The adversary model is MobileNet.

| | Method | Extracted model with the highest robustness | | | | | | Final model after extraction | | | | | |
|---|---|---|---|---|---|---|---|---|---|---|---|---|---|
| | | CA | rCA | RA | | | | CA | rCA | RA | | | |
| | | | | PGD20 | PGD100 | CW100 | AA | | | PGD20 | PGD100 | CW100 | AA |
| Label | Vanilla | 29.36 | 36.28 | 1.68 | 1.48 | 1.56 | 1.00 | 45.06 | 57.26 | 1.20 | 1.00 | 1.32 | 0.74 |
| | JBDA | 16.22 | 19.50 | 4.02 | 3.92 | 4.30 | 3.72 | 11.78 | 14.08 | 1.90 | 1.90 | 2.00 | 1.76 |
| | ARD | 33.72 | 46.22 | 15.66 | 15.56 | 14.72 | 13.64 | 32.18 | 43.90 | 12.72 | 12.64 | 11.96 | 11.24 |
| | IAD | 32.66 | 43.86 | 15.42 | 15.32 | 14.36 | 13.58 | 30.86 | 41.26 | 11.54 | 11.38 | 11.32 | 10.40 |
| | RSLAD | 34.24 | 45.96 | 15.28 | 15.18 | 14.26 | 13.46 | 32.98 | 44.96 | 12.98 | 12.92 | 12.38 | 11.68 |
| Logits | Vanilla | 39.54 | 46.90 | 1.36 | 1.14 | 0.82 | 0.34 | 39.10 | 46.70 | 1.28 | 1.06 | 1.02 | 0.36 |
| | JBDA | 12.60 | 14.48 | 0.96 | 0.82 | 0.84 | 0.44 | 8.64 | 9.66 | 0.42 | 0.42 | 0.50 | 0.20 |
| | ARD | 13.84 | 17.62 | 5.12 | 5.16 | 3.14 | 2.66 | 12.96 | 17.38 | 4.30 | 4.28 | 2.92 | 2.42 |
| | IAD | 17.72 | 23.18 | 10.62 | 10.58 | 8.96 | 8.30 | 16.92 | 22.26 | 9.30 | 9.30 | 7.62 | 7.10 |
| | RSLAD | 25.32 | 33.14 | 11.56 | 11.48 | 8.60 | 7.46 | 24.34 | 31.70 | 10.04 | 9.94 | 7.76 | 7.04 |
| BEST | | 42.72 | 57.38 | 12.00 | 11.72 | 12.78 | 11.26 | 42.20 | 56.84 | 11.38 | 11.26 | 12.32 | 10.76 |

Table 36: Results of model extraction attacks on CIFAR100. The victim model is WRN-AT. The adversary model is MobileNet.

| | Method | Extracted model with the highest robustness | | | | | | Final model after extraction | | | | | |
|---|---|---|---|---|---|---|---|---|---|---|---|---|---|
| | | CA | rCA | RA | | | | CA | rCA | RA | | | |
| | | | | PGD20 | PGD100 | CW100 | AA | | | PGD20 | PGD100 | CW100 | AA |
| Label | Vanilla | 29.44 | 39.94 | 2.34 | 2.06 | 2.10 | 1.48 | 42.22 | 55.66 | 1.90 | 1.66 | 1.98 | 1.10 |
| | JBDA | 20.22 | 26.28 | 5.34 | 5.24 | 5.50 | 5.06 | 18.84 | 23.08 | 3.96 | 3.92 | 4.14 | 3.72 |
| | ARD | 33.06 | 47.64 | 15.22 | 15.08 | 14.26 | 13.50 | 30.54 | 43.80 | 13.30 | 13.22 | 12.60 | 11.94 |
| | IAD | 30.46 | 44.24 | 15.00 | 14.92 | 14.14 | 13.24 | 29.76 | 42.48 | 12.94 | 12.92 | 11.98 | 11.40 |
| | RSLAD | 32.44 | 46.64 | 14.84 | 14.72 | 14.06 | 13.24 | 31.14 | 44.78 | 12.88 | 12.76 | 12.40 | 11.56 |
| Logits | Vanilla | 31.94 | 41.48 | 2.30 | 1.92 | 1.02 | 0.36 | 31.82 | 41.16 | 2.16 | 1.76 | 1.04 | 0.36 |
| | JBDA | 12.52 | 15.94 | 0.86 | 0.64 | 0.50 | 0.24 | 11.76 | 15.18 | 0.82 | 0.68 | 0.40 | 0.18 |
| | ARD | 11.62 | 17.60 | 5.06 | 4.90 | 2.68 | 2.12 | 11.54 | 17.48 | 4.68 | 4.66 | 2.66 | 2.24 |
| | IAD | 15.18 | 22.40 | 9.36 | 9.34 | 7.54 | 7.14 | 14.30 | 21.12 | 9.04 | 9.02 | 7.26 | 6.82 |
| | RSLAD | 21.06 | 30.50 | 10.24 | 10.16 | 7.22 | 6.54 | 20.82 | 30.06 | 9.74 | 9.78 | 7.06 | 6.60 |
| BEST | | 37.84 | 54.10 | 12.54 | 12.30 | 13.06 | 11.88 | 38.74 | 55.68 | 11.88 | 11.68 | 12.34 | 11.28 |

Table 37: Results of model extraction attacks on CIFAR100. The victim model is ResNet-TRADES. The adversary model is MobileNet.

| Method | | Extracted model with the highest robustness | | | | | | Final model after extraction | | | | | |
|---|---|---|---|---|---|---|---|---|---|---|---|---|---|
| | | CA | rCA | RA | | | | CA | rCA | RA | | | |
| | | | | PGD20 | PGD100 | CW100 | AA | | | PGD20 | PGD100 | CW100 | AA |
| Label | Vanilla | 30.18 | 39.02 | 2.36 | 2.16 | 2.26 | 1.40 | 43.10 | 55.04 | 1.48 | 1.28 | 1.50 | 0.94 |
| | JBDA | 20.64 | 26.36 | 5.44 | 5.32 | 5.74 | 5.12 | 14.54 | 18.28 | 2.36 | 2.34 | 2.48 | 2.16 |
| | ARD | 32.00 | 44.30 | 15.80 | 15.80 | 14.48 | 14.00 | 31.28 | 42.92 | 12.80 | 12.72 | 12.34 | 11.58 |
| | IAD | 32.04 | 43.88 | 15.76 | 15.68 | 14.64 | 14.02 | 30.66 | 42.22 | 12.16 | 12.12 | 11.56 | 10.94 |
| | RSLAD | 31.34 | 43.56 | 15.16 | 15.04 | 14.20 | 13.24 | 31.02 | 43.04 | 12.36 | 12.26 | 11.90 | 11.30 |
| Logits | Vanilla | 33.16 | 40.56 | 2.24 | 1.82 | 1.12 | 0.54 | 33.04 | 40.26 | 2.02 | 1.84 | 1.24 | 0.46 |
| | JBDA | 10.86 | 13.36 | 0.94 | 0.88 | 0.54 | 0.30 | 7.44 | 8.74 | 0.66 | 0.56 | 0.42 | 0.26 |
| | ARD | 10.92 | 15.72 | 3.90 | 3.80 | 1.96 | 1.66 | 10.56 | 15.14 | 3.26 | 3.32 | 1.96 | 1.58 |
| | IAD | 15.82 | 20.68 | 10.16 | 10.24 | 8.12 | 7.70 | 15.28 | 20.24 | 8.90 | 8.94 | 7.28 | 6.86 |
| | RSLAD | 22.00 | 29.50 | 10.94 | 10.82 | 7.66 | 6.84 | 21.10 | 28.34 | 9.56 | 9.48 | 6.94 | 6.12 |
| BEST | | 39.90 | 54.44 | 12.04 | 11.82 | 12.66 | 11.26 | 40.26 | 55.08 | 11.52 | 11.38 | 12.06 | 10.94 |

Table 38: Results of model extraction attacks on CIFAR100. The victim model is WRN-TRADES. The adversary model is MobileNet.

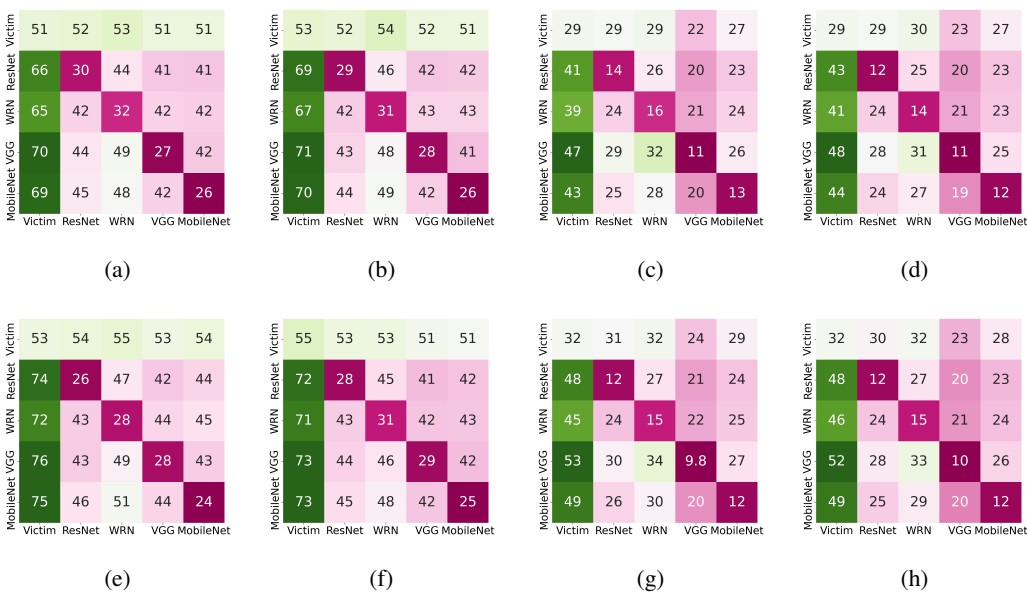

Figure 5: Transferability stabilization of our BEST. The dataset of (a), (b), (e) and (f) is CIFAR10. The dataset of (c), (d), (g) and (h) is CIFAR100. The victim model of (a) and (c) is ResNet-AT. The victim model of (e) and (g) is WRN-AT. The victim model of (b) and (d) is ResNet-TRADES. The victim model of (f) and (h) is WRN-TRADES. We generate adversarial examples by using PGD100. The vertical axis represents the model which generates adversarial examples. The horizontal axis represents the model which is attacked by other models' adversarial examples. The number inside each square is the prediction accuracy.