# OpenReview forum: "Extracting Robust Models with Uncertain Examples"
_ICLR.cc/2023/Conference — ICLR 2023 poster_

### Official Review · Reviewer_TSwB · 2022-10-23

**Confidence:** 4
**Correctness:** 3
**Technical Novelty And Significance:** 3
**Empirical Novelty And Significance:** 3
**Recommendation:** 6

**Clarity, Quality, Novelty And Reproducibility:**

Comments:

1. Figure 1c - combining the standard extraction followed by adversarial training on the attacker side can extract a good enough robust model but slightly decreases the accuracy. The assumption to limit the query budget to 5000 is arbitrary. It should be shown how the attack performs with more queries against the victim model.
2. Figure 1c and 1d - are the pre-trained models used in both cases as the starting point for the stolen copies?
3. Why is the performance of JBDA better than CloudLeak? (Figure 1b)
4. Attack cost - end of page 2. The other budget should be allocated for the training of the stolen copy. The adversary has to compare the total cost of extraction and the creation of the stolen copy with the cost of training a new model from scratch.
5. Figure 2 illustrates intuitively that the UE (uncertain examples) are the most informative for the stolen copy since they capture most of the knowledge about the decision boundaries.
6. Property \textbf{P1} is rather not clear. Figure 1b shows that JBDA potentially overfits too much to the adversarial examples since the clean accuracy is lower, but the connection to the robust feature is rather weak.
7. The formulation of the double minimization problem in Equation 2 on page 5 is not clear. Why do you include the minimization of the loss value using the stolen/attack model $M_{\mathcal{A}}$?
8. The results in Tables 2 and 3 are not useful since they do not compare with other methods but only present results for BEST.
9. It should be mentioned (at least in the appendix) - how the ARD, IAD, and RSLAD work.
10. According to ARD - "knowledge distillation using only natural images can preserve much of the teacher’s robustness to adversarial attacks (see Table 1), enabling the production of efficient robust models without the expensive cost of adversarial training." So, why doesn't the model stealing with clean samples extract the robustness of the victim model?
11. The results at the end of page 7 indicate that finding a simple uncertain example (using the generation of adversarial examples) is enough. This indicates that JBDA with FGSM might be further improved.
12. Note that standard vision and NLP tasks do offer pre-trained models. However, many real-world and specific tasks, for example, in the medical domain, do not have pre-trained models.
13. "There can be other robust solutions, e.g., certified defense" How would you extract a model that was made robust via a certified defense?

Neat:

- Threat model: is there any service that exposes logits? For example, Google's API responds with confidence scores that are not even softmax scores. Other APIs (such as Clarifai), return only labels.
- Figure 1 - it should be added that ResNet18 is used.
- At the end of page 4: it should be - every sample in $\mathcal{R}^N$ and not in $\mathcal{R}^n$.
- The quality of writing could be improved.
- The main text merges with the caption of Figure 3. The authors should not modify the standard latex setting to that extent.



**Strength And Weaknesses:**

Strength(s):

1. The paper shows many experimental results, however, it is not clear how the pre-trained model is used to initialize the stolen copy.
2. The idea of extracting both functionality and robustness from a victim model is novel.

Weaknesses:

1. There is no novelty in the proposed attack. This is obvious that the goal of an attacker is to extract the decision boundaries of the victim model.
1. There is a  robustness gap between the victim model and its stolen copy.
2. The difference between the combination of standard extraction with adversarial training vs BEST is tiny (compare Figure 1c and 1d).
3. The paper proposes only an attack and does not consider any possible mitigation/defense techniques against BEST.
4. The defense proposed in [1] slows down the attacks that generate uncertain queries (similar to UE) against the victim model. The defense makes the attack much more costly when the victim model is more uncertain about its predictions. Such a defense is very effective against BEST. A simple approach to check it would be to take the defense method even with the basic entropy cost (the information leakage of a given query is based on the entropy computed on the victim's softmax output for a given query) and then check how much slow down in the query time the defense would incur in comparison to attacking the undefended victim model.

**References:**

[1] "Increasing the Cost of Model Extraction with Calibrated Proof of Work" Adam Dziedzic, Muhammad Ahmad Kaleem, Yu Shen Lu, Nicolas Papernot, ICLR (International Conference on Learning Representations) 2022.


**Summary Of The Paper:**

This work proposes how to extract the accuracy/fidelity of a victim model and its robustness (if this property holds for the victim model). First, it is shown that standard attacks like "Vanilla" and Knockoff Nets cannot extract the robustness property of the victim model. Second, extracting models with adversarial examples can transfer (to some extent ~ 20%) of the robustness of the victim model when the JBDA attack is used, however, it slightly decreases the clean accuracy. Third, combining the standard extraction followed by adversarial training on the attacker side can extract a good enough robust model but slightly decreases the accuracy. Finally, the authors propose a new approach (called BEST - Binary Entropy Searching Thief) that queries the victim model with uncertain examples. These examples are close to the decision boundaries, which help to extract robustness property along with the accuracy of the victim model.

**Summary Of The Review:**

The paper shows how to extract a victim model with its accuracy and robustness. The experimental part is extensive. The insight is to use uncertain examples which lie on the junction of decision boundaries (not only adversarial which are usually on a decision boundary).

---

> ### Author Response · Authors · 2022-11-16
> **Response to Reviewer TSwB (1/n)**
>
> We thank Reviewer TSwB for comments and for providing thoughtful feedback on our work.
>
>
>
> >Q1: There is no novelty in the proposed attack. This is obvious that the goal of an attacker is to extract the decision boundaries of the victim model.
>
>
>
> We would like to clarify that our attack is novel and challenging. Specifically, (1) **this paper presents the first study towards robust model extraction.** Although a variety of techniques have been designed to steal a remote machine learning model with high accuracy and fidelity, how to extract a robust model with similar resilience against adversarial attacks is never investigated. (2) This goal is particularly challenging as previous works can either extract satisfactory clean accuracy or robustness, but not both, as discussed in Section 3. We propose a new type of data samples and algorithm to achieve efficient robustness extraction. (3) We appreciate that Reviewers DrkA, byXK and sinT also acknowledge the novelty of our work and think that the proposed problem is interesting.
>
>
>
> >Q2: A robustness gap
>
> Thanks for your comments. Indeed, there exists a gap between the victim model’s robustness and the restored model’s robustness. However, we would like to emphasize that with the same clean accuracy, our method outperforms the SOTA model extraction attacks with 30% robustness improvement. It is non-trivial and significant. Moreover, it is the first work that studies the feasibility of stealing clean accuracy and robustness simultaneously, which is challenging without any prior experience. This inevitably requires more future efforts to further improve its practicality.  We have discussed how to further improve the robustness in Appendix C.5, where we show that a significant improvement can be achieved by using additional out-of-distribution data to query the victim model. For example, when using additional data from STL10 to extract a victim model trained on CIFAR10, the clean accuracy increases about 3% and the robust accuracy under PGD20 increases about 5%. We will devote more efforts to reducing this gap in the future. We also believe this work can inspire more researchers to investigate this problem with better solutions.
>
>
>
> >Q3: The difference between the combination of standard extraction with adversarial training vs BEST is tiny (compare Figure 1c and 1d).
>
>  We would like to clarify that the performance gap between BEST and existing work is not tiny.  Comparing the models with the highest robust accuracy of Extraction-AT and BEST (see Figure 1c and Figure 1d), **it is clear that BEST outperforms one of Extraction-AT by up to about 10% in clean accuracy.** In the revision, we have added more comparisons between BEST and Extraction-AT in Appendix C.7 and Table 7 (see below). We adopt various model extraction techniques to first restore the victim model, and then use PGD adversarial training to train the restored model, and report results of the model with the highest robust accuracy as well as the final model. The victim model is ResNet18 trained with PGD-AT. The adversary’s model is ResNet18. The results prove that our method is better than Extraction-AT methods in terms of clean accuracy. This is consistent with our analysis in Section 3.
>
> |  |  |  |  |  |  |
> |:---:|:---:|:---:|:---:|:---:|:---:|
> | Dataset  | Method     | Extracted model with the highest robustness |  | Final model after extraction |  |
> |  |  | $CA$ | $RA$ | $CA$  | $RA$ |
> |  |  |  | PGD20 |  | PGD20 |
> | CIFAR10 | Vanilla + AT | 63.54 | 30.38 | 68.58 | 23.41 |
> |  | JBDA + AT | 61.96 | 32.92 | 68.00  | 24.62 |
> |  |  ARD + AT | 60.46 | 29.74 | 67.16 | 25.58  |
> |  |  IAD + AT | 61.44 | 29.12  | 65.74  | 25.50  |
> |  | RSLAD + AT | 60.34  | 30.20 | 65.70  | 23.62 |
> |  | BEST | 71.24 | 29.88 | 71.50  | 29.76 |
> | CIFAR100 | Vanilla + AT | 34.86  | 13.80 | 39.38  | 11.70 |
> |  | JBDA + AT | 36.52 | 15.50  | 38.92  | 12.32 |
> |  |  ARD + AT | 32.52 | 13.62 | 36.68 | 11.14  |
> |  |  IAD + AT | 32.80 | 13.68  | 37.04  | 12.86  |
> |  | RSLAD + AT | 36.72 | 13.32 | 37.20  | 12.66 |
> |  | BEST | 44.38 | 14.26 | 44.78  | 13.72 |

---

> > ### Author Response · Authors · 2022-11-16
> > **Response to Reviewer TSwB (2/n)**
> >
> > >Q4: The paper proposes only an attack and does not consider any possible mitigation/defense techniques against BEST.
> >
> >
> >
> > Thanks for the suggestion. **The main goal of this paper is to disclose the threat of robustness extraction, which is proposed for the first time.** We show existing state-of -the-art defenses do not work well to defeat our attack in Section 5.4 in the revision. As discussed in Section 6 (last point), we discuss some possible directions to mitigate BEST. We will consider them as future work.
> >
> > >Q5: The effectiveness of POW defense against our attack.
> >
> >
> >
> > Thanks for your valuable comments! We would like to discuss this defense against our attack from three perspectives (See Section 5.4 in our revision). (1) Under this defense, the financial cost is not increased. The adversary still submits the same number of queries to the victim model. (2) The defense can indeed increase the query time due to the introduction of POW. We conduct experiments to verify this. The victim model is ResNet18 trained with PGD-AT on CIFAR10, and the adversary’s model is ResNet18. From the table below, we can observe the query time of our BEST for each epoch is increased from 2 seconds to 21 seconds. However, we also need to consider the training time for this epoch, which is 17 seconds for ResNet-18 and 80 seconds for WideResNet-28-10. It means the total time cost for BEST is increased by (21+17)/(2+17) =2.00X and (21+80)/(2+80) =1.23X respectively, which is still acceptable. The results indicate that when restoring a robust model, the training time overhead for one epoch is trivial. (3) Just considering the query time, our BEST still has the least extra cost from POW, compared to other baseline methods.
> >
> > | **Adversary's Model** | **Method** | **Entropy** | **Gap** | **Pate** | **Query Time Before POW (Sec)** | **Query Time After POW (Sec)** | **Training Time Before POW (Sec)** | **Training Time After POW (Sec)** | **Training Time Overhead Ratio** |
> > |:---------------------:|:----------:|:-----------:|:-------:|:--------:|:-------------------------------:|:------------------------------:|:----------------------------------:|:---------------------------------:|:--------------------------------:|
> > | ResNet                | Standard   | 108.9       | 139.8   | 4.36     | 2                               | 4                              | -                                  | -                                 | -                                |
> > |                       | ARD        | 364.3       | 511.1   | 9.02     | 2                               | 26                             | 19                                 | 43                                | 2.26                             |
> > |                       | JBDA       | 390.9       | 524.2   | 9.84     | 2                               | 30                             | 19                                 | 47                                | 2.47                             |
> > |                       | BEST       | 356.8       | 493.9   | 8.21     | 2                               | 21                             | 19                                 | 38                                | 2.00                             |
> > | WRN                   | Standard   | 108.9       | 139.8   | 4.36     | 2                               | 4                              | -                                  | -                                 | -                                |
> > |                       | ARD        | 364.3       | 511.1   | 9.02     | 2                               | 26                             | 82                                 | 106                               | 1.29                             |
> > |                       | JBDA       | 390.9       | 524.2   | 9.84     | 2                               | 30                             | 82                                 | 110                               | 1.34                             |
> > |                       | BEST       | 356.8       | 493.9   | 8.21     | 2                               | 21                             | 82                                 | 101                               | 1.23                             |

---

> > > ### Author Response · Authors · 2022-11-16
> > > **Response to Reviewer TSwB (3/n)**
> > >
> > > >Q6: Figure 1c - combining the standard extraction followed by adversarial training on the attacker side can extract a good enough robust model but slightly decreases the accuracy. The assumption to limit the query budget to 5000 is arbitrary. It should be shown how the attack performs with more queries against the victim model.
> > >
> > >
> > >
> > > We are sorry for the unclear explanation of Figure 1c. In fact, the query budget of model extraction attacks is the amount of data in the extraction set $\times$ the number of extraction epochs (See Appendix B.1 in our revision).  Figure 1c shows the process of the adversarial training process, in which we train the model with adversarial training fashion on the extraction set under 200 training epochs. For the LabelDataset case, the query budget is equal to the size of the extraction set. For the FineTune case, the query budget is 5,000 * 200, as the adversary first performs a model extraction attack to obtain a model, in which the adversary uses 200 epochs to train a local model with his extraction set. In Table 7 of Appendix C.7, we show more results of the FineTune case. The results indicate that even with a very high query budget, the improvement of clean accuracy and robust accuracy is trivial. The reason is that we limit the power of the adversary and assume that he can only have limited data, which makes the baseline methods and adversarial training methods fail to provide sufficient information for model training. So, the results prove that our method is better than baselines and can provide models with more training information even with more query budget.
> > >
> > > >Q7: Figure 1c and 1d - are the pre-trained models used in both cases as the starting point for the stolen copies?
> > >
> > >
> > >
> > > Yes, the pre-trained models are the same if the adversary’s model structures are the same one. We add more clarifications about how the pre-trained models are used in our experiments in footnote 1 of page 3 and Appendix B.1 in the revision.
> > >
> > >
> > >
> > > >Q8: Why is the performance of JBDA better than CloudLeak? (Figure 1b)
> > >
> > >
> > >
> > > Thanks for your question. This stems from the distinct goals between JBDA and CloudLeak. CloudLeak is designed to only restore the clean accuracy, while JBDA aims to restore model’s boundaries. Specifically, CloudLeak is a non-active learning method. The adversary only generates AEs once and uses them to query the victim model. Then the adversary adopts these AEs with the returned labels to train his model. For JBDA, it is an active learning method, which means that the adversary will generate data before each training step. So, the restored model by JBDA has more similar boundaries as the victim model’s boundaries, making the restored model obtain partial robustness of the victim model. That is the reason that the robust performance of JBDA is better than CloudLeak. We have added this analysis in Appendix A.1 in the revision.
> > >
> > > >Q9: Attack cost
> > >
> > >
> > >
> > > Thanks for your valuable comments. We compare the total time cost of stealing a restored model and training a robust model from scratch to highlight the advantages. For a training set containing 5,000 data, we set the batch size as 128 and use SGD as our optimizer. When training a ResNet-18 on a single V100 card, the time cost for one epoch is 17s. When training a WideResNet-28-10 on a single V100 card, the time cost for one epoch is 80s. If the adversary adopts ResNet-18 as his model structure to extract a victim model (ResNet-18 or WideResNet-28-10), the time cost for 5,000 query budgets (including querying victim model and training local model) is 19s (without considering network latency). So, model extraction will cost less time when stealing a bigger model. The reason is that in our method, we adopt a similar training pipeline as in the adversarial training. First, the UE generation process is similar to the AE generation process. We only modify the loss function in the original AE generation process. So, the time cost for AE generation and UE generation is the same. Second, the model extraction requires the adversary to query the victim model, which will not cost too much time if we do not consider the network latency. Third, the model training process is the same as the adversarial training. Overall, model extraction attacks are more efficient when restoring a huge deep learning model. We add this analysis of the attack cost in Appendix C.2 in the revision.

---

> > > > ### Author Response · Authors · 2022-11-16
> > > > **Response to Reviewer TSwB (4/n)**
> > > >
> > > > >Q10: Property \textbf{P1} is rather not clear?
> > > >
> > > >
> > > >
> > > > Sorry for the confusion of our Property \textbf{P1}. The connection between JBDA and robust feature is that the technique used to augment the clean samples in JBDA is very close to the FGSM attack, which will introduce robust features into the generated samples. This property is clearer when we analyze the results of ARD, IAD and RSLAD. Straightforwardly, when comparing our method with ARD, IAD and RSLAD (these three attacks generate adversarial examples first, which contain robust features), we also find that the robust features in query samples of ARD, IAD and RSLAD cause overfitting. So, the Property \textbf{P1} can be proven from the experimental results. Our UEs do not contain robust features and obtain the robustness of the victim model by shaping the restored classification boundaries as the victim model's boundaries.  So, they will not cause overfitting. We add this analysis in Appendix C.8 in the revision.
> > > >
> > > >
> > > > >Q11: The formulation of the double minimization problem in Equation 2 on page 5 is not clear. Why do you include the minimization of the loss value using the stolen/attack model $M_A$?
> > > >
> > > >
> > > >
> > > > Sorry for the confusion of our method. Our UEs are generated based on the adversary’s restored model, iteratively. So, the uncertainty is mainly for the local model, instead of the victim model. And the adversary queries the victim model to obtain certain predictions to train his model, making his restored model have similar classification boundaries as the victim model. That is the reason that UEs are generated on the $M_A$. We add these details and reasons in Section 4.2 in the revision.
> > > >
> > > > >Q12: It should be mentioned (at least in the appendix) - how the ARD, IAD, and RSLAD work.
> > > >
> > > >
> > > >
> > > > Thanks for your suggestions. We add details of these methods in Appendix A.2 in the revision to better explain how they work. Specifically, ARD adopts PGD attack to generate adversarial examples for the student model and uses Kullback–Leibler divergence to minimize the differences between the predictions of the student model and the predictions of the teacher model for the clean data and adversarial examples. IAD further proposes an adaptive distillation process to overcome the challenge that the teacher model will return unreliable answers, with the increasing training time. IAD adopts PGD attack to generate adversarial examples for the student model and uses teacher guidance loss combined with a student introspection loss to train the student model. RSLAD replaces the PGD generated adversarial examples with TRADES generated adversarial examples in ARD to train the student model.
> > > >
> > > > >Q13: So, why doesn't the model stealing with clean samples extract the robustness of the victim model?
> > > >
> > > >
> > > >
> > > > Thanks for your valuable question. Compared with model extraction attacks, **users in the knowledge distillation have more information of the teacher model.**  In knowledge distillation, the user adopts the **same training set** as the teacher model’s one to train a student model. Furthermore, the user can **have the complete logits vectors** from the teacher model, so that the student model can be trained with Kullback–Leibler divergence or mean squared error loss to align the predictions with the teacher model. These information advantages enable the user to only rely on the clean data in the training set to obtain a similar decision boundary as the teacher model’s one. So, it is possible to obtain robustness only with clean data in knowledge distillation. However, in model extraction attacks, the adversary **does not have the same training set** as the victim model’s one. The data distribution shift prevents the adversary from obtaining the correct information (i.e., predicting behaviors) of the victim model on the training data. On the other hand, in most cases, the victim model **does not return the whole logits vectors**, and the adversary mainly adopts cross-entropy loss to train the local model, which further hampers the restored model from mimicking the victim model’s predicting behaviors. These two points make it impossible to restore the victim model’s robust accuracy, which can be found in the results of the vanilla method. We also analyze the differences between model extraction attacks and knowledge distillation in Appendix A.2 in the revision.

---

> > > > > ### Author Response · Authors · 2022-11-16
> > > > > **Response to Reviewer TSwB (5/5)**
> > > > >
> > > > > >Q14: This indicates that JBDA with FGSM might be further improved?
> > > > >
> > > > > Thanks for your comments. In fact, JBDA with FGSM will not further improve the model extraction results, due to two reasons. First, the samples generated by JBDA with FGSM are still adversarial examples, which will contain robust features. Based on our experimental analysis in Appendix C.8 in the revision, robust features have a close connection with robust overfitting. Second, JBDA with FGSM can only generate weak AEs. When considering ARD, IAD and RSLAD, which can generate AEs with PGD attacks, they adopt stronger AEs to extract the victim model’s robustness. Comparing the results between ARD, IAD, RSLAD and our method, we find that even using stronger AEs, ARD, IAD and RSLAD cannot beat our method. So, JBDA with FGSM will not outperform our method. Compared with our method, using AEs to query the victim model and train the local model will cause robust overfitting, hurting the restored model’s robustness. So, we adopt UEs to directly shape the classification boundaries to improve the robustness. We add this analysis in Appendix C.8 in the revision.
> > > > >
> > > > > >Q15: However, many real-world and specific tasks, for example, in the medical domain, do not have pre-trained models.
> > > > >
> > > > > Thanks for pointing it out. In Appendix C.3, we analyze the impact of the pre-trained models. The results indicate that even without using the pre-trained models, with limited data, our BEST can still obtain some robustness and clean accuracy. And we further study how to improve the extraction results in Appendix C.4. The results prove that our BEST is robust about the extraction data distribution, and the results can be improved by introducing out-of-distribution data.
> > > > >
> > > > > >Q16: Threat model
> > > > >
> > > > >
> > > > >
> > > > > Thanks for pointing it out. This setting is only for the baseline methods, to make the adversary have more knowledge about the victim model. For our BEST, we assume that the model can only return the hard label. The results indicate that even the adversary can obtain logits vectors, our BEST achieves the best clean accuracy and robustness.
> > > > >
> > > > >
> > > > >
> > > > >
> > > > >
> > > > > >Q17: Other minors
> > > > >
> > > > > Thanks for pointing out other minor flaws. We modify these minor errors and improve our paper’s readability.

---

> > > ### Comment · Reviewer_TSwB · 2022-12-05
> > > **POW defense**
> > >
> > > Thank you for the answers. What is the dataset used for stealing? Why do you report the query time per epoch? Why do we observe such low Entropy and Gap for uncertain examples?

---

> > > > ### Author Response · Authors · 2022-12-06
> > > > **Thanks for your comments!**
> > > >
> > > > We follow Table 3 in the paper ``Increasing the Cost of Model Extraction with Calibrated Proof of Work'' (we will call it POW paper for short in the following) to set up our experiments. For the query dataset, we adopt the extraction set from CIFAR-10, which is introduced in Section 5.1 of our paper. This setting is aligned with the JBDA setup in the POW paper.
> > > >
> > > > Due to the time limitation, we only test the number of queries for 5k data. Because from the official code [1] of POW work in GitHub, we find that the query time and the number of queries are approximately linear, which means that we can approximately infer the time for any number of queries.
> > > >
> > > > As for the lower Entropy and Gap for UEs, we think there are two reasons. First, the victim model in the experiments is ResNet-18, which is trained with PGD-AT. Second, our UEs are not AEs, so they will not fool the models. For the first reason, the model is robust, which means that it will persist in giving correct predictions with high confidence scores for inputs (not AEs) with perturbation. Under this situation, both entropy and gap will be low, as the model is confident in its predictions. For the second reason, the UEs are generated based on the adversary's model, and we find that the robust model can correctly classify them with high confidence scores (so the Entropy and Gap will be lower), which means that the UEs are not like AEs that have transferability. But the UEs can provide many informative supervised signals for the adversary's model, causing the adversary successfully restores a robust model.
> > > >
> > > > Hope our explanations can address your questions and concerns!
> > > >
> > > > [1] https://github.com/cleverhans-lab/model-extraction-iclr

---

> > > > > ### Comment · Reviewer_TSwB · 2022-12-06
> > > > > **POW: cost of queries and entropy**
> > > > >
> > > > > Thank you for your response.
> > > > >
> > > > > >*we find that the query time and the number of queries are approximately linear*
> > > > >
> > > > > First, the POW paper shows an exponential growth of the query cost with a higher number of queries, for instance, check  Figure 2 in the original paper on POW.
> > > > >
> > > > > >*the UEs are generated based on the adversary's model, and we find that the robust model can correctly classify them with high confidence scores (so the Entropy and Gap will be lower)*
> > > > >
> > > > > Second, the objective of the uncertain examples in Definition 1 (Section 4.1 in your paper) clearly suggests that the entropy increases with lower $\delta$. What is the $\delta$ used in your experiments and the performance of the stolen models for the experiments that you run in the above table (also Table 39 in your paper)?

---

> > > > > > ### Author Response · Authors · 2022-12-08
> > > > > > **Thanks for your comments!**
> > > > > >
> > > > > > >1. Thanks a lot for pointing this out! After double checking this paper carefully, we agree that the query cost grows exponentially with the number of queries. This defense work can indeed increase the cost of our attack significantly and disincentivize the adversary from stealing the model. We apologize for this mistake, and will correct this in the revision.
> > > > > > \
> > > > > > >Actually, POW is a very strong defense solution, and can defend weaken all existing query-based model extraction attacks. It is interesting to improve our attack to bypass this method. For instance, the adversary can try to behave like a normal user when querying the model. Since normal users may also possibly send normal images which the victim model has low confidence or uncertainty about their classes (e.g., images in the wild following different distributions from the training set), to reduce such false positives and make the service practical, the model owner should allow certain privacy budget for each account. Then the adversary can set up a large number of accounts and ensure the queries in each account will not exceed such privacy budget. Although the POW paper discussed that the adoption of multiple accounts can be defeated by summing over all the users, we believe the adversary can still succeed if he tries to mimic normal users for each account. The more accounts he has, the higher feasible it is to mimic normal users within the privacy budget. We will discuss such possible adaptive attack in the revision and consider it as an interesting future work.
> > > > > >
> > > > > > >2. Sorry for the confusion. Definition 1 gives a formal definition of $\delta$-UE. To generate UE practically, we do not need to explicitly set $\delta$, but try to minimize its value using Algorithm 1. Hence, for different samples, the corresponding $\delta$ will be different. Indeed, UEs can increase the entropy of the model’s predictions, but as we generate UEs on the adversary’s model, the effects on the robust victim model are trivial compared with AEs and JBDA samples. This is the reason that causes our UE to achieve lower entropy.
> > > > > > \
> > > > > > >For the performance of the restored model, we use the same setting as our main experiments in Section 5. Specifically, $\epsilon$=8/255, $\eta$=2/255 and $B_S$=10. So the restored model can achieve 71.24% $CA$, 82.12% $rCA$ and 29.88% $RA$ under PGD20 attack.

---

> > > > > > > ### Comment · Reviewer_TSwB · 2022-12-12
> > > > > > > **POW: Sybil attack**
> > > > > > >
> > > > > > > Please note that the authors of the POW paper explicitly addressed the concern of the Sybil attack in Section 4.4. A simple approach is to compute the cost per query instead of per user.

---

> ### Author Response · Authors · 2022-12-05
> **Response to Reviewer TSwB**
>
> Dear Reviewer TSwB,
>
> Thanks again for your comments! As the discussion period will end soon, could you please kindly check our responses and revisions? We believe that our responses and revisions have addressed your questions and concerns.
>
> Best regards,
>
> Authors

---

> ### Comment · Reviewer_TSwB · 2022-12-12
> **Final comments**
>
> Overall, the idea of stealing robust models is new, however, the motivation and the results (especially the robustness gap) could be improved. Additionally, the presentation of the paper and the consideration of the related work should be revised. The authors went the extra mile to address my comments, so I increased the score.

---

> > ### Author Response · Authors · 2022-12-13
> > **Thanks for your reply!**
> >
> > Thanks for your suggestions and comments! We will discuss with the authors of POW about the Sybil attack and discuss these details in our final revision to address your concerns. We will also polish our paper structure and the presentation of related works to make it clearer. Thanks again for your help during the rebuttal phase!

---

### Official Review · Reviewer_sinT · 2022-10-23

**Confidence:** 4
**Correctness:** 3
**Technical Novelty And Significance:** 3
**Empirical Novelty And Significance:** 2
**Recommendation:** 6

**Clarity, Quality, Novelty And Reproducibility:**

The paper is generally written clearly, and the threat model is clearly stated. The paper has novelty in assessing the robustness of a stolen model. The reproducibility is medium with some unclear parameters and settings such as how the pretrained model is used, what optimizer is used, and if the model is indeed trained on a single example in Line 12 in Algorithm 1.

**Details Of Ethics Concerns:**

The goal of an attack paper is to identify which defense works and which doesn't, and warn the user of a certain system. The victim can potentially consider the robustness as the advantage and a knockoff model with similar benign accuracy but low robustness fine but this point is not clear, and I think the paper needs to use some more warning voice.

**Strength And Weaknesses:**

Strengths.
- S1: The paper proposes a new metric to evaluate a solution to an existing problem.
- S2: The proposed approach uses a reasonable threat model and approach.
- S3: The proposed approach is evaluated with various parameters such as victim model architectures, two datasets, without robustness training by the victim model, and 5 baselines.

Weaknesses.
- W1: The paper is unclear on how the pretrained model was used, and when it was used, especially for the baseline attacks. Since this is a very important factor based on the appendix, this needs to be clarified.
- W2: The baselines do not include Extraction-AT which is seemingly the best alternative, and this approach might be tuned to perform better. But instead, this is only shown to motivate the problem with a limited setting.
- W3: The main varying parameter for each configuration is epochs, not budget. This limits the understanding of alternative approaches and overall utility as the attacker is more limited by the budget than the number of epochs. Also, the maximum epoch tested here looks rather a bit too high as ResNet-18 on CIFAR can train much faster. Does the attack augment the data somehow with more epochs despite the limited budget?

**Summary Of The Paper:**

This paper considers a model stealing problem through remote access such as Rest API, returning the prediction given an input by an attacker. While the scenario is generally similar to many existing works, this paper aims at stealing the robustness of the victim model while maintaining the stolen model accuracy. The main idea is to find delta-uncertain examples to query the victim model to examine the boundary of the model. The experiment shows the approach outperforms baselines in terms of stolen model's accuracy and robustness.

**Summary Of The Review:**

The paper is well written, and point out an interesting aspect of model stealing. The core of the approach sounds reasonable, although it is unclear if the conjecture is indeed true as the effect of delta-UE is only shown with the downstream task of model extraction. The paper also has many unclear points in the setting, such as the use of a pretrained model, and how the train part is performed. The baseline configurations are also unclear despite many experimental results in the appendix. The paper's performance is also based on the use of a pretrained model, which can have very high impact and such a model can carry innate robustness with a large pretrained data and being free from the adversarial example specific to the task. Although Jagielski et al. (2020a) briefly mentions the use of a pretrained model or available data on the web, it focuses on the data rather than the model. The model extraction has a lower impact for such a task with a pretrained model available, compared to many proprietary models such as credit evaluation model. Overall, this paper explores an interesting new metric to evaluate an existing problem, but it also has some unclear descriptions and limited impact lowering the rating.

-----
After reading the rebuttal and revision, I updated my recommendation to weak accept. I think the paper is now thoroughly prepared to be presented at a conference. The paper is generally well written, although a lot of contents had overflow into the appendix, and the comparison with many commonsense baselines is thorough. The problem is novel as well as the solution. My only concern to reserve a strong accept is the practicality as the robustness of a stolen model is usually afterthought, and the defender wants to prevent stealing in general, not specifically the robustness. As the ultimate goal of adversarial research is to understand threats and protect an asset from them, this might pose a limited impact. Also, the empirical result shows small improvement only despite beating the baselines.

---

> ### Author Response · Authors · 2022-11-16
> **Response to Reviewer sinT (1/3)**
>
> We thank Reviewer sinT for comments and for providing thoughtful feedback on our work.
>
>
>
> >Q1: The paper is unclear on how the pretrained model was used, and when it was used, especially for the baseline attacks
>
>
>
> Sorry for the lack of details. The pretrained model is used to initialize the adversary’s local model, which means that the adversary uses a pretrained model as a start to restore the victim’s model. All baseline methods and our attack follow the same attack pipeline, i.e., using the same pre-trained model as a start and restoring the victim model by training the pre-trained model with the query data. Specifically, all the pre-trained models are downloaded from the open repository from GitHub. The pre-trained models are trained on Tiny-ImageNet. There are four network structures for the pre-trained models, i.e., ResNet, VGG, MobileNet and WideResNet. We have added these details in Appendix B.1 in the revision.
>
>
>
> >Q2: The baselines do not include Extraction-AT
>
> Thanks for the suggestions. In the revision, we have added the comparisons between BEST and Extraction-AT in Appendix C.7 and Table 7 (see below). We adopt various model extraction attacks to first restore the victim model, and then use adversarial training (AT) to train the restored model and report results of the model with the highest robust accuracy as well as the final model. The victim model is ResNet18 trained with PGD-AT. The adversary’s model is ResNet18. The results prove that our method is better than Extraction-AT methods in terms of clean accuracy. This is consistent with our analysis in Section 3.
>
>
> |  |  |  |  |  |  |
> |:---:|:---:|:---:|:---:|:---:|:---:|
> | **Dataset**  | **Method**     | **Extracted model with the highest robustness** |  | **Final model after extraction** |  |
> |  |  | $CA$ | $RA$ | $CA$  | $RA$ |
> |  |  |  | PGD20 |  | PGD20 |
> | CIFAR10 | Vanilla + AT | 63.54 | 30.38 | 68.58 | 23.41 |
> |  | JBDA + AT | 61.96 | 32.92 | 68.00  | 24.62 |
> |  |  ARD + AT | 60.46 | 29.74 | 67.16 | 25.58  |
> |  |  IAD + AT | 61.44 | 29.12  | 65.74  | 25.50  |
> |  | RSLAD + AT | 60.34  | 30.20 | 65.70  | 23.62 |
> |  | BEST | 71.24 | 29.88 | 71.50  | 29.76 |
> | CIFAR100 | Vanilla + AT | 34.86  | 13.80 | 39.38  | 11.70 |
> |  | JBDA + AT | 36.52 | 15.50  | 38.92  | 12.32 |
> |  |  ARD + AT | 32.52 | 13.62 | 36.68 | 11.14  |
> |  |  IAD + AT | 32.80 | 13.68  | 37.04  | 12.86  |
> |  | RSLAD + AT | 36.72 | 13.32 | 37.20  | 12.66 |
> |  | BEST | 44.38 | 14.26 | 44.78  | 13.72 |

---

> > ### Author Response · Authors · 2022-11-16
> > **Response to Reviewer sinT (2/3)**
> >
> > >Q3: The main varying parameter for each configuration is epochs, not budget
> >
> >
> >
> > We are sorry for the unclear explanation of the query budget. In fact, for all the methods in our experiments, the number of epochs is proportional to the query budget, as we use all data in the extraction set in one training epoch (see Appendix B.1 in the revision). For instance, 1 training epoch means the query budget of 5,000, and 10 training epoch means the query budget of 50,000. We also discuss the impact of query budget in Appendix C.2 in the revision. We believe the budget does not incur high financial cost in practice, as many MLaaS providers offer certain free queries for users. For example, AWS [1] provides a Free Tier for new accounts to analyze 5,000 images per month for free. Google [2] provides all accounts with a discount of predicting 1,000 images per month free. Microsoft [3] provides all accounts with a discount of predicting 5,000 images per month for free.  It is feasible to use more accounts to steal the victim model, which can significantly reduce the financial cost as creating new accounts is easy and trivial. Hence, the query budget is not the principal limitation in model extraction attacks.
> >
> > >Q4: Does the attack augment the data somehow with more epochs despite the limited budget? No explanations.
> >
> > Sorry for the confusion of the attack. Augmenting the query data is very efficient without introducing more epochs. So it will not affect the budget.
> >
> >
> >
> > >Q5: The baseline configurations are also unclear despite many experimental results in the appendix.
> >
> >
> >
> > Sorry for the lack of details. In this revision, we have added more configuration details in Appendix B.1, where the optimizer in all experiments is SGD, with start learning rate 0.1, momentum 0.9 and weight decay 0.0001. We also introduce ARD, IAD and RSLAD with more details in Section A.2 in the revision. We have uploaded the code for better reproducibility.
> >
> >
> >
> > >Q6: if the model is indeed trained on a single example in Line 12 in Algorithm 1.
> >
> >
> >
> > Sorry for the confusion. Line 12 represents that the training process is on a batch of data instead of a single example. Specifically in Algorithm 1, we use data $X$ with the batch size of 128, which means that the adversary first queries the victim model with 128 data samples, and then uses them to train his local model. Then, the adversary adds 128 to the query budget $b_q$. We have added the details in Section 4.2 in the revision.

---

> > > ### Author Response · Authors · 2022-11-16
> > > **Response to Reviewer sinT (3/3)**
> > >
> > > >Q7: The paper's performance is also based on the use of a pretrained model, which can have very high impact and such a model can carry innate robustness with a large pretrained data and being free from the adversarial example specific to the task.
> > >
> > > Sorry for the confusion, and thanks for your good question. In fact, using pre-trained models does not affect the superiority of our method, for the following reasons. First, all the baseline methods adopt the same pre-trained model to initialize the adversary’s model. This gives us very fair comparisons. Second, the adopted pre-trained models are normal without any robustness features. This explains why other baseline methods using these models cannot extract the robustness of the victim model (Section 5.2 and Appendix C.8). Third, we prove that even without pre-trained models, our method can still restore robustness from the victim models. The main reason is that using UEs to query the victim model can obtain the most informative outputs from the victim model and using such samples to train the adversary’s model can better shape the classification boundaries to fit the victim model’s boundaries, which makes the adversary’s model achieve higher robustness. We have added these details in Appendix C.3 in the revision.
> > >
> > >
> > > [1] https://aws.amazon.com/rekognition/pricing/?nc1=h ls
> > >
> > > [2] https://cloud.google.com/vision/product-search/pricing
> > >
> > > [3] https://azure.microsoft.com/en-us/pricing/details/cognitive-services/computer-vision/

---

### Official Review · Reviewer_byXK · 2022-10-25

**Confidence:** 3
**Correctness:** 4
**Technical Novelty And Significance:** 3
**Empirical Novelty And Significance:** 3
**Recommendation:** 6

**Clarity, Quality, Novelty And Reproducibility:**

The paper is very well written and is easy to follow. The proposed idea of leveraging uncertain examples to effectively extract models is novel.

**Strength And Weaknesses:**

Strength:
1. the problem stealing robust models is interesting and is of practical interest in future.
2. the proposed method outperforms existing baseline attacks significantly.
3. the evaluations are quite comprehensive.

weakness:
1. the robustness of the extracted model is still very low and is not useful in practice. Also, the performance gap between the proposed approach and best performing baseline is not that significant.

**Summary Of The Paper:**

This paper studies the problem of extracting robust models by black-box queries to the model. The authors find that questing uncertain samples with respect to the current local model are helpful for extracting the decision boundary of the remote (black-box) victim model. Empirical results show that the proposed method can extract models with competitive clean and robust accuracy while existing baselines failed to preserve the robustness of the remote model after extraction.

**Summary Of The Review:**

Overall, I like the idea of the leveraging uncertain samples for efficient model extraction. The empirical results also support the main claims in the paper well. The only concern here is, although the proposed method works better than the baselines, the robustness of extracted models are not high, and is of limited practicality. Therefore, a weak accept is recommended.

---

> ### Author Response · Authors · 2022-11-16
> **Response to Reviewer byXK**
>
> We thank Reviewer byXK for comments and for providing thoughtful feedback on our work.
>
>
>
> >Q1: The robustness of the extracted model is still very low and is not useful in practice. Also, the performance gap between the proposed approach and best performing baseline is not that significant.
>
>
>
> Thanks for your comments. Indeed, there exists a gap between the victim model’s robustness and the restored model’s robustness. However, we would like to emphasize that with the same clean accuracy, our method outperforms the SOTA model extraction attacks with 30% robustness improvement. It is non-trivial and significant. Moreover, it is the first work that studies the feasibility of stealing clean accuracy and robustness simultaneously, which is challenging without any prior experience. This inevitably requires more future efforts to further improve its practicality.  We have discussed how to further improve the robustness in Appendix C.5, where we show that a significant improvement can be achieved by using additional out-of-distribution data to query the victim model. For example, when using additional data from STL10 to extract a victim model trained on CIFAR10, the clean accuracy increases about 3% and the robust accuracy under PGD20 increases about 5%. We will devote more efforts to reducing this gap in the future. We also believe this work can inspire more researchers to investigate this problem with better solutions.

---

> > ### Comment · Reviewer_byXK · 2022-12-04
> > **Thanks for the clarification**
> >
> > Thanks for the clarification. Robustness improvement is a limiting factor, but I also agree with the authors that showing some promising signs as a preliminary result is acceptable.

---

> > > ### Author Response · Authors · 2022-12-05
> > > **Thanks for your reply!**
> > >
> > > Thanks for your suggestions and comments!

---

### Official Review · Reviewer_DrkA · 2022-10-26

**Confidence:** 4
**Correctness:** 4
**Technical Novelty And Significance:** 3
**Empirical Novelty And Significance:** 4
**Recommendation:** 8

**Clarity, Quality, Novelty And Reproducibility:**

As mentioned before the paper is very well written and organized. The background and the related work are covered quite well. The description of the proposed method is well motivated and very intuitive. In the experiments, the experimental settings and the results are well detailed, which helps with the reproducibility of the results.
In terms of novelty, the paper aims at analyzing the problem of model extraction attacks by taking into account robustness. Some of the previous works have analyzed this aspect, but not with the angle given in this paper.


**Strength And Weaknesses:**

Strengths:
+ The paper is well motivated, organized and written. It reads very well. The description of the method and the rationale for crafting the “uncertain examples” are very clear and intuitive.
+ The way the attack is intended to retain the model’s robustness is novel. Yet some of the previous works are capable of retaining part of the victim’s model robustness, the proposed method is capable of maintaining a good trade-off between accuracy and robustness and it is capable of working relatively well with more restrictive threat models.
+ The experimental evaluation is very comprehensive and helps to convince the reader about the benefits of the proposed approach and its limitations. In this sense, the authors present an honest work recognizing the need for investigating more advanced model extraction methods capable of reducing the gap between accuracy on clean and adversarial examples. Nevertheless, I think the paper provides interesting insights and can foster further research in this area.

Weaknesses:
+ Although the experimental evaluation is very comprehensive, I think that the results in Table 1 should be discussed further, highlighting more the good results obtained with BEST compared to the other state-of-the-art methods, especially taking into consideration that some of them rely on more restrictive assumptions than BEST.
+ I think that the discussion about the ability of BEST to bypass defenses should be included in the main paper rather than in the appendices (by, for instance, moving some of the results in 5.3 to the appendix). I think this is an important aspect of the attack that, perhaps, deserves to be better emphasized in the paper.


**Summary Of The Paper:**

The paper introduces a novel method for performing model extraction attacks capable of not only retaining a good performance but also capturing the robustness of the target model when using adversarial training. The proposed attack, which can be applied under a very practical (and restrictive) threat model, creates “uncertain examples” for querying the victim’s model. These examples are designed so that they have quite uniform confidence scores across the different classes in the target classifier. The experimental results show the benefits of the proposed approach in keeping a good trade-off between accuracy and robustness when compared with existing state-of-the-art attacks. The experiments also provide a comprehensive analysis of the behavior of the proposed method with different configurations of datasets, models and types of adversarial training and attacks, which help to convince the readers about the benefits of the method.

**Summary Of The Review:**

I find the paper quite interesting and opens the door to the investigation of model extraction attacks capable of retaining both clean and adversarial accuracy. Although the experiments show that there is still a gap for retaining the adversarial accuracy, this paper proposes the first attack that is really designed to extract the robustness of the victim’s model while keeping the clean accuracy. It is also interesting to observe that the attack is also capable of bypassing some existing defenses, although the analysis with respect to this is shallower in the paper. Nevertheless, the comprehensive experimental evaluation is quite convincing to show the benefits (and limitations) of the proposed approach.

---

> ### Author Response · Authors · 2022-11-16
> **Response to Reviewer DrkA**
>
> We thank Reviewer DrkA for comments and for providing thoughtful feedback on our work.
>
>
> >Q1: Results in Table 1 should be discussed further, highlighting more the good results obtained with BEST compared to the other state-of-the-art methods
>
>
>
> Thanks for your suggestion. We have added further analysis in Appendix C.8 in the revision to better highlight the advantages of our method. From the results, we can find that our BEST has three main advantages compared to other baselines. First, BEST can restore high clean accuracy and relative clean accuracy, which is impossible for robust knowledge distillation methods (e.g., ARD, IAD, RSLAD).  Second, BEST can obtain higher robustness under limited clean data when restoring a robust victim model. Third, BEST can relieve the annoying robust overfitting problem.  Overall, BEST is better than previous baselines and achieves higher clean accuracy and robust accuracy under limited clean data.
>
>
>
>
>
> >Q2: The ability of BEST to bypass defenses should be included in the main paper rather than in the appendices
>
>
>
> Thanks for your suggestion. We have included more experiment results of defenses in the revision (refer to Section 5.4) and moves the discussion of using out-of-distribution data in Section 5.3 to Appendix C.4.  The defenses include perturbations [1], PRADA [2] and SEAT [3] to demonstrate the performance of BSET under various defenses. Experimental results show that BEST can bypass these defenses to obtain the desired attack effectiveness, indicating that our method is robust against various defenses.
>
>
>
> [1] Taesung Lee, Benjamin Edwards, Ian M. Molloy, and Dong Su. Defending against neural network
> model stealing attacks using deceptive perturbations. In Proc. of the SPW, pp. 43–49, 2019.
>
> [2] Mika Juuti, Sebastian Szyller, Samuel Marchal, and N Asokan. Prada: protecting against dnn model
> stealing attacks. In Proc. of the EuroS&P, pp. 512–527, 2019.
>
> [3] Zhanyuan Zhang, Yizheng Chen, and David Wagner. Seat: Similarity encoder by adversarial training
> for detecting model extraction attack queries. In Proc. of the ACM Workshop on AIS, pp. 37–48,
> 2021b.

---

> > ### Comment · Reviewer_DrkA · 2022-11-21
> > **Response to the authors' comments**
> >
> > Thank you very much for your clarifications. I have also checked the other reviewers' commennts and your replies. Some of the reviewers made good observations about the impact of the pretrained models, but I think that the replies provided by the authors are reasonable. I think that the paper is interesting and the experimental results validate the usefulnenss of the proposed approach and provide useful insights. Thus, I decided to keep my score.

---

> > > ### Author Response · Authors · 2022-11-22
> > > **Thanks for your reply!**
> > >
> > > Thanks for your suggestions and comments!

---

### Decision · Program_Chairs · 2023-01-20

**Decision:**

Accept: poster

**Justification For Why Not Higher Score:**

There is earlier work on crafting examples around the decision boundaries of a model so the method is not completely novel. The application is novel. The method doesn't work so well in stealing robustness properties of the target model.

**Justification For Why Not Lower Score:**

The paper is the first work to study the feasibility of stealing both robustness and clean functionality of a target model.

**Metareview: Summary, Strengths And Weaknesses:**

The paper introduces a method for model extraction attacks capable of stealing both robustness and functionality from the victim model. The proposed attack works by creating uncertain examples for querying the victim’s model. Most of the concerns the reviewers had in the initial reviews were clarified by the author response. While the method still has a big gap between the robustness of target model and the stolen model, reviewers agree it is a good first attempt in studying an attack for stealing a model from both robustness and clean accuracy perspective and may inspire future work in the direction.

**Note From Pc:**

if the above contains the word "oral" or "spotlight" please see: "oral" presentation means -> notable-top-5% and "spotlight" means -> notable-top-25%. As stated in our emails, we are disassociating presentation type from AC recommendations